# Cooperative TD Learning in Heterogeneous Environments via Joint Linear Approximation

## Abstract

We study cooperative temporal-difference (TD) learning with heterogeneous agents, in which a collection of agents interacts with different environments and jointly learns their respective value functions. We focus on the setting where there exists a shared linear representation and the agents' optimal weights collectively lie in an unknown linear subspace. Two competing intuitions exist: On the one hand, the existence of a common structure, even if unknown, may accelerate the learning of individual agents. On the other hand, heterogeneity in state transition kernels can lead to misaligned learning signals across agents, which may significantly hinder convergence and impair generalization. This raises a natural question: do the benefits of collaboration outweigh the drawbacks – or is it the other way around? In this paper, we take a step toward answering this question. Inspired by the recent success of personalized federated learning (PFL), we study the convergence of cooperative single-timescale TD learning in which agents iteratively estimate the common subspace and local heads. We showed that this decomposition can filter out conflicting signals, effectively mitigating the negative impacts of "misaligned" signals. The main technical challenges lie in the heterogeneity, the Markovian sampling, and their intricate interplay in shaping error evolutions. Specifically, not only are the error dynamics of multiple variables are closely interconnected, but there is also no direct contraction for the principal angle distance between the optimal subspace and the estimated subspace. We hope our analytical techniques can be useful to inspire research on deeper exploration into leveraging common structures.

## 1 Introduction

In many real-world applications, ranging from simple assistive robotics to autonomous vehicles, autonomous agents operate in different local environments. For example, consider the simple robot vacuums. Their interacting environments are mostly shaped by the household conditions, which can vary significantly in floor plans, obstacle types (e.g., moving humans and furniture), and spatial configurations. Similarly, autonomous vehicles deployed across different regions of a metropolitan area may encounter substantial variability in road conditions and traffic patterns. Such environmental heterogeneity can lead to misaligned learning signals across agents, which may significantly hinder convergence and impair generalization of jointly learning. Nevertheless, applying standard single-agent reinforcement learning (RL) solutions may lead to overall intensive, yet possibly redundant, computation and sample collection. This is because the cumulative knowledge obtained by one agent may still be partially useful to others when some common structure (though unknown) exists. These two competing intuitions raise a natural question:

*Question: Do the benefits of collaboration outweigh the drawbacks – or is it the other way around?*

In this paper, we take a step toward answering this question by focusing on mitigating the negative impacts, while deferring a deeper exploitation of the common structure for future work, for which a fundamental fine-grained problem setup may be necessary – see Section 6 for a detailed discussion.

Existing studies of multi-agent RL (MARL) with heterogeneous environments are mostly restricted to training a common policy or value-/Q-function (Jin et al., 2022; Wang et al., 2023; Xie & Song, 2023; Zhang et al., 2023a), and the optimality is often described with respect to some imaginary environment constructed as a weighted average of the agents' state-transition kernels Jin et al. (2022).

Though this approach has practical value – particularly when environmental heterogeneity is mild – in many applications (such as on-device recommender systems (Maeng et al., 2022) and smart healthcare (Nguyen et al., 2021)), heterogeneity is often moderate to severe, which can significantly degrade the performance of a common policy. In a parallel line of research on multi-agent Markov games or competitive MARL, learning personalized (or localized) policies or value-/Q-functions has recently garnered considerable attention (Filar & Vrieze, 2012; Zhang et al., 2018; Qu et al., 2022; Zhang et al., 2023b; Daskalakis et al., 2023). Specifically, in their setting, agents interact with one another within a shared environment, and the rewards received by individual agents are determined by the joint actions of all agents. It is easy to see that in such a setup, the more agents there are, the more complicated an agent's decision problem becomes. In fact, this line of work often suffers from the so-called curse of multi-agents—the sample complexity increases linearly or even exponentially (Song et al., 2022; Jin et al., 2024; Zhang et al., 2018) with respect to the number of agents.

In this paper, we take a step toward addressing the above question by studying cooperative temporal-difference (TD) learning, where a collection of agents collaboratively learn their heterogeneous value functions through local TD updates. Inspired by the recent success of personalized federated learning (PFL), we focus on the specific underlying common structure that the agents' optimal weights under a shared linear representation collectively lie in a low-dimensional linear subspace (Collins et al., 2021; Niu et al., 2024). We propose and analyze the convergence of a cooperative single-timescale TD method in which agents iteratively estimate the common subspace and agent-specific heads. We showed that this decomposition can filter out conflicting signals while amplifying shared structures. Extending the analysis from the PFL to the TD is fundamentally nontrivial. In addition to the Markovian sampling noises that are typically encountered in RL analysis, a unique and major difficulty that arises from environmental heterogeneity lies in the fact that the errors caused by the principal angle distance between the underlying true subspace and the estimated subspace get further distorted and in an unstructured, heterogeneous way.

Our main contributions are summarized as follows:

- We propose and analyze the convergence of a cooperative single-timescale TD method in which agents iteratively estimate the common subspace and agent-specific heads. We provide a finite-time analysis under Markovian sampling. We show that the overall reward estimate errors decay at a rate of $\mathcal{O}(\log T/\sqrt{T})$, and subspace estimation errors decay at a rate $\mathcal{O}(\frac{r \log T}{\sqrt{T}})$, where $r$ is the dimension of the common subspace.

- On the positive side, these rates do not decrease in the number of agents, implying that our method is resilient to the environmental heterogeneity. under the heterogeneity. Nevertheless, no clear acceleration is shown. We conjecture that this may stem from the widely adopted bounded feature assumption (Assumption 3). In PFL, the existence of a shared common subspace helps to reduce the local learning problem from a $d$-dimensional problem to an $r$-dimensional problem. When $r \ll d$, this simplification can be substantial. Under Assumption 3, however, the dependence on $d$ is obscured, which may explain the absence of an explicit speedup in our bound. Relaxing this assumption, however, requires a fundamental rethinking of the problem, which we leave to future work.

- We extend the theoretical understanding of personalized value function learning by providing guarantees under commonly used step sizes (i.e., $\Theta(1/\sqrt{T})$). A detailed discussion of related work on personalization in heterogeneous environments can be found in Section 2.

- Technically, the Markovian sample of TD error makes it difficult to obtain a direct contraction for the principal angle distance between the optimal subspace and the estimated subspace, but only an indirect subtraction in terms of the local weights error. To address this challenge, we show that the local weights error can be lower-bounded by the principal angle distance times a constant that depends on the diversity of the optimal local weights.

## 2 RELATED WORK

**Convergence speedup of MARL under homogeneous environments.** When the environments of all agents are homogeneous, it has been shown that the federated version of classic reinforcement learning algorithms can significantly alleviate the data collection burden on individual agents (Woo

et al., 2023; Khodadadian et al., 2022). Specifically, the per-agent sample complexity decreases linearly in terms of the number of agents – an effect commonly referred to in existing literature as linear speedup. Recent work has made significant progress in understanding fed RL under homogeneous environments. Woo et al. (2023) studied federated Q-learning. In addition to a linear speedup, Woo et al. (2023) uncovered the blessing of heterogeneity in terms of state-action exploration – a completely different notion of heterogeneity from our focus. Salgia & Chi (2025) studied the sample-communication complexity trade-off, and developed a new algorithm that achieves order-optimal sample and communication complexities.

**MARL under heterogeneous environments.** There is an emerging interest in mathematically understanding the role of environmental heterogeneity in the performance of the federated versions of classic reinforcement learning algorithms (Jin et al., 2022; Wang et al., 2023; Xie & Song, 2023; Zhang et al., 2023a), yet most existing studies focus on training a common policy or approximating a common function. Jin et al. (2022) studied federated Q-learning and policy gradient methods, assuming known transition kernels. Wang et al. (2025) studied federated Q-learning with unknown transition kernels, and found that, in the presence of environmental heterogeneity, the eventual convergence rate may depend solely on the number of communication rounds – multiple updates within each round cannot accelerate convergence. Wang et al. (2023) proposed FedTD(0) with linear function approximation, and linear convergence speedup in a low-heterogeneity regime. Xie & Song (2023) used the KL-divergence to penalize the deviation of local update from the global policy, and proved that under the setting of heterogeneous environments, the local update is beneficial for global convergence using their method. Yet, no convergence rate is provided. Zhang et al. (2024) proposed FedSARSA and proved that the algorithm can converge to a near-optimal solution. Neither (Xie & Song, 2023) nor (Zhang et al., 2024) characterized sample complexity.

**Model/Policy personalization.** In FL, model personalization (often referred to as PFL) has garnered significant attention in recent years. The success of personalization techniques depends on balancing the bias introduced by using global knowledge that may not generalize to individual clients, and the variance inherent in learning from limited local datasets. Popular PFL techniques include regularized local objectives (T Dinh et al., 2020; Li et al., 2021), local-global parameter interpolation (Deng et al., 2020), meta-learning (Fallah et al., 2020; Jiang et al., 2019), and representation learning (Collins et al., 2021; Xu et al., 2023). Extending these ideas of PFL to RL introduces an even greater level of difficulty that arises from the inherent sample correlation induced by sequential decision-making, and the non-stationarity of sample quality as the policy evolves.

In the RL literature, training personalized (or localized) policies has recently garnered considerable attention in the context of multi-agent Markov games or competitive MARL Filar & Vrieze (2012); Zhang et al. (2018); Qu et al. (2022); Zhang et al. (2023b); Daskalakis et al. (2023), which depart fundamentally from our focus. Specifically, in their setting, agents interact with one another within a shared environment, and the rewards received by individual agents are determined by the joint actions of all agents. Each agent aims to learn a localized policy that contributes to a joint policy across all agents, with the goal of maximizing its own welfare. A commonly adopted objective among the participating agents is to learn some form of equilibrium, such as a Nash Equilibrium. Departing from Markov games, we focus on cooperative MARL, where we try to avoid the curse of heterogeneity in the joint learning.

**Personalization in heterogeneous environments.** Training personalized policies in cooperative multi-agent systems in the presence of environment heterogeneity remains largely underexplored. Yang et al. (2024) studied the training of personalized policies in the context where each agent has a private reward function but shares a common transition kernel for the environment. The analysis focused solely on the convergence of the global policy, defined as the average of the local policies. Jin et al. (2022) proposed a heuristic deep FedRL method, where the policy network can be decomposed into a shared subnetwork and environment embedding layer. XIONG et al. studied personalized TD learning with two timescales and established a linear convergence speedup. However, under their notation, the validity of the error contraction in Eq. (54) appears to rely on the assumptions $D_1 = o(\beta_t)$ and $D_2 = o(\alpha_t)$, which may be challenging to satisfy in practice due to the involvement of the stepsize ratios $\frac{\beta_t}{\alpha_t}$ and $\frac{\beta_t}{\alpha_{t+1}}$.

## 3 PROBLEM SETUP AND PRELIMINARIES

**MARL.** In this work, we consider a multi-agent system that consists of one parameter server and $K$ agents. The parameter server can orchestrate the learning at the agents through iteratively collecting and aggregating the updates at the agents. We consider MARL with heterogeneous environments, where the agents are modelled as Markov Decision Processes (MDP) $\{\mathcal{M}^k \mid \mathcal{M}^k =< P^k, \mathcal{S}, \mathcal{A}, R >$ for $k = 1, 2, ..., K\}$, where $\mathcal{S} = \mathbb{R}$ is the state space, $\mathcal{A}$ is a finite action space, $P^k$'s are the transition kernels, and $R$ is the reward function: $\mathcal{S} \times \mathcal{A} \to [-U_r, U_r]$. Slightly departing from existing work on federated reinforcement learning, we consider undiscounted rewards—specifically, the time-averaged expected reward (Tsitsiklis & Van Roy, 1999).

Let $\pi$ be a given policy. For each agent $k$, we assume that the limiting state distribution of agent $k$ under the transition kernel $P^k$ and the given policy $\pi$ exists and is denoted as $\mu^k$. The time-average expected reward at agent $k$ is defined as:

$$J^k(\pi) := \lim_{T \to \infty} \frac{1}{T} \mathbb{E} \left[ \sum_{t=0}^{T-1} R(s_t, a_t) \right] = \mathbb{E}_{s \sim \mu^k, a \sim \pi} [R(s, a)], \tag{1}$$

where the first expectation is taken over the state-action trajectories generated under the Markov chain, while the second is taken over their stationary distribution. The value function is defined as

$$V^{k,\pi}(s) = \mathbb{E}_{\pi_\theta, P^k} \left[ \sum_{t=0}^{\infty} R(s_t, a_t) - J^k(\pi) \mid s_0 = s \right]. \tag{2}$$

We focus on learning the value function for each agent $k$ under the given $\pi$ based on their collective experience with their environments. For ease of exposition, we drop the index and function argument in $\pi$. That is, for each $k$, $J^k(\pi)$ is written as $J^k$, and $V^{k,\pi}(s)$ as $V^k(s)$ for short.

**Linear function approximation via TD learning.** Let $\mathcal{V}$ be a function space that $V^k$ belongs to. Let $\phi : \mathcal{S} \to L^2(\mathbb{N})$ be a feature map of $\mathcal{V}$. Denote the coefficients of the value $V^k$ with respect to $\phi$ as $z^{k,*}$, where $z^{k,*} \in L^2(\mathbb{N})$. For simplicity, in this paper, we assume $\phi$ is known and has finite dimension $d$, i.e., $\phi : \mathcal{S} \to \mathbb{R}^d$. We consider the following linear function approximation:

$$\widehat{V}(s; z^k) = \phi(s)^\top z^k.$$

To drive the approximation $\widehat{V}(s; z)$ to $V^k$ TD learning (particularly TD(0)) update is applied at an agent to learn its value function:

$$\begin{aligned}
z_{t+1}^k &= z_t^k + \beta_t \left( R(a_t^k, s_t^k) - \eta_t^k + \left( \phi(s_{t+1}^k) - \phi(s_t^k) \right)^\top z_t^k \right), \\
\eta_{t+1}^k &= \eta_t^k + \gamma_t \left( R(a_t^k, s_t^k) - \eta_t^k \right),
\end{aligned} \tag{3}$$

where $\beta_t$ and $\gamma_t$ are the stepsizes. It is well-known that under some standard technical assumptions $\lim_{t \to \infty} z_t^k \to z^{k,*}$ (Tsitsiklis & Van Roy, 1999; Sutton & Barto, 2018).

**Common structure: Low-dimensional subspace.** To capture the potential gains of learning federation compared with learning individually, following Collins et al. (2021); Niu et al. (2024), we assume that $\{z^{k,*} : k = 1, \cdots, K\}$ belongs to and fully span an $r$-dimensional subspace of $\mathbb{R}^d$. That is, there exists an orthonormal matrix $\mathbf{B}^* \in \mathbb{R}^{d \times r}$ such that

$$z^{k,*} = \mathbf{B}^* \omega^{k,*}, \quad \text{where } \omega^{k,*} \in \mathbb{R}^r. \tag{4}$$

To see the generality of this formulation: $r = 1$ when $z^{k,*} = z^{k',*}$ for all $k, k'$ (i.e., homogeneous transitional kernels $P^k$); $r = K$ when $z^{k,*} \perp z^{k',*}$ for all $k' \neq k$ – which is possible when $K \leq d$.

## 4 ALGORITHM: FEDERATED SINGLE-TIMESCALE TD

We propose and analyze a natural collaborative TD algorithm, wherein the local linear estimates $z^k$ are decomposed into the product of the subspace estimate $B \in \mathbb{R}^{d \times r}$, which is common over all agents, and an agent-specific head $w \in \mathbb{R}^r$; formally, $z^k = B\omega^k$ for all $k$. A formal description of the algorithm can be found in Algorithm 1.

The algorithm can be roughly understood as follows. $K$ agents try to estimate their value functions via a joint linear approximation:

$$\min_{B,\omega^1,\cdots,\omega^K} \frac{1}{2} \sum_{k=1}^{K} \mathbb{E}_{\mu^k}[(\widehat{V}^k(s;\omega^k,\mathbf{B}) - V^k(s))^2], \quad \text{where} \quad \widehat{V}^k(s;\omega^k,\mathbf{B}) = \phi(s)^\top \mathbf{B}\omega^k. \quad (5)$$

A natural stochastic gradient descent of Eq. (5) updates of $\mathbf{B}$ and $\omega^k$ for all $k$ are:

$$\omega_{t+1}^k = \omega_t^k + \beta_t \left[V^k(s_t^k) - \phi(s_t^k)^\top \mathbf{B}_t \omega_t^k\right] \mathbf{B}_t^\top \phi(s_t^k), \quad (6)$$

$$\mathbf{B}_{t+1}^k = \mathbf{B}_t^k + \zeta_t \left[V^k(s_t^k) - \phi(s_t^k)^\top \mathbf{B}_t \omega_t^k\right] \phi(s_t^k)(\omega_t^k)^\top, \quad (7)$$

where $\beta_t$ and $\zeta_t$ are stepsizes of $\omega$ and $B$, respectively; the specific choices of stepsizes can be found in Section 5. However, $V^k$ is unknown for each $k$. Following the semi-gradient method TD(0), we use $R(s_t^k, a_t^k) - J^k + \widehat{V}^k(s_{t+1}^k; \omega_t^k, \mathbf{B}_t)$ to estimate $V^k(s_t^k)$, and use $\eta_t^k$ (the same as in Eq. (3)) to estimate $J^k$ in a stochastic approximation manner. For ease of exposition, we define the TD error at each agent as

$$\delta_t^k = r_t^k - \eta_t^k + (\phi(s_{t+1}^k) - \phi(s_t^k))^\top \mathbf{B}_t \omega_t^k. \quad (8)$$

In addition, we use the projection operator $\Pi_{U_\omega}(\cdot)$ to ensure the boundedness (with respect to $\ell_2$ norm) of the local heads $\omega$, where $U_\omega$ is some known constant. In each round, the agents communicate only their local subspace estimates $\mathbf{B}_t^k$ to the parameter server, which then performs averaging followed by a QR decomposition to ensure that $\mathbf{B}_t$ has orthonormal columns (Collins et al., 2021).

---

**Algorithm 1** Federated Single-Timescale TD (FSTTD)

---

1: **Input:** Initial critic parameter $\{\omega_0^k\}_{k=1}^K$, orthonomal $B_0 \in \mathbb{R}^{d \times r}$, $\beta_t$ for local heads, $\zeta_t$ for subspace, and $\gamma_t$ for reward estimator, $T$, and $U_\omega$ the projection upper bound.
2: **for** $k = 1, 2, \ldots, K$ **do**
3:     $\eta_0^k = 0$
4:     Draw $s_0^k$ from some initial distribution independently
5: **end for**
6: **for** $t = 0, 1, 2, \ldots, T-1$ **do**
7:     **for** $k = 1, 2, \ldots, K$ **do**
8:         Take action $a_t^k \sim \pi(\cdot|s_t^k)$.
9:         Observe next state $s_t^k \sim \mathcal{P}^k(\cdot|s_t^k, a_t^k)$ and reward $r_t^k = R(s_t^k, a_t^k)$.
10:        $\delta_t^k = r_t^k - \eta_t^k + (\phi(s_{t+1}^k) - \phi(s_t^k))^\top \mathbf{B}_t \omega_t^k$.
11:        $\eta_{t+1}^k = \eta_t^k + \gamma_t(r_t^k - \eta_t^k)$.
12:        $\omega_{t+1}^k = \Pi_{U_\omega}(\omega_t^k + \beta_t \delta_t^k \mathbf{B}_t^\top \phi(s_t^k))$.
13:     **end for**
14:     **for** $k = 1, 2, \ldots, K$ **do**
15:        $\mathbf{B}_{t+1}^k = \mathbf{B}_t + \zeta_t \delta_t^k \phi(s_t^k)(\omega_t^k)^\top$
16:     **end for**
17:     $\bar{\mathbf{B}}_{t+1} = \frac{1}{K} \sum_{k=1}^K \mathbf{B}_{t+1}^k$,
18:     $\mathbf{B}_{t+1}, \mathbf{R}_{t+1} = \mathtt{QR}(\bar{\mathbf{B}}_{t+1})$
19:     Server sends current $\mathbf{B}_{t+1}$ to agents.
20: **end for**

---

## 5   CONVERGENCE ANALYSIS

### 5.1   ASSUMPTIONS

We present some technical assumptions adopted in our convergence analysis. Assumptions 1, 2, and 3 are widely adopted in the RL literature. Assumption 4 is introduced to control the heterogeneous distortion of the matrix $A^k$ on the principal angle distance evolution. Assumption 5 quantifies how well spread the underlying truth $z^{k,*}$ is in covering the $r$-dimensional subspace of $\mathbf{B}^*$.

For each MDP $\mathcal{M}^k$, it is well-known that given the feature representation $\phi(s)$ and policy $\pi$, there exists a TD fixed point $z^{k,*}$ minimizing the projected Bellman error if the following assumption holds (Sutton & Barto, 2018).

*Assumption* 1 (Exporation). The matrix $A^k = \mathbb{E}_{(s,a,s') \sim \mu^k \otimes \pi \otimes P^k}[\phi(s)(\phi(s') - \phi(s))^\top]$ is negative definite and its largest eigenvalue is upper bounded by $-\lambda$.

This assumption is widely adopted in the literature. Intuitively, $A^k$ captures the exploration of the policy $\pi$ under the transition kernel $P^k$. For the tabular setting, Assumption 1 holds when the policy $\pi$ explores all state-action pairs.

Recall that $\mu^k$ is the stationary distribution induced by policy $\pi$ and the transition probability $P^k, \forall k$. Uniform ergodicity is often assumed in the literature to characterize the noise induced by Markovian sampling (Chen & Zhao, 2024; Olshevsky & Gharesifard, 2023; Zou et al., 2019; Wu et al., 2020).

*Assumption* 2 (Uniform ergodicity). Let $\mathbb{P}^k_{0:\tau}(\cdot|s^k_0 = s)$ denote the state distribution after $\tau$ steps given $s^k_0 = s$ for agent $k$. There exist $m > 0$ and $\rho \in (0, 1)$, such that

$$d_{TV}(\mathbb{P}^k_{0:\tau}(\cdot|s^k_0 = s), \mu^k(\cdot)) \leq m\rho^\tau, \ \forall \tau \geq 0, \forall s \in \mathcal{S}, \ \forall k \in [K], \tag{9}$$

where $d_{TV}$ is the total variation distance.

We further define (Chen & Zhao, 2024)

$$\tau_T := \min\{i \mid m\rho^{i-1} \leq \frac{1}{\sqrt{T}}, \forall i \geq 0.\}, \tag{10}$$

which will be used in our formal results statements. Explicitly,

$$\tau_T = \frac{\log(m\rho^{-1})}{\log(\rho^{-1})} + \frac{\log T}{2\log(\rho^{-1})} = \mathcal{O}(\log(T)).$$

Furthermore, it is assumed that the shared features are bounded. Formally,

*Assumption* 3. $\|\phi(s)\| \leq 1$ for each $s \in \mathcal{S}$.

In PFL, the existence of a shared common subspace helps to reduce the local learning problem from a $d$-dimensional problem to an $r$-dimensional problem. When $r \ll d$, this simplification can be substantial. Under Assumption 3, however, the dependence on $d$ is obscured, which may explain the absence of an explicit speedup in our bound. Relaxing this assumption, however, requires a fundamental rethinking of the problem, which we leave to future work.

Though popular, we conjecture that this assumption may hide the benefits of shared subspace.

The following assumption is imposed to ensure the negative drift of the principal angle distance $\|\mathbf{B}^{*\top}_\perp \mathbf{B}_t\|$, which is used in deriving Lemma 5.4.

*Assumption* 4. The subspace $\text{span}(\mathbf{B}^*)$ is $A^k$-invariant for all $k \in [K]$. That is, if $v \in \text{span}(\mathbf{B}^*)$, then $A^k v \in \text{span}(\mathbf{B}^*)$.

Let

$$\mathbf{Z}^* \in \mathbb{R}^{d \times K}, \quad \text{with } z^{k,*} \text{ as the } k\text{-th column.} \tag{11}$$

By Eq. (4), we know that the rank of $\mathbf{Z}^* \mathbf{Z}^{*\top}$ is $r$. Let $\lambda^+_{\min}$ denote the smallest nonzero eigenvalue of $\mathbf{Z}^* \mathbf{Z}^{*\top}$. We require that each of the $r$ dimensions of $\mathbf{B}^*$ is well-covered by $\{z^{k,*}\}^K_{k=1}$, formally stated in Assumption 5.

*Assumption* 5. $\frac{1}{K}\lambda^+_{\min} \geq \alpha$ for some absolute constant $\alpha > 0$.

## 5.2 Main Convergence Results

In Algorithm 1, three groups of variables are updated: the common subspace estimate $B_t$, the local heads $\omega^k_t$, and estimates of local time-averaged rewards (defined in Eq. (1)). In this subsection, we characterize their evolutions over time. For ease of exposition, we introduce a set of notation:

$$x^k_t = \mathbf{B}_t \omega^k_t - z^{k,*}, \qquad m_t = \mathbf{B}^{*\top}_\perp \mathbf{B}_t, \qquad y^k_t = \eta^k_t - J^k, \tag{12}$$

$$\widetilde{\omega}^k_t = \omega^k_t + \beta_t \delta^k_t \mathbf{B}^\top_t \phi(s^k_t), \quad \widetilde{x}^k_t = \bar{\mathbf{B}}_t \widetilde{\omega}^k_t - z^{k,*}, \quad \bar{m}_t = \mathbf{B}^{*\top}_\perp \bar{\mathbf{B}}_t. \tag{13}$$

In addition, let

$$\xi^k_t = (r^k_t - J^k + (\phi(s^k_{t+1}) - \phi(s^k_t))^\top \mathbf{B}_t \omega^k_t)\phi(s^k_t)$$
$$- \mathbb{E}_{\mu^k, \pi, P^k}[(r^k_t - J^k + (\phi(s^k_{t+1}) - \phi(s^k_t))^\top \mathbf{B}_t \omega^k_t)\phi(s^k_t)], \tag{14}$$

which can be viewed as the Markovian noise of the TD error of agent $k$ at time $t$. The following rewriting appears repeatedly in our analysis.

**Proposition 1.** *For any $t$ and agent $k$, $\delta_t^k \phi(s_t^k)$ can be rewritten as*

$$\delta_t^k \phi(s_t^k) = \xi_t^k + A^k x_t^k + y_t^k \phi(s_t^k).$$

**Lemma 5.1** (Upper bound of local head errors). *Let $\mathbf{P}^* = \mathbf{B}^*(\mathbf{B}^*)^\top$, and $\mathbf{P}_t = \mathbf{B}_t \mathbf{B}_t^\top$ for each $t$. In addition, let $U_\delta = 2U_r + 2U_\omega$. For each agent $k$ and time $t$, it holds that*

$$\mathbb{E}\|\widetilde{x}_{t+1}^k\|^2 \leq (1 - 2\lambda\beta_t - 4\zeta_t U_\omega^2)\mathbb{E}\|x_t^k\|^2 + 2\beta_t \mathbb{E}\langle x_t^k, \mathbf{P}_t \xi_t^k \rangle + 16\beta_t U_\omega \sqrt{\mathbb{E}\|x_t^k\|^2}\sqrt{\mathbb{E}\|m_t\|^2}$$

$$+ 2\beta_t \sqrt{\mathbb{E}\|x_t^k\|^2}\sqrt{\mathbb{E}|y_t^k|^2} + 4\zeta_t U_\omega^2 \frac{1}{K}\sum_{i=1}^K \sqrt{\mathbb{E}\|x_t^k\|^2}\sqrt{\mathbb{E}\|x_t^i\|^2} + \frac{2\zeta_t}{K}\sum_{i=1}^K \mathbb{E}\langle x_t^k, \xi_t^i(\omega_t^i)^\top \omega_t^k \rangle$$

$$+ \frac{2\zeta_t U_\omega^2}{K}\sum_{i=1}^K \sqrt{\mathbb{E}\|x_t^k\|^2}\sqrt{\mathbb{E}|y_t^i|^2} + 4\zeta_t\beta_t U_\omega^2 U_\delta^2 + 3\beta_t^2 U_\delta^2 + 3\zeta_t^2 U_\delta^2 U_\omega^4 + 3\beta_t^2 \zeta_t^2 U_\delta^4 U_\omega^2.$$

Intuitively, when $\mathbb{E}\|m_t\|^2 \lesssim \mathbb{E}\|x_t^k\|^2$. The above upper bound is somewhat similar to the traditional single-agent setting Chen & Zhao (2024). We further show (in Lemma 5.2) that it is impossible for $\mathbb{E}\|m_t\|^2 \gg \mathbb{E}\|x_t^k\|^2$.

Deriving Lemma 5.1 is highly nontrivial and requires a fundamental detour from the existing analysis Chen & Zhao (2024). Furthermore, existing analysis in PFL Collins et al. (2021) is not applicable to our problem due to the Markovian sampling, the TD updates, and the lack of responses. Roughly speaking, the main drift of the decay in $\|x_t^k\|$ may arise from $2\beta_t \langle x_t^k, \mathbf{B}_t \mathbf{B}_t^\top \phi(s_t^k)\delta_t^k \rangle$. However, the environmental heterogeneity significantly complicates the characterization of this term. Specifically, by Proposition 1, it can be written as:

$$2\beta_t \langle x_t^k, \mathbf{B}_t \mathbf{B}_t^\top \phi(s_t^k)\delta_t^k \rangle$$
$$= 2\beta_t \langle x_t^k, \mathbf{P}_t A^k x_t^k \rangle + 2\beta_t \langle x_t^k, \mathbf{P}_t \xi_t^k \rangle + 2\beta_t \langle x_t^k, \mathbf{P}_t y_t^k \phi(s_t^k) \rangle.$$

Zooming into this expression, the main negative drift arises from the term $\langle x_t^k, \mathbf{P}_t A^k x_t^k \rangle$. Intuitively, in traditional single-agent or homogeneous environment settings, $\langle x_t^k, \mathbf{P}_t A^k x_t^k \rangle \approx \langle x_t^k, A x_t^k \rangle$, which is mainly controlled by the spectrum of $A$, with the desired property directly assumed in Assumption 1. However, in the presence of environmental heterogeneity, Assumption 1 does not directly guarantee a negative drift of the $x_t^k$ due to the existence of $\mathbf{P}_t$ and its intricate interplay with the heterogeneous $A^k$. Specifically, (1) $\mathbf{P}_t$ is not of full rank, (2) the local head error will be distorted in a different manner due to the product $\mathbf{P}_t A^k$, and (3) $\mathbf{P}_t$ varies over time.

**Lemma 5.2** (Lower bound of local head errors). *Suppose that $d \geq 2r$. It holds that*

$$\frac{1}{K}\sum_{k=1}^K \mathbb{E}\|x_t^k\|^2 \geq \frac{\mathbb{E}\|m_t\|^2}{K}\lambda_{\min}^+ \geq \frac{\mathbb{E}\|m_t\|_F^2}{rK}\lambda_{\min}^+.$$

Intuitively, Lemma 5.2 says that when the principal angle distance between $\mathbf{B}_t$ and $\mathbf{B}^*$ is large, the well coverage of all the $r$ directions by the underlying truth $\{z^{1,*}, \cdots, z^{K,*}\}$ ensures that the aggregate errors in the local head estimates remain bounded away from zero. In other words, the local heads cannot be learned significantly faster than the subspace itself.

For the convergence result, we choose our target to be the time average expected error bound. Therefore, we define

$$Y_T^k := \frac{1}{T - \tau_T}\sum_{t=\tau_T}^{T-1} \mathbb{E}|y_t^k|^2, \quad X_T^k := \frac{1}{T - \tau_T}\sum_{t=\tau_T}^{T} \mathbb{E}\|x_t^k\|^2,$$

$$\bar{X}_T := \frac{1}{K}\sum_{k=1}^K X_T^k, \quad\quad M_T := \frac{1}{T - \tau_T}\sum_{t=\tau_T}^{T} \mathbb{E}\|m_t\|^2.$$

**Lemma 5.3** (Reward analysis). *Choose the stepsize $\gamma_t = \frac{c_\gamma}{\sqrt{T}}$ for $t \leq T$, where $c_\gamma$ is some absolute constant. It holds that*

$$\frac{1}{T - \tau_T}\sum_{t=\tau_T}^{T-1} \mathbb{E}|y_t^k|^2 \leq \frac{\sqrt{T}}{T - \tau_T}\frac{2U_r^2}{c_\gamma} + \frac{c_\gamma \tau_T 4U_r^2 + 4U_r^2}{\sqrt{T}} + \frac{2c_\gamma U_r^2}{\sqrt{T}},$$

*where $\tau_T$ is defined in Eq. (10).*

Lemma 5.3 says that the time-averaged reward estimation error decays at the rate of $\widetilde{\mathcal{O}}(\frac{1}{\sqrt{T}})$. The average starts from time index $\tau_T$ to address the Markovian sampling noise.

**Lemma 5.4** (Principal angle distance analysis). *Choose stepsize $\zeta_t = \frac{c_\zeta}{\sqrt{T}}$, where $c_\zeta$ is some absolute constans. It holds that,*

$$M_T \leq \frac{\sqrt{T}}{T - \tau_T} \frac{r^2}{\lambda \alpha c_\zeta} + \frac{2rc_\zeta U_\delta^2 U_\omega^2}{\sqrt{T}\lambda K\alpha} + \frac{4rC_1}{\lambda \alpha \sqrt{T}} + \frac{4rU_\omega}{\lambda K\alpha} \sum_{k=1}^{K} \sqrt{M_T Y_T^k},$$

*where $C_1 = \left[(8U_\omega^2 + 4U_\omega U_r) 10c_\zeta \tau_T U_\delta U_\omega + (8U_\omega + 4U_r)c_\beta \tau_T U_\delta U_{\mathbf{B}} + (4U_r + 4U_\omega)U_\omega\right]$, and $\tau_T$ is defined in Eq. (10).*

**Lemma 5.5** (Local weight analysis). *Choose stepsize $\beta_t = \frac{c_\beta}{\sqrt{T}}$ and $\zeta_t = \frac{c_\zeta}{\sqrt{T}}$, where $c_\zeta, c_\beta$ are some absolute constans. It holds that,*

$$X_T^k \leq \mathcal{O}(\frac{\log(T)}{\sqrt{T}}) + 16U_\omega \sqrt{X_T^k M_T} + \frac{1}{\lambda}\sqrt{X_T^k Y_T^k}$$

$$+ \frac{2c_\zeta U_\omega^2}{c_\beta \lambda} \frac{1}{K} \sum_{i=1}^{K} \sqrt{X_T^k X_T^i} + \frac{c_\zeta U_\omega^2}{c_\beta \lambda K} \sum_{i=1}^{K} \sqrt{X_T^k Y_T^i} + \mathcal{O}(1).$$

**Theorem 1** (Convergence). *Consider Algorithm 1 with $\beta_t = \frac{c_\beta}{\sqrt{T}}, \zeta_t = \frac{c_\zeta}{\sqrt{T}}, \gamma_t = \frac{c_\gamma}{\sqrt{T}}$, where $c_\beta, c_\zeta, c_\gamma$ are absolute constants depending on problem parameters. We have for $T \geq 2\tau_T$, $r \leq \log(T)$,*

$$Y_T^k = \mathcal{O}(\frac{\log(T)}{\sqrt{T}}), \bar{X}_T = \mathcal{O}(1), M_T = \mathcal{O}(\frac{r \log(T)}{\sqrt{T}}).$$

For $\bar{X}_T$, there is an unavoidable $\mathcal{O}(1)$ error which arises from the gap between the raw updates and the updates after QR decomposition and projection (see equation 25). When this gap is neglible, then $\bar{X}_T = \mathcal{O}(\frac{r \log(T)}{\sqrt{T}})$ as can be seen from equation 46.

### 5.3 PROOF SKETCH

The challenge of single-timescale convergence analysis for this problem is that the estimation errors of the time-average reward, the critic, and the subspace misalignment are strongly coupled, whereas in double-loop or two-timescale analysis, these error terms are usually naturally decoupled.

However, as shown in Lemma 5.3, Lemma 5.4, and Lemma 5.5, the reward error is independent of the other two error terms, the subspace misalignment error is only coupled with the reward error, and the critic error is coupled with the other two errors. Therefore, we can sequentially solve the coupled system. We now provide an overall proof sketch for each error terms.

**Reward estimation error analysis.**

By Eq. (12) and the update of $\eta^k$ in Algorithm 1, we can decompose the reward error into:

$$(y_{t+1}^k)^2 = (1 - 2\gamma_t)(y_t^k)^2 + 2\gamma_t y_t^k (r_t^k - J^k) + (\gamma_t(r_t^k - \eta_t^k))^2.$$

The first term is a contraction term, the second term is a Markovian bias term, which decays to 0 in expectation, and the last term is a variance term of order $\mathcal{O}(\gamma_t^2)$.

**Principal angle distance analysis.**

Using the update rule in Algorithm 1, we can decompose the squared error into

$$\|m_{t+1}\|_F^2 \leq \|\bar{m}_{t+1}\|_F^2 \|\mathbf{R}_{t+1}^{-1}\|^2,$$

$$\|\bar{m}_{t+1}\|_F^2 \leq \|m_t\|_F^2 + \|\frac{\zeta_t}{K} \sum_{k=1}^{K} \mathbf{B}_\perp^{*\top} \delta_t^k \phi(s_t^k)(\omega_t^k)^\top\|_F^2 + 2\langle m_t, \frac{\zeta_t}{K} \sum_{k=1}^{K} \mathbf{B}_\perp^{*\top} \xi_t^k(\omega_t^k)^\top\rangle$$

$$+ 2\langle m_t, \frac{\zeta_t}{K} \sum_{k=1}^{K} \mathbf{B}_\perp^{*\top} y_t^k \phi(s_t^k)(\omega_t^k)^\top\rangle + 2\langle m_t, \frac{\zeta_t}{K} \sum_{k=1}^{K} \mathbf{B}_\perp^{*\top} A^k x_t^k(\omega_t^k)^\top\rangle.$$

For the first inequality, we will show in Lemma B.1 that $\|\mathbf{R}_{t+1}^{-1}\| \leq \frac{1}{1-\mathcal{O}(\zeta_t)}$ and can be absorbed into the contraction term we get from the last inner product of the second inequality. Turning to the second inequality, the second-order terms will decay very fast, and we focus on the three inner products. The first inner product is a Markovian noise term which decays in expectation, and the second inner product is coupled by the reward error $y_t^k$ and $m_t$, which decays eventually as long as $y_t^k$ decays and there are no other non-decaying terms. For the last inner product, we invoke Lemma 5.2 and it can be bounded by $-\frac{2\lambda\lambda_{\min}^{+}\zeta_t}{r}\|m_t\|_F^2$, giving us a contraction.

**Critic error analysis.** Using the update rule in Algorithm 1, we can decompose the squared error into

$$
\begin{aligned}
\|\widetilde{x}_{t+1}^k\|^2 \leq &\|x_t^k\|^2 + 2\beta_t\langle x_t^k, \mathbf{B}_t\mathbf{B}_t^\top \phi(s_t^k)\delta_t^k\rangle \\
&+ 2\zeta_t\langle x_t^k, \frac{1}{K}\sum_{i=1}^K \phi(s_t^i)\delta_t^i(\omega_t^i)^\top\omega_t^k\rangle \\
&+ 2\zeta_t\beta_t\left\langle x_t^k, (\frac{1}{K}\sum_{i=1}^K \delta_t^i\phi(s_t^i)(\omega_t^i)^\top)\left(\delta_t^k\boldsymbol{B}_t^\top\phi(s_t^k)\right)\right\rangle \\
&+ 3\beta_t^2\|\mathbf{B}_t(\delta_t^k\boldsymbol{B}_t^\top\phi(s_t^k))\|^2 + 3\zeta_t^2\|\frac{1}{K}\sum_{i=1}^K \delta_t^i\phi(s_t^i)(\omega_t^i)^\top\omega_t^k\|^2 \\
&+ 3\beta_t^2\zeta_t^2\|(\frac{1}{K}\sum_{i=1}^K \delta_t^i\phi(s_t^i)(\omega_t^i)^\top)\left(\delta_t^k\boldsymbol{B}_t^\top\phi(s_t^k)\right)\|^2.
\end{aligned}
$$

The key operation here is to push out a negative drift of the main error term, i.e., a contraction, and to show the remaining term decays. For ease of comprehension, we omit the specific terms and only talk about what terms can be pushed out. The detailed explanation can be found in Section C.1.

The first inner product can give us a negative drift of $-2\lambda\beta_t\|x_t^k\|^2$, and the remaining terms are Markovian noise and errors coupled by $\|m_t\|$ and $\|x_t^k\|$. The second inner product can be decomposed into Markovian noise, a term coupled by $\|x_t^k\|$ and $\|x_t^i\|, i \in [K]$, and a term coupled by $\|y_t^k\|$ and $\|y_t^k\|$. The second-order terms decay even faster.

## 6 CONCLUSION AND DISCUSSIONS

In this work, we studied cooperative TD learning in heterogeneous environments, where agents share a low-dimensional common structure but face diverse local dynamics. By combining subspace estimation with personalized value function updates, we established finite-time convergence guarantees under Markovian sampling and clarified the limitations imposed by standard assumptions. We hope that our analysis framework and insights provide a foundation for future work on exploiting common structures in multi-agent reinforcement learning while balancing the benefits and drawbacks of collaboration.

**Limitations.** While our analysis demonstrates resilience to heterogeneity, the absence of explicit acceleration highlights the need for understanding the benefits of cooperative learning of shared structure. Specifically, in PFL, the existence of a shared common subspace helps to reduce the local learning problem from a $d$-dimensional problem to an $r$-dimensional problem. When $r \ll d$, this simplification can be substantial. Under Assumption 3, however, the dependence on $d$ is obscured, which may explain the absence of an explicit speedup in our bound. Relaxing this assumption, however, requires a fundamental rethinking of the problem, which we leave to future work. Furthermore, we need a deeper understanding of error due to QR decomposition and projection when analyzing the convergence of local critics.

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

Appendices

# A    PROOF OF PROPOSITION 1

$$
\begin{aligned}
\delta_t^k \phi(s_t^k) =& (r_t^k - \eta_t^k + (\phi(s_{t+1}^k) - \phi(s_t^k))^\top \mathbf{B}_t \omega_t^k) \phi(s_t^k) \\
=& (r_t^k - J^k + (\phi(s_{t+1}^k) - \phi(s_t^k))^\top \mathbf{B}_t \omega_t^k) \phi(s_t^k) + (J^k - \eta_t^k) \phi(s_t^k) \\
=& \mathbb{E}_{\mu^k, \pi, P^k}[(r_t^k - J^k + (\phi(s_{t+1}^k) - \phi(s_t^k))^\top \mathbf{B}_t \omega_t^k) \phi(s_t^k)] \\
& + (r_t^k - J^k + (\phi(s_{t+1}^k) - \phi(s_t^k))^\top \mathbf{B}_t \omega_t^k) \phi(s_t^k) \\
& - \mathbb{E}_{\mu^k, \pi, P^k}[(r_t^k - J^k + (\phi(s_{t+1}^k) - \phi(s_t^k))^\top \mathbf{B}_t \omega_t^k) \phi(s_t^k)] \\
& + (J^k - \eta_t^k) \phi(s_t^k) \\
=& \mathbb{E}_{\mu^k, \pi, P^k}[(r_t^k - J^k + (\phi(s_{t+1}^k) - \phi(s_t^k))^\top \mathbf{B}_t \omega_t^k) \phi(s_t^k)] \\
& + \xi_t^k + y_t^k \phi(s_t^k).
\end{aligned}
$$

Recall that, by definition, that $z^{k,*} = \mathbf{B}^* \omega^{k,*}$ is the TD limiting point of agent $k$. Hence, $\mathbb{E}_{\mu^k, \pi, P^k}[(r_t^k - J^k + (\phi(s_{t+1}^k) - \phi(s_t^k))^\top \mathbf{B}^* \omega^{k,*}) \phi(s_t^k)] = 0$. So, we have

$$
\begin{aligned}
\delta_t^k \phi(s_t^k) =& \mathbb{E}_{\mu^k, \pi, P^k}[(r_t^k - J^k + (\phi(s_{t+1}^k) - \phi(s_t^k))^\top \mathbf{B}_t \omega_t^k) \phi(s_t^k)] \\
& - \mathbb{E}_{\mu^k, \pi, P^k}[(r_t^k - J^k + (\phi(s_{t+1}^k) - \phi(s_t^k))^\top \mathbf{B}^* \omega^{k,*}) \phi(s_t^k)] \\
& + \xi_t^k + y_t^k \phi(s_t^k) \\
=& \mathbb{E}_{\mu^k, \pi, P^k}[(\phi(s_{t+1}^k) - \phi(s_t^k))^\top \left( \mathbf{B}_t \omega_t^k - \mathbf{B}^* \omega^{k,*} \right) \phi(s_t^k)] + \xi_t^k + y_t^k \phi(s_t^k) \\
\overset{(a)}{=}& \mathbb{E}_{\mu^k, \pi, P^k}[\phi(s_t^k)(\phi(s_{t+1}^k) - \phi(s_t^k))^\top] \left( \mathbf{B}_t \omega_t^k - \mathbf{B}^* \omega^{k,*} \right) + \xi_t^k + y_t^k \phi(s_t^k) \\
=& A^k (\mathbf{B}_t \omega_t^k - \mathbf{B}^* \omega^{k,*}) + \xi_t^k + y_t^k \phi(s_t^k),
\end{aligned}
$$

where equality (a) holds because of the fact that $(\phi(s_{t+1}^k) - \phi(s_t^k))^\top \left( \mathbf{B}_t \omega_t^k - \mathbf{B}^* \omega^{k,*} \right)$ is a scalar.

# B    BOUND ON QR DECOMPOSITION

Recall from Algorithm 1 the update of $\bar{\mathbf{B}}$:

$$
\bar{\mathbf{B}}_{t+1} = \mathbf{B}_t + \frac{\zeta_t}{K} \sum_{k=1}^K \delta_t^k \phi(s_t^k)(\omega_t^k)^\top
$$

For ease of exposition, let

$$
\mathbf{Q}_t := \frac{\zeta_t}{K} \sum_{k=1}^K \delta_t^k \phi(s_t^k)(\omega_t^k)^\top. \tag{15}
$$

The following lemma enables us to bound the "distortion" caused by the QR decomposition in terms of $\|\mathbf{Q_t}\|$. It is worth noting that, in the traditional PFL Collins et al. (2021), such distortion can be well controlled by the number of $i.i.d.$ local samples that are freshly drawn in each iteration. However, their analysis does not apply to our setting due to the Markov sampling. We fundamentally depart from the analysis in Collins et al. (2021) in deriving upper bounds on those distortions via constructing a fixed-point iteration.

**Lemma B.1** (Perturbation of QR). *For each $t \geq 1$, when $U_\delta U_\omega \zeta_t \leq 1/2$, it holds that*

$$
\|\mathbf{R}_{t+1} - \mathbf{I}\|_2 \leq 4\|\mathbf{Q}_t\|, \tag{16}
$$

$$
\|\mathbf{R}_{t+1}^{-1}\| \leq \frac{1}{1 - \|\mathbf{R}_{t+1} - \mathbf{I}\|_2} \leq \frac{1}{1 - 4\|\mathbf{Q}_t\|}, \tag{17}
$$

$$
\|\mathbf{R}_{t+1}^{-1} - \mathbf{I}\|_2 \leq 8\|\mathbf{Q}_t\|. \tag{18}
$$

*Proof.* Recall that $\bar{\mathbf{B}}_{t+1} = \mathbf{B}_{t+1}\mathbf{R}_{t+1}$. We rewrite $\mathbf{R}_{t+1}^\top\mathbf{R}_{t+1}$ as

$$\mathbf{R}_{t+1}^\top\mathbf{R}_{t+1} = \bar{\mathbf{B}}_{t+1}^\top\bar{\mathbf{B}}_{t+1} = \mathbf{B}_t^\top\mathbf{B}_t + \mathbf{B}_t^\top\mathbf{Q}_t + \mathbf{Q}_t^\top\mathbf{B}_t + \mathbf{Q}_t^\top\mathbf{Q}_t$$

$$= \mathbf{I} + \mathbf{B}_t^\top\mathbf{Q}_t + \mathbf{Q}_t^\top\mathbf{B}_t + \mathbf{Q}_t^\top\mathbf{Q}_t.$$

It can also be rewritten as

$$\mathbf{R}_{t+1}^\top\mathbf{R}_{t+1} = (\mathbf{I} + \mathbf{R}_{t+1} - \mathbf{I})^\top(\mathbf{I} + \mathbf{R}_{t+1} - \mathbf{I})$$

$$= \mathbf{I} + (\mathbf{R}_{t+1} - \mathbf{I})^\top + (\mathbf{R}_{t+1} - \mathbf{I}) + (\mathbf{R}_{t+1} - \mathbf{I})^\top(\mathbf{R}_{t+1} - \mathbf{I}).$$

Thus, we get

$$(\mathbf{R}_{t+1} - \mathbf{I})^\top + (\mathbf{R}_{t+1} - \mathbf{I}) = \mathbf{B}_t^\top\mathbf{Q}_t + \mathbf{Q}_t^\top\mathbf{B}_t + \mathbf{Q}_t^\top\mathbf{Q}_t - (\mathbf{R}_{t+1} - \mathbf{I})^\top(\mathbf{R}_{t+1} - \mathbf{I}). \qquad (19)$$

We construct a fixed-point iteration to "solve" $\mathbf{R}_{t+1} - \mathbf{I}$ in terms of $\mathbf{B}_t^\top\mathbf{Q}_t + \mathbf{Q}_t^\top\mathbf{B}_t + \mathbf{Q}_t^\top\mathbf{Q}_t$ under Eq. (19). Toward this, we decompose the symmetric matrix $\mathbf{B}_t^\top\mathbf{Q}_t + \mathbf{Q}_t^\top\mathbf{B}_t + \mathbf{Q}_t^\top\mathbf{Q}_t$ into

$$\mathbf{B}_t^\top\mathbf{Q}_t + \mathbf{Q}_t^\top\mathbf{B}_t + \mathbf{Q}_t^\top\mathbf{Q}_t = \mathbf{R}_0 + \mathbf{R}_0^\top,$$

where $\mathbf{R}_0$ is upper triangular. Let $\Delta_{t+1} \triangleq \mathbf{R}_{t+1} - \mathbf{I} - \mathbf{R}_0$. Then Eq. (19) is equivalent to

$$\Delta_{t+1}^\top + \Delta_{t+1} = -(\Delta_{t+1}^\top\Delta_{t+1} + \Delta_{t+1}^\top\mathbf{R}_0 + \mathbf{R}_0^\top\Delta_{t+1} + \mathbf{R}_0^\top\mathbf{R}_0). \qquad (20)$$

Next, we define a sequence of $\Delta_{t+1,i}$ for $i \geq 0$. Let $\Delta_{t+1,0} = 0$ for $i \geq 1$ and let $\Delta_{t+1,i}$ be an upper triangular satisfying

$$\Delta_{t+1,i}^\top + \Delta_{t+1,i} = -(\Delta_{t+1,i-1}^\top\Delta_{t+1,i-1} + \Delta_{t+1,i-1}^\top\mathbf{R}_0 + \mathbf{R}_0^\top\Delta_{t+1,i-1} + \mathbf{R}_0^\top\mathbf{R}_0).$$

We first show the boundedness of the sequence. Note that

$$\|\Delta_{t+1,i}\|_2 \leq \|\Delta_{t+1,i}^\top + \Delta_{t+1,i}\|_2 \leq \|\Delta_{t+1,i-1}\|_2^2 + 2\|\mathbf{R}_0\|_2\|\Delta_{t+1,i-1}\|_2 + \|\mathbf{R}_0\|_2^2, \qquad (21)$$

where the second inequality applies the triangle inequality, and the first inequality holds because

$$\|\Delta_{t+1,i}\|_2 \leq \left\|\frac{\Delta_{t+1,i} + \Delta_{t+1,i}^\top}{2}\right\|_2 + \left\|\frac{\Delta_{t+1,i} - \Delta_{t+1,i}^\top}{2}\right\|_2$$

$$= \max_{\|x\|_2=1}\left|x^H\frac{\Delta_{t+1,i} + \Delta_{t+1,i}^\top}{2}x\right| + \max_{\|x\|_2=1}\left|x^H\frac{\Delta_{t+1,i} - \Delta_{t+1,i}^\top}{2}x\right|$$

$$\leq 2\max_{\|x\|_2=1}|x^H\Delta_{t+1,i}x|$$

$$= \max_{\|x\|_2=1}\left|x^H(\Delta_{t+1,i} + \Delta_{t+1,i}^\top)x\right|$$

$$\leq \|\Delta_{t+1,i} + \Delta_{t+1,i}^\top\|_2. \qquad (22)$$

Since $\|\mathbf{R}_0\|_2 \leq 1/4$, for any $\|\Delta_{t+1,i-1}\|_2 \leq 1/4$, we have $\|\Delta_{t+1,i}\|_2 \leq 1/4$ by Eq. (21). By induction, the entire sequence satisfies $\|\Delta_{t+1,i}\|_2 \leq 1/4$ for all $i \geq 0$.

Next, we show the convergence of the sequence $\Delta_{t+1,i}$. By definition,

$$(\Delta_{t+1,i+1}^\top + \Delta_{t+1,i+1}) - (\Delta_{t+1,i}^\top + \Delta_{t+1,i})$$

$$= -(\Delta_{t+1,i}^\top(\Delta_{t+1,i} - \Delta_{t+1,i-1}) + (\Delta_{t+1,i} - \Delta_{t+1,i-1})^\top\Delta_{t+1,i-1}$$

$$+ (\Delta_{t+1,i} - \Delta_{t+1,i-1})^\top\mathbf{R}_0 + \mathbf{R}_0^\top(\Delta_{t+1,i} - \Delta_{t+1,i-1})).$$

By taking the norm on both sides, we obtain

$$\|\Delta_{t+1,i+1} - \Delta_{t+1,i}\|_2 \leq \|(\Delta_{t+1,i+1} - \Delta_{t+1,i})^\top + (\Delta_{t+1,i+1} - \Delta_{t+1,i})\|_2$$

$$\leq \frac{1}{4}\|\Delta_{t+1,i} - \Delta_{t+1,i-1}\|_2 + \frac{1}{4}\|\Delta_{t+1,i} - \Delta_{t+1,i-1}\|_2$$

$$+ 2\|\mathbf{R}_0\|_2\|\Delta_{t+1,i} - \Delta_{t+1,i-1}\|_2.$$

Therefore, since $\|\mathbf{R}_0\|_2 < \frac{1}{4}$, the difference $\|\Delta_{t+1,i+1} - \Delta_{t+1,i}\|_2$ geometrically converges to zero, which implies that the sequence $\Delta_{t+1,i}$ converges.

The limit of the sequence satisfies the fixed-point Eq. (20). Taking norm on both sides of equation 20 yields

$$\|\Delta_{t+1}\|_2 \leq \|\Delta_{t+1}^\top + \Delta_{t+1}\|_2 \leq \|\Delta_{t+1}\|_2^2 + 2\|\mathbf{R}_0\|_2\|\Delta_{t+1}\|_2 + \|\mathbf{R}_0\|_2^2.$$

Since $\|\Delta_{t+1}\|_2 \leq 1/4$, we obtain that

$$\|\Delta_{t+1}\|_2 \leq \frac{1 - 2\|\mathbf{R}_0\|_2 - \sqrt{1 - 4\|\mathbf{R}_0\|_2}}{2} \leq 4\|\mathbf{R}_0\|_2^2.$$

We conclude that

$$\|\mathbf{R}_{t+1} - \mathbf{I}\|_2 = \|\mathbf{R}_0 + \Delta_{t+1}\|_2 \leq \|\mathbf{R}_0\|_2 + 4\|\mathbf{R}_0\|_2^2 \leq 2\|\mathbf{R}_0\|_2.$$

Since $\mathbf{B}_t^\top \mathbf{Q}_t + \mathbf{Q}_t^\top \mathbf{B}_t + \mathbf{Q}_t^\top \mathbf{Q}_t = \mathbf{R}_0 + \mathbf{R}_0^\top$, we have when $U_\delta U_\omega \zeta_t \leq 1/2$,

$$\|\mathbf{R}_0\|_2 \leq \|\mathbf{Q}_t\| + \frac{1}{2}\|\mathbf{Q}_t\|^2 \leq \|\mathbf{Q}_t\| + \|\mathbf{Q}_t\|\frac{U_\delta U_\omega \zeta_t}{2} \leq 2\|\mathbf{Q}_t\|.$$

Moreover,

$$\sigma_{\min}(\mathbf{R}_{t+1}) \geq 1 - \|\mathbf{R}_{t+1} - \mathbf{I}\|_2, \quad \text{by Weyl's inequality,}$$
$$\sigma_{\max}(\mathbf{R}_{t+1}) \leq 1 + \|\mathbf{R}_{t+1} - \mathbf{I}\|_2, \quad \text{by triangle inequality,}$$

and

$$\|\mathbf{R}_{t+1}^{-1} - \mathbf{I}\|_2 = \|\mathbf{R}_{t+1}^{-1}(\mathbf{I} - \mathbf{R}_{t+1})\|_2 \leq \|\mathbf{R}_{t+1}^{-1}\|_2\|\mathbf{I} - \mathbf{R}_{t+1}\|_2.$$

Therefore,

$$\|\mathbf{R}_{t+1} - \mathbf{I}\|_2 \leq 4\|\mathbf{Q}_t\|,$$
$$\|\mathbf{R}_{t+1}^{-1}\| \leq \frac{1}{1 - \|\mathbf{R}_{t+1} - \mathbf{I}\|_2} \leq \frac{1}{1 - 4\|\mathbf{Q_t}\|_2},$$
$$\|\mathbf{R}_{t+1}^{-1} - \mathbf{I}\|_2 \leq 8\|\mathbf{Q}_t\|.$$

For the last inequality, we used $4U_\delta U_\omega \zeta_t \leq 1/2$ so that $\|\mathbf{R}_{t+1}^{-1}\| \leq 2$.

$\square$

## C    Bounds on Local Head Errors

### C.1    Upper bound: Proof of Lemma 5.1

Recall that $\omega_{t+1}^k = \Pi_{U_\omega}(\omega_t^k + \beta_t \delta_t^k \mathbf{B}_t^\top \phi(s_t^k))$, $\mathbf{B}_{t+1} = \bar{\mathbf{B}}_{t+1}\mathbf{R}_{t+1}^{-1}$, and $\widetilde{\omega}_{t+1}^k = \omega_t^k + \beta_t \delta_t^k \mathbf{B}_t^\top \phi(s_t^k)$.
We have

$$
\begin{aligned}
\|x_{t+1}^k\| =& \|\mathbf{B}_{t+1}\omega_{t+1}^k - z^{k,*}\| \\
=& \|\mathbf{B}_{t+1}\Pi_{U_\omega}(\widetilde{\omega}_{t+1}^k) - z^{k,*}\| \\
=& \|\mathbf{B}_{t+1}\mathbf{R}_{t+1}\widetilde{\omega}_{t+1}^k - z^{k,*} + \mathbf{B}_{t+1}\Pi_{U_\omega}(\widetilde{\omega}_{t+1}^k) - \mathbf{B}_{t+1}\mathbf{R}_{t+1}\widetilde{\omega}_{t+1}^k\| \\
=& \|\bar{\mathbf{B}}_{t+1}\widetilde{\omega}_{t+1}^k - z^{k,*} + \mathbf{B}_{t+1}\Pi_{U_\omega}(\widetilde{\omega}_{t+1}^k) - \mathbf{B}_{t+1}\mathbf{R}_{t+1}\widetilde{\omega}_{t+1}^k\| \\
=& \|\widetilde{x}_{t+1}^k + \mathbf{B}_{t+1}\Pi_{U_\omega}(\widetilde{\omega}_{t+1}^k) - \mathbf{B}_{t+1}\mathbf{R}_{t+1}\widetilde{\omega}_{t+1}^k\|.
\end{aligned}
$$

Taking the square on both sides, we get

$$
\begin{aligned}
\|x_{t+1}^k\|^2 =& \|\widetilde{x}_{t+1}^k\|^2 + \|\mathbf{B}_{t+1}\Pi_{U_\omega}(\widetilde{\omega}_{t+1}^k) - \mathbf{B}_{t+1}\mathbf{R}_{t+1}\widetilde{\omega}_{t+1}^k\|^2 \\
&+ 2\langle \mathbf{B}_{t+1}\Pi_{U_\omega}(\widetilde{\omega}_{t+1}^k) - \mathbf{B}_{t+1}\mathbf{R}_{t+1}\widetilde{\omega}_{t+1}^k, \widetilde{x}_{t+1}^k \rangle.
\end{aligned}
\tag{23}
$$

We bound $\|\mathbf{B}_{t+1}\Pi_{U_\omega}(\widetilde{\omega}_{t+1}^k) - \mathbf{B}_{t+1}\mathbf{R}_{t+1}\widetilde{\omega}_{t+1}^k\|^2$ as follows:
We have

$$
\begin{aligned}
&\|\mathbf{B}_{t+1}\Pi_{U_\omega}(\widetilde{\omega}_{t+1}^k) - \mathbf{B}_{t+1}\mathbf{R}_{t+1}\widetilde{\omega}_{t+1}^k\| \\
=& \|\Pi_{U_\omega}(\widetilde{\omega}_{t+1}^k) - \mathbf{R}_{t+1}\widetilde{\omega}_{t+1}^k\| \\
=& \|\Pi_{U_\omega}(\widetilde{\omega}_{t+1}^k) - \widetilde{\omega}_{t+1}^k + \widetilde{\omega}_{t+1}^k - \mathbf{R}_{t+1}\widetilde{\omega}_{t+1}^k\| \\
\leq& \|\Pi_{U_\omega}(\widetilde{\omega}_{t+1}^k) - \widetilde{\omega}_{t+1}^k\| + \|(\mathbf{I} - \mathbf{R}_{t+1})\widetilde{\omega}_{t+1}^k\| \\
\leq& \|\beta_t \delta_t^k \mathbf{B}_t^\top \phi(s_t^k)\| + \|(\mathbf{I} - \mathbf{R}_{t+1})\widetilde{\omega}_{t+1}^k\| \\
\leq& \beta_t U_\delta + 8U_\omega^2 U_\delta \zeta_t,
\end{aligned}
$$

where the last inequality follows from Eq. (16) and we use $U_\delta \beta_t \leq U_\omega$ for large $T$ such that $\|\widetilde{\omega}_{t+1}^k\| \leq 2U_\omega$. Then, the second term in Eq. (23) can be bounded as

$$
\|\mathbf{B}_{t+1}\Pi_{U_\omega}(\widetilde{\omega}_{t+1}^k) - \mathbf{B}_{t+1}\mathbf{R}_{t+1}\widetilde{\omega}_{t+1}^k\|^2 \leq (U_\delta \beta_t + 8U_\omega^2 U_\delta \zeta_t)^2 \leq 2U_\delta^2 \beta_t^2 + 128U_\omega^4 U_\delta^2 \zeta_t^2. \tag{24}
$$

To bound the third term, requiring $U_\delta U_\omega \zeta_t \leq 1$, we have

$$
\|\widetilde{x}_{t+1}^k\| \leq \|\mathbf{B}_t + \frac{\zeta_t}{K}\sum_{k=1}^K \delta_t^k \phi(s_t^k)(\omega_t^k)^\top\|\|\widetilde{\omega}_{t+1}^k\| \leq 4U_\omega.
$$

Then, by Cauchy-Schwarz, we have

$$
\begin{aligned}
&2\langle \mathbf{B}_{t+1}\Pi_{U_\omega}(\widetilde{\omega}_{t+1}^k) - \mathbf{B}_{t+1}\mathbf{R}_{t+1}\widetilde{\omega}_{t+1}^k, \widetilde{x}_{t+1}^k \rangle \\
\leq& 2\|\mathbf{B}_{t+1}\Pi_{U_\omega}(\widetilde{\omega}_{t+1}^k) - \mathbf{B}_{t+1}\mathbf{R}_{t+1}\widetilde{\omega}_{t+1}^k\|\|\widetilde{x}_{t+1}^k\| \\
\leq& (2\beta_t U_\delta + 6U_\omega^2 U_\delta \zeta_t)4U_\omega.
\end{aligned}
\tag{25}
$$

Then, equation 23 can be written as

$$
\|x_{t+1}^k\|^2 \leq \|\widetilde{x}_{t+1}^k\|^2 + 2U_\delta^2 \beta_t^2 + 18U_\omega^4 U_\delta^2 \zeta_t^2 + (2\beta_t U_\delta + 6U_\omega^2 U_\delta \zeta_t)4U_\omega. \tag{26}
$$

We bound $\|\widetilde{x}_{t+1}^k\|$ as follows:

$$
\begin{aligned}
\|\widetilde{x}_{t+1}^k\| =& \|\bar{\mathbf{B}}_{t+1}\widetilde{\omega}_{t+1}^k - z^{k,*}\| \\
=& \|(\mathbf{B}_t + \frac{\zeta_t}{K}\sum_{i=1}^K \delta_t^i \phi(s_t^i)(\omega_t^i)^\top)\left(\omega_t^k + \beta_t \delta_t^k \mathbf{B}_t^\top \phi(s_t^k)\right) - z^{k,*}\| \\
=& \|\mathbf{B}_t \omega_t^k - z^{k,*} + \beta_t \mathbf{B}_t(\delta_t^k \mathbf{B}_t^\top \phi(s_t^k)) + \frac{\zeta_t}{K}\sum_{i=1}^K \delta_t^i \phi(s_t^i)(\omega_t^i)^\top \omega_t^k \\
&+ \zeta_t \beta_t(\frac{1}{K}\sum_{i=1}^K \delta_t^i \phi(s_t^i)(\omega_t^i)^\top)\left(\delta_t^k \mathbf{B}_t^\top \phi(s_t^k)\right)\|.
\end{aligned}
$$

Squaring both sides, we have

$$
\begin{aligned}
\|\widetilde{x}_{t+1}^k\|^2 =& \|\bar{\mathbf{B}}_{t+1}\widetilde{\omega}_{t+1}^k - z^{k,*}\|^2 \\
\leq & \|\mathbf{B}_t\omega_t^k - z^{k,*}\|^2 + 2\beta_t\langle \mathbf{B}_t\omega_t^k - z^{k,*}, \mathbf{B}_t\mathbf{B}_t^\top\phi(s_t^k)\delta_t^k\rangle \\
& + 2\zeta_t\langle \mathbf{B}_t\omega_t^k - z^{k,*}, \frac{1}{K}\sum_{i=1}^K \phi(s_t^i)\delta_t^i(\omega_t^i)^\top\omega_t^k\rangle \\
& + 2\zeta_t\beta_t\left\langle \mathbf{B}_t\omega_t^k - z^{k,*}, (\frac{1}{K}\sum_{i=1}^K \delta_t^i\phi(s_t^i)(\omega_t^i)^\top)\left(\delta_t^k\mathbf{B}_t^\top\phi(s_t^k)\right)\right\rangle \\
& + 3\beta_t^2\|\mathbf{B}_t(\delta_t^k\mathbf{B}_t^\top\phi(s_t^k))\|^2 + 3\zeta_t^2\|\frac{1}{K}\sum_{i=1}^K \delta_t^i\phi(s_t^i)(\omega_t^i)^\top\omega_t^k\|^2 \\
& + 3\beta_t^2\zeta_t^2\|(\frac{1}{K}\sum_{i=1}^K \delta_t^i\phi(s_t^i)(\omega_t^i)^\top)\left(\delta_t^k\mathbf{B}_t^\top\phi(s_t^k)\right)\|^2.
\end{aligned}
$$

Note that

$$
\|\mathbf{B}_t(\delta_t^k\mathbf{B}_t^\top\phi(s_t^k))\|^2 \leq \|\delta_t^k\|^2 \leq U_\delta^2,
$$

where the last inequality follows from Lemma G.2 and $U_\delta = 2U_r + 2U_\omega$ as defined in Lemma G.2. Similarly,

$$
\|\frac{1}{K}\sum_{i=1}^K \delta_t^i\phi(s_t^i)(\omega_t^i)^\top\omega_t^k\|^2 \leq U_\delta^2 U_\omega^4,
$$

$$
\|(\frac{1}{K}\sum_{i=1}^K \delta_t^i\phi(s_t^i)(\omega_t^i)^\top)\left(\delta_t^k\mathbf{B}_t^\top\phi(s_t^k)\right)\|^2 \leq U_\delta^4 U_\omega^2.
$$

So, $\|\widetilde{x}_{t+1}^k\|^2$ is upper bounded as

$$
\begin{aligned}
\|\widetilde{x}_{t+1}^k\|^2 \leq & \|x_t^k\|^2 + 2\beta_t\langle x_t^k, \mathbf{B}_t\mathbf{B}_t^\top\phi(s_t^k)\delta_t^k\rangle + 2\zeta_t\langle x_t^k, \frac{1}{K}\sum_{i=1}^K \phi(s_t^i)\delta_t^i(\omega_t^i)^\top\omega_t^k\rangle \\
& + 2\zeta_t\beta_t\left\langle x_t^k, (\frac{1}{K}\sum_{i=1}^K \delta_t^i\phi(s_t^i)(\omega_t^i)^\top)\left(\delta_t^k\mathbf{B}_t^\top\phi(s_t^k)\right)\right\rangle + 3\beta_t^2 U_\delta^2 + 3\zeta_t^2 U_\delta^2 U_\omega^4 + 3\beta_t^2\zeta_t^2 U_\delta^4 U_\omega^2.
\end{aligned}
\tag{27}
$$

Roughly speaking, the main drift of the decay in $\|x_t^k\|$ may arise from $2\beta_t\langle x_t^k, \mathbf{B}_t\mathbf{B}_t^\top\phi(s_t^k)\delta_t^k\rangle$ – the first inner product in Eq. (27). However, the environmental heterogeneity significantly complicates the characterization of this term. Specifically, by Proposition 1, it can be written as:

$$
\begin{aligned}
& 2\beta_t\langle x_t^k, \mathbf{B}_t\mathbf{B}_t^\top\phi(s_t^k)\delta_t^k\rangle \\
=& 2\beta_t\langle x_t^k, \mathbf{B}_t\mathbf{B}_t^\top\xi_t^k\rangle + 2\beta_t\langle x_t^k, \mathbf{B}_t\mathbf{B}_t^\top A^k x_t^k\rangle + 2\beta_t\langle x_t^k, \mathbf{B}_t\mathbf{B}_t^\top y_t^k\phi(s_t^k)\rangle \\
=& 2\beta_t\langle x_t^k, \mathbf{P}_t A^k x_t^k\rangle + 2\beta_t\langle x_t^k, \mathbf{P}_t\xi_t^k\rangle + 2\beta_t\langle x_t^k, \mathbf{P}_t y_t^k\phi(s_t^k)\rangle,
\end{aligned}
\tag{28}
$$

recalling that $\mathbf{P}_t = \mathbf{B}_t\mathbf{B}_t^\top$. Inside the first term of Eq. (27), the main negative drift arises from the term $\langle x_t^k, \mathbf{P}_t A^k x_t^k\rangle$. Intuitively, in traditional single-agent or homogeneous environment settings, $\langle x_t^k, \mathbf{P}_t A^k x_t^k\rangle \approx \langle x_t^k, A x_t^k\rangle$, which is mainly controlled by the spectrum of $A$, with the desired property directly assumed in Assumption 1. However, in the presence of environmental heterogeneity, Assumption 1 does not directly guarantee a negative drift of the $x_t^k$ due to the existence of $\mathbf{P}_t$ and its intricate interplay with the heterogeneous $A^k$. Specifically, (1) $\mathbf{P}_t$ is not of full rank, (2) the local head error will be distorted in a different manner due to the product $\mathbf{P}_t A^k$, and (3) $\mathbf{P}_t$ varies over time. To address this, we further decompose $\langle x_t^k, \mathbf{P}_t A^k x_t^k\rangle$ as

$$
\begin{aligned}
& \langle x_t^k, \mathbf{P}_t A^k x_t^k\rangle \\
=& \langle x_t^k, (\mathbf{P}_t - \mathbf{P}^*)A^k x_t^k\rangle + \langle x_t^k, \mathbf{P}^* A^k x_t^k\rangle \\
=& \langle x_t^k, A^k x_t^k\rangle + \langle x_t^k, (\mathbf{P}_t - \mathbf{P}^*)A^k x_t^k\rangle - \langle x_t^k, \mathbf{P}_\perp^* A^k x_t^k\rangle.
\end{aligned}
\tag{29}
$$

The first term of Eq. (29) is a contraction by Assumption 1, the second term can be controlled via the principal angle distance between the subspace estimate $\mathbf{B}_t$ and the underlying truth $\mathbf{B}^*$. The third term can be bounded as

$$
\begin{aligned}
\langle x_t^k, \mathbf{P}_\perp^* A^k x_t^k \rangle &= (\mathbf{P}_\perp^* x_t^k)^\top A^k x_t^k = (\mathbf{P}_\perp^* \left( \mathbf{B}_t \omega_t^k - \mathbf{B}^* \omega^{k,*} \right))^\top A^k x_t^k \\
&= \left( \mathbf{B}_\perp^* \mathbf{B}_\perp^{*\top} \left( \mathbf{B}_t \omega_t^k - \mathbf{B}^* \omega^{k,*} \right) \right)^\top A^k x_t^k \\
&= (\mathbf{B}_\perp^* \mathbf{B}_\perp^{*\top} \mathbf{B}_t \omega_t^k)^\top A^k x_t^k,
\end{aligned}
\tag{30}
$$

wherein the local head error $x_t^k$ and the principal angle distance $\|\mathbf{B}_\perp^{*\top} \mathbf{B}_t\|$ are also coupled. It is worth noting that when $\|\mathbf{B}_\perp^{*\top} \mathbf{B}_t\| = 0$, i.e., when $\mathbf{B}_t$ and $\mathbf{B}^*$ span the same $r$-dimensional subspace, the second and the third terms in Eq. (29) become zero, and Eq. (29) reduces to the standard negative drift term. We postpone the characterization of the convergence of $\|\mathbf{B}_\perp^{*\top} \mathbf{B}_t\|$ to Lemma 5.4.

Combining Eq. (28), (29), and (30), we get

$$
\begin{aligned}
&2\beta_t \langle x_t^k, \mathbf{B}_t \mathbf{B}_t^\top \phi(s_t^k) \delta_t^k \rangle \\
=&2\beta_t \langle x_t^k, A^k x_t^k \rangle + 2\beta_t \langle x_t^k, (\mathbf{P}_t - \mathbf{P}^*) A^k x_t^k \rangle - 2\beta_t (\mathbf{B}_\perp^* \mathbf{B}_\perp^{*\top} \mathbf{B}_t \omega_t^k)^\top A^k x_t^k \\
&+ 2\beta_t \langle x_t^k, \mathbf{P}_t \xi_t^k \rangle + 2\beta_t \langle x_t^k, \mathbf{P}_t y_t^k \phi(s_t^k) \rangle.
\end{aligned}
\tag{31}
$$

By definitions of $A^k$, $x_t^k$, the orthonormality of $\mathbf{B}_t$ and $\mathbf{B}^*$, the projection operation of $\prod_{U_\omega}$, and Assumption 3, it holds that $\|A^k x_t^k\| \leq 4U_\omega$. Invoking Assumption 1, we have

$$
\begin{aligned}
&2\beta_t \langle x_t^k, \mathbf{B}_t \mathbf{B}_t^\top \phi(s_t^k) \delta_t^k \rangle \\
\leq& - 2\beta_t \lambda \|x_t^k\|^2 + 8U_\omega \beta_t \|x_t^k\| \|\mathbf{P}_t - \mathbf{P}^*\| + 8U_\omega \beta_t \|\mathbf{B}_\perp^{*\top} \mathbf{B}_t\| \|x_t^k\| \\
&+ 2\beta_t \langle x_t^k, \mathbf{P}_t \xi_t^k \rangle + 2\beta_t \|x_t^k\| |y_t^k|.
\end{aligned}
\tag{32}
$$

By Proposition 1, the second inner product of Eq. (27) can be written as,

$$
\begin{aligned}
&\frac{2\zeta_t}{K} \langle x_t^k, \sum_{i=1}^K \phi(s_t^i) \delta_t^i (\omega_t^i)^\top \omega_t^k \rangle \\
=&\frac{2\zeta_t}{K} \langle x_t^k, \sum_{i=1}^K A^i x_t^i (\omega_t^i)^\top \omega_t^k \rangle + \frac{2\zeta_t}{K} \langle x_t^k, \sum_{i=1}^K \xi_t^i (\omega_t^i)^\top \omega_t^k \rangle + \frac{2\zeta_t}{K} \langle x_t^k, \sum_{i=1}^K y_t^i \phi(s_t^i) (\omega_t^i)^\top \omega_t^k \rangle \\
\leq&\frac{2\zeta_t}{K} \langle x_t^k, \sum_{i=1,i\neq k}^K A^i x_t^i (\omega_t^i)^\top \omega_t^k \rangle + \frac{2\zeta_t}{K} \langle x_t^k, \sum_{i=1}^K \xi_t^i (\omega_t^i)^\top \omega_t^k \rangle + \frac{2\zeta_t}{K} \langle x_t^k, \sum_{i=1}^K y_t^i \phi(s_t^i) (\omega_t^i)^\top \omega_t^k \rangle \\
\leq&\frac{2\zeta_t}{K} \sum_{i=1,i\neq k}^K \|x_t^k\| \|A^i\| \|x_t^i\| |(\omega_t^i)^\top \omega_t^k| + \frac{2\zeta_t}{K} \sum_{i=1}^K \langle x_t^k, \xi_t^i (\omega_t^i)^\top \omega_t^k \rangle + \frac{2\zeta_t}{K} \sum_{i=1}^K \langle x_t^k, y_t^i \phi(s_t^i) (\omega_t^i)^\top \omega_t^k \rangle \\
\leq&U_\omega^2 \frac{4\zeta_t}{K} \sum_{i=1,i\neq k}^K \|x_t^k\| \|x_t^i\| + \frac{2\zeta_t}{K} \sum_{i=1}^K \langle x_t^k, \xi_t^i (\omega_t^i)^\top \omega_t^k \rangle + \frac{2\zeta_t U_\omega^2}{K} \sum_{i=1}^K \|x_t^k\| |y_t^i|,
\end{aligned}
\tag{33}
$$

where the last inequality holds because $\|A^i\| \leq 2$ by definition.

In addition, the third inner product term of Eq. (27) can be upper bounded as

$$
\begin{aligned}
2\zeta_t \beta_t \left\langle x_t^k, \left(\frac{1}{K} \sum_{i=1}^K \delta_t^i \phi(s_t^i) (\omega_t^i)^\top \right) \left(\delta_t^k \mathbf{B}_t^\top \phi(s_t^k)\right) \right\rangle &\leq 2\zeta_t \beta_t \|x_t^k\| \|\frac{1}{K} \sum_{i=1}^K \delta_t^i \phi(s_t^i) (\omega_t^i)^\top \| \|\delta_t^k \mathbf{B}_t^\top \phi(s_t^k)\| \\
&\leq 4\zeta_t \beta_t U_\omega \frac{1}{K} \sum_{i=1}^K |\delta_t^i| \|\omega_t^i\| |\delta_t^k| \\
&\leq 4\zeta_t \beta_t U_\omega^2 U_\delta^2,
\end{aligned}
\tag{34}
$$

where the last inequality follows from the boundedness of $\omega$ and Lemma G.2.

Plugging the upper bounds in Eq. (32), (33), and (34) back to Eq. (27), we have

$$
\begin{aligned}
\|\widetilde{x}_{t+1}^k\|^2 &\leq (1 - 2\lambda\beta_t)\|x_t^k\|^2 + 2\beta_t\langle x_t^k, \mathbf{P}_t\xi_t^k\rangle + 8\beta_t U_\omega \|x_t^k\|\|\mathbf{P}_t - \mathbf{P}^*\| + 8\beta_t U_\omega \|x_t^k\|\|\mathbf{B}_\perp^{*\top}\mathbf{B}_t\| \\
&+ 2\beta_t\|x_t^k\||y_t^k| + 4\zeta_t U_\omega^2 \frac{1}{K}\sum_{i\neq k}\|x_t^k\|\|x_t^i\| + \frac{2\zeta_t}{K}\sum_{i=1}^K\langle x_t^k, \xi_t^i(\omega_t^i)^\top\omega_t^k\rangle + \frac{2\zeta_t U_\omega^2}{K}\sum_{i=1}^K\|x_t^k\||y_t^i| \\
&+ 4\zeta_t\beta_t U_\omega^2 U_\delta^2 + 3\beta_t^2 U_\delta^2 + 3\zeta_t^2 U_\delta^2 U_\omega^4 + 3\beta_t^2\zeta_t^2 U_\delta^4 U_\omega^2 \\
\leq &(1 - 2\lambda\beta_t - \frac{4\zeta_t U_\omega^2}{K})\|x_t^k\|^2 + 2\beta_t\langle x_t^k, \mathbf{P}_t\xi_t^k\rangle + 16\beta_t U_\omega\|x_t^k\|\|\mathbf{P}_t - \mathbf{P}^*\| \\
&+ 2\beta_t\|x_t^k\||y_t^k| + 4\zeta_t U_\omega^2\frac{1}{K}\sum_{i=1}^K\|x_t^k\|\|x_t^i\| + \frac{2\zeta_t}{K}\sum_{i=1}^K\langle x_t^k, \xi_t^i(\omega_t^i)^\top\omega_t^k\rangle + \frac{2\zeta_t U_\omega^2}{K}\sum_{i=1}^K\|x_t^k\||y_t^i| \\
&+ 4\zeta_t\beta_t U_\omega^2 U_\delta^2 + 3\beta_t^2 U_\delta^2 + 3\zeta_t^2 U_\delta^2 U_\omega^4 + 3\beta_t^2\zeta_t^2 U_\delta^4 U_\omega^2.
\end{aligned}
\tag{35}
$$

Taking expectation and invoking Cauchy–Schwarz inequality, we have

$$
\begin{aligned}
\mathbb{E}\|\widetilde{x}_{t+1}^k\|^2 &\leq (1 - 2\lambda\beta_t - \frac{4\zeta_t U_\omega^2}{K})\mathbb{E}\|x_t^k\|^2 + 2\beta_t\mathbb{E}\langle x_t^k, \mathbf{P}_t\xi_t^k\rangle + 16\beta_t U_\omega\sqrt{\mathbb{E}\|x_t^k\|^2}\sqrt{\mathbb{E}\|m_t\|^2} \\
&+ 2\beta_t\sqrt{\mathbb{E}\|x_t^k\|^2}\sqrt{\mathbb{E}|y_t^k|^2} + 4\zeta_t U_\omega^2\frac{1}{K}\sum_{i=1}^K\sqrt{\mathbb{E}\|x_t^k\|^2}\sqrt{\mathbb{E}\|x_t^i\|^2} + \frac{2\zeta_t}{K}\sum_{i=1}^K\mathbb{E}\langle x_t^k, \xi_t^i(\omega_t^i)^\top\omega_t^k\rangle \\
&+ \frac{2\zeta_t U_\omega^2}{K}\sum_{i=1}^K\sqrt{\mathbb{E}\|x_t^k\|^2}\sqrt{\mathbb{E}|y_t^i|^2} + 4\zeta_t\beta_t U_\omega^2 U_\delta^2 + 3\beta_t^2 U_\delta^2 + 3\zeta_t^2 U_\delta^2 U_\omega^4 + 3\beta_t^2\zeta_t^2 U_\delta^4 U_\omega^2.
\end{aligned}
$$

Putting equation 35 back to equation 26, we get

$$
\begin{aligned}
\|x_{t+1}^k\|^2 &\leq \|\widetilde{x}_{t+1}^k\|^2 + 2U_\delta^2\beta_t^2 + 18U_\omega^4 U_\delta^2\zeta_t^2 + (2\beta_t U_\delta + 6U_\omega^2 U_\delta\zeta_t)4U_\omega \\
\leq &(1 - 2\lambda\beta_t - \frac{4\zeta_t U_\omega^2}{K})\|x_t^k\|^2 + 2\beta_t\langle x_t^k, \mathbf{P}_t\xi_t^k\rangle + 16\beta_t U_\omega\|x_t^k\|\|\mathbf{P}_t - \mathbf{P}^*\| \\
&+ 2\beta_t\|x_t^k\||y_t^k| + 4\zeta_t U_\omega^2\frac{1}{K}\sum_{i=1}^K\|x_t^k\|\|x_t^i\| + \frac{2\zeta_t}{K}\sum_{i=1}^K\langle x_t^k, \xi_t^i(\omega_t^i)^\top\omega_t^k\rangle + \frac{2\zeta_t U_\omega^2}{K}\sum_{i=1}^K\|x_t^k\||y_t^i| \\
&+ 4\zeta_t\beta_t U_\omega^2 U_\delta^2 + 3\beta_t^2 U_\delta^2 + 3\zeta_t^2 U_\delta^2 U_\omega^4 + 3\beta_t^2\zeta_t^2 U_\delta^4 U_\omega^2 + 2U_\delta^2\beta_t^2 + 18U_\omega^4 U_\delta^2\zeta_t^2 + (2\beta_t U_\delta + 6U_\omega^2 U_\delta\zeta_t)4U_\omega.
\end{aligned}
$$

Taking expectation and rearrange, we get

$$
\begin{aligned}
\mathbb{E}\|x_{t+1}^k\|^2 &\leq (1 - 2\lambda\beta_t - \frac{4\zeta_t U_\omega^2}{K})\mathbb{E}\|x_t^k\|^2 + 2\beta_t\mathbb{E}\langle x_t^k, \mathbf{P}_t\xi_t^k\rangle + 16\beta_t U_\omega\mathbb{E}\|x_t^k\|\|\mathbf{P}_t - \mathbf{P}^*\| \\
&+ 2\beta_t\mathbb{E}\|x_t^k\||y_t^k| + 4\zeta_t U_\omega^2\frac{1}{K}\sum_{i=1}^K\mathbb{E}\|x_t^k\|\|x_t^i\| + \frac{2\zeta_t}{K}\sum_{i=1}^K\mathbb{E}\langle x_t^k, \xi_t^i(\omega_t^i)^\top\omega_t^k\rangle + \frac{2\zeta_t U_\omega^2}{K}\sum_{i=1}^K\mathbb{E}\|x_t^k\||y_t^i| \\
&+ 4\zeta_t\beta_t U_\omega^2 U_\delta^2 + 3\beta_t^2 U_\delta^2 + 3\zeta_t^2 U_\delta^2 U_\omega^4 + 3\beta_t^2\zeta_t^2 U_\delta^4 U_\omega^2 + 2U_\delta^2\beta_t^2 + 18U_\omega^4 U_\delta^2\zeta_t^2 + (2\beta_t U_\delta + 6U_\omega^2 U_\delta\zeta_t)4U_\omega. \\
\Rightarrow &(2\lambda\beta_t + \frac{4\zeta_t U_\omega^2}{K})\mathbb{E}\|x_t^k\|^2 \leq (\mathbb{E}\|x_t^k\|^2 - \mathbb{E}\|x_{t+1}^k\|^2) + 2\beta_t\mathbb{E}\langle x_t^k, \mathbf{P}_t\xi_t^k\rangle + 16\beta_t U_\omega\sqrt{\mathbb{E}\|x_t^k\|^2}\sqrt{\mathbb{E}\|m_t\|^2} \\
&+ 2\beta_t\sqrt{\mathbb{E}\|x_t^k\|^2}\sqrt{\mathbb{E}|y_t^k|^2} + 4\zeta_t U_\omega^2\frac{1}{K}\sum_{i=1}^K\sqrt{\mathbb{E}\|x_t^k\|^2}\sqrt{\mathbb{E}\|x_t^i\|^2} + \frac{2\zeta_t}{K}\sum_{i=1}^K\mathbb{E}\langle x_t^k, \xi_t^i(\omega_t^i)^\top\omega_t^k\rangle \\
&+ \frac{2\zeta_t U_\omega^2}{K}\sum_{i=1}^K\sqrt{\mathbb{E}\|x_t^k\|^2}\sqrt{\mathbb{E}|y_t^i|^2} + 4\zeta_t\beta_t U_\omega^2 U_\delta^2 + 3\beta_t^2 U_\delta^2 + 3\zeta_t^2 U_\delta^2 U_\omega^4 + 3\beta_t^2\zeta_t^2 U_\delta^4 U_\omega^2 \\
&+ 2U_\delta^2\beta_t^2 + 18U_\omega^4 U_\delta^2\zeta_t^2 + (2\beta_t U_\delta + 6U_\omega^2 U_\delta\zeta_t)4U_\omega.
\end{aligned}
$$

We can drop $\frac{4\zeta_t U_\omega^2}{K}$ since it does not help much, then,

$$\mathbb{E}\|x_t^k\|^2 \le \frac{1}{2\lambda\beta_t}(\mathbb{E}\|x_t^k\|^2 - \mathbb{E}\|x_{t+1}^k\|^2) + \frac{1}{\lambda}\mathbb{E}\langle x_t^k, \mathbf{P}_t\xi_t^k\rangle + 8U_\omega 2\sqrt{\mathbb{E}\|x_t^k\|^2}\sqrt{\mathbb{E}\|m_t\|^2}$$

$$+ \frac{1}{\lambda}\sqrt{\mathbb{E}\|x_t^k\|^2}\sqrt{\mathbb{E}|y_t^k|^2} + \frac{2c_\zeta U_\omega^2}{c_\beta\lambda}\frac{1}{K}\sum_{i=1}^{K}\sqrt{\mathbb{E}\|x_t^k\|^2}\sqrt{\mathbb{E}\|x_t^i\|^2} + \frac{c_\zeta}{c_\beta\lambda K}\sum_{i=1}^{K}\mathbb{E}\langle x_t^k, \xi_t^i(\omega_t^i)^\top\omega_t^k\rangle$$

$$+ \frac{c_\zeta U_\omega^2}{c_\beta\lambda K}\sum_{i=1}^{K}\sqrt{\mathbb{E}\|x_t^k\|^2}\sqrt{\mathbb{E}|y_t^i|^2} + \frac{1}{\lambda}\Bigg(2\zeta_t U_\omega^2 U_\delta^2 + 2\beta_t U_\delta^2 + 2\zeta_t\frac{c_\zeta}{c_\beta}U_\delta^2 U_\omega^4 + 2\beta_t\zeta_t^2 U_\delta^4 U_\omega^2$$

$$+ U_\delta^2\beta_t + 9U_\omega^4 U_\delta^2\zeta_t\frac{c_\zeta}{c_\beta} + (U_\delta + 3U_\omega^2 U_\delta\frac{c_\zeta}{c_\beta})4U_\omega\Bigg).$$

Taking time average from $\tau_T$ to $T - 1$, and applying Cauchy-Schwartz, we get

$$\frac{1}{T-\tau_T}\sum_{t=\tau_T}^{T}\mathbb{E}\|x_t^k\|^2 \le \underbrace{\frac{1}{T-\tau_T}\sum_{t=\tau_T}^{T}\frac{1}{2\lambda\beta_t}(\mathbb{E}\|x_t^k\|^2 - \mathbb{E}\|x_{t+1}^k\|^2)}_{I_1} + \underbrace{\frac{1}{T-\tau_T}\sum_{t=\tau_T}^{T}\frac{1}{\lambda}\mathbb{E}\langle x_t^k, \mathbf{P}_t\xi_t^k\rangle}_{I_2}$$

$$+ 8U_\omega 2\sqrt{\frac{1}{T-\tau_T}\sum_{t=\tau_T}^{T}\mathbb{E}\|x_t^k\|^2}\sqrt{\frac{1}{T-\tau_T}\sum_{t=\tau_T}^{T}\mathbb{E}\|m_t\|^2}$$

$$+ \frac{1}{\lambda}\sqrt{\frac{1}{T-\tau_T}\sum_{t=\tau_T}^{T}\mathbb{E}\|x_t^k\|^2}\sqrt{\frac{1}{T-\tau_T}\sum_{t=\tau_T}^{T}\mathbb{E}|y_t^k|^2}$$

$$+ \frac{2c_\zeta U_\omega^2}{c_\beta\lambda}\frac{1}{K}\sum_{i=1}^{K}\sqrt{\frac{1}{T-\tau_T}\sum_{t=\tau_T}^{T}\mathbb{E}\|x_t^k\|^2}\sqrt{\frac{1}{T-\tau_T}\sum_{t=\tau_T}^{T}\mathbb{E}\|x_t^i\|^2}$$

$$+ \underbrace{\frac{1}{T-\tau_T}\sum_{t=\tau_T}^{T}\frac{c_\zeta}{c_\beta\lambda K}\sum_{i=1}^{K}\mathbb{E}\langle x_t^k, \xi_t^i(\omega_t^i)^\top\omega_t^k\rangle}_{I_3}$$

$$+ \frac{c_\zeta U_\omega^2}{c_\beta\lambda K}\sum_{i=1}^{K}\sqrt{\frac{1}{T-\tau_T}\sum_{t=\tau_T}^{T}\mathbb{E}\|x_t^k\|^2}\sqrt{\frac{1}{T-\tau_T}\sum_{t=\tau_T}^{T}\mathbb{E}|y_t^i|^2}$$

$$+ \frac{1}{\lambda}\Bigg(2\frac{c_\zeta}{\sqrt{T}}U_\omega^2 U_\delta^2 + 2\frac{c_\beta}{\sqrt{T}}U_\delta^2 + 2\frac{c_\zeta}{\sqrt{T}}\frac{c_\zeta}{c_\beta}U_\delta^2 U_\omega^4 + 2\frac{c_\beta}{\sqrt{T}}\zeta_t^2 U_\delta^4 U_\omega^2$$

$$+ U_\delta^2\frac{c_\beta}{\sqrt{T}} + 9U_\omega^4 U_\delta^2\frac{c_\zeta}{\sqrt{T}}\frac{c_\zeta}{c_\beta} + (U_\delta + 3U_\omega^2 U_\delta\frac{c_\zeta}{c_\beta})4U_\omega\Bigg). \tag{36}$$

For $I_1$, we notice it is a telescoping sum since we pick $\beta_t = \frac{c_\beta}{\sqrt{T}}$, therefore,

$$\frac{1}{T-\tau_T}\sum_{t=\tau_T}^{T}\frac{1}{2\lambda\beta_t}(\mathbb{E}\|x_t^k\|^2 - \mathbb{E}\|x_{t+1}^k\|^2)$$

$$= \frac{\sqrt{T}}{(T-\tau_T)(2\lambda c_\beta)}(\mathbb{E}\|x_{\tau_T}^k\|^2 - \mathbb{E}\|x_T^k\|^2)$$

$$\le \frac{\sqrt{T}}{(T-\tau_T)\lambda c_\beta}2U_\omega^2. \tag{37}$$

For $I_2$, we invoke Lemma G.5 and Lemma G.3, and choose $\tau = \tau_T$,

$$\frac{1}{T - \tau_T} \sum_{t=\tau_T}^{T} \frac{1}{\lambda} \mathbb{E}\langle x_t^k, \mathbf{P}_t \xi_t^k \rangle$$

$$\leq \frac{1}{T - \tau_T} \sum_{t=\tau_T}^{T} \frac{(20 U_\omega U_r + 28 U_\omega^2)}{\lambda} \|\mathbf{B}_t - \mathbf{B}_{t-\tau}\|$$

$$+ \frac{(4 U_r + 12 U_\omega)}{\lambda} \|\omega_t^k - \omega_{t-\tau}^k\| + \frac{8 U_\omega (U_r + U_\omega)}{\lambda \sqrt{T}}$$

$$\leq \frac{(20 U_\omega U_r + 28 U_\omega^2)}{\lambda} 10 \frac{c_\zeta}{\sqrt{T}} \tau_T U_\delta U_\omega + \frac{(4 U_r + 12 U_\omega)}{\lambda} \frac{c_\beta}{\sqrt{T}} \tau_T U_\delta + \frac{8 U_\omega (U_r + U_\omega)}{\lambda \sqrt{T}}. \tag{38}$$

Similarly, for $I_3$, we invoke Lemma G.6 and Lemma G.3, and pick $\tau = \tau_T$, that is,

$$I_3 = \frac{1}{T - \tau_T} \sum_{t=\tau_T}^{T} \frac{c_\zeta}{c_\beta \lambda} \frac{1}{K} \sum_{i=1}^{K} \mathbb{E}\langle x_t^k, \xi_t^k (\omega_t^i)^\top \omega_t^k \rangle$$

$$\leq \frac{1}{T - \tau_T} \sum_{t=\tau_T}^{T} \frac{c_\zeta}{c_\beta \lambda} \Bigg( (8 U_\omega^4 + 4(U_r + U_\omega) U_\omega^2) \|\mathbf{B}_t - \mathbf{B}_{t-\tau}\|$$

$$+ (12 U_r U_\omega^2 + 12 U_\omega^3) \|\omega_t^k - \omega_{t-\tau}^k\| + \frac{1}{K} \sum_{i=1}^{K} (16 U_\omega^3 + 8 U_r U_\omega^3) \|\omega_t^i - \omega_{t-\tau}^i\| + 4 U_\omega^3 m \rho^{\tau-1} \Bigg)$$

$$\leq \frac{c_\zeta}{c_\beta \lambda} \Bigg( (8 U_\omega^4 + 4(U_r + U_\omega) U_\omega^2) 10 \frac{c_\zeta}{\sqrt{T}} \tau_T U_\delta U_\omega + (12 U_r U_\omega^2 + 12 U_\omega^3) \frac{c_\beta}{\sqrt{T}} \tau_T U_\delta$$

$$+ \frac{1}{K} \sum_{i=1}^{K} (16 U_\omega^3 + 8 U_r U_\omega^3) \frac{c_\beta}{\sqrt{T}} \tau_T U_\delta + 4 U_\omega^3 \frac{1}{\sqrt{T}} \Bigg). \tag{39}$$

Putting equation 37, equation 38, equation 39 back to equation 36, and omitting the constant terms, we get

$$X_T^k \leq \frac{\sqrt{T}}{(T - \tau_T) \lambda c_\beta} 2 U_\omega^2 + \mathcal{O}\left(\frac{\tau_T}{\sqrt{T}}\right) + 16 U_\omega \sqrt{X_T^k M_T} + \frac{1}{\lambda} \sqrt{X_T^k Y_T^k}$$

$$+ \frac{2 c_\zeta U_\omega^2}{c_\beta \lambda} \frac{1}{K} \sum_{i=1}^{K} \sqrt{X_T^k X_T^i} + \frac{c_\zeta U_\omega^2}{c_\beta \lambda K} \sum_{i=1}^{K} \sqrt{X_T^k Y_T^i} + \mathcal{O}(1).$$

Since $\tau_T = \mathcal{O}(\log(T))$, we further have

$$X_T^k \leq \mathcal{O}\left(\frac{\log(T)}{\sqrt{T}}\right) + 16 U_\omega \sqrt{X_T^k M_T} + \frac{1}{\lambda} \sqrt{X_T^k Y_T^k}$$

$$+ \frac{2 c_\zeta U_\omega^2}{c_\beta \lambda} \frac{1}{K} \sum_{i=1}^{K} \sqrt{X_T^k X_T^i} + \frac{c_\zeta U_\omega^2}{c_\beta \lambda K} \sum_{i=1}^{K} \sqrt{X_T^k Y_T^i} + \mathcal{O}(1). \tag{40}$$

## C.2 LOWER BOUND: PROOF OF LEMMA 5.2

Recall that $x_t^k = \mathbf{B}_t\omega_t^k - z^{k,*} = \mathbf{B}_t\omega_t^k - \mathbf{B}^*\omega^{k,*}$. By definition, $\mathbf{B}_t\omega_t^k$ lies in the subspace spanned by $\mathbf{B}_t$. As illustrated in Fig.1, the distance between $\mathbf{B}_t\omega_t^k$ and $z^{k,*}$ is no smaller than the distance between $z^{k,*}$ and the subspace spanned by $\mathbf{B}_t$.
Hence, it holds that

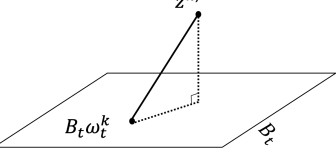

$$\|x_t^k\| = \|\mathbf{B}_t\omega_t^k - z^{k,*}\| \geq \|\mathbf{P}_{t,\perp}z^{k,*}\|$$
$$= \|(\mathbf{I} - \mathbf{P}_t)z^{k,*}\| = \|(\mathbf{P}^* - \mathbf{P}_t)z^{k,*}\|$$
$$= \|(\mathbf{P}_t - \mathbf{P}^*)z^{k,*}\|.$$

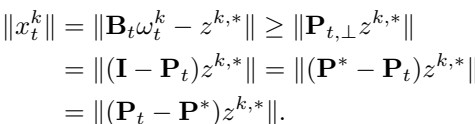

Figure 1: Geometry illustration.

Recall that $\mathbf{P}_t = \mathbf{B}_t\mathbf{B}_t^\top$ and $\mathbf{P}^* = \mathbf{B}^*(\mathbf{B}^*)^\top$. We first rewrite $\mathbf{P}_t - \mathbf{P}^*$ to relate it to the cross terms $\mathbf{B}_t^\top\mathbf{B}_\perp^*$ and $\mathbf{B}_{t,\perp}^\top\mathbf{B}^*$. Observe that

$$\begin{bmatrix} \mathbf{B}_t^\top \\ \mathbf{B}_{t,\perp}^\top \end{bmatrix} (\mathbf{B}_t\mathbf{B}_t^\top - \mathbf{B}^*\mathbf{B}^{*\top}) [\mathbf{B}_\perp^* \quad \mathbf{B}^*] = \begin{bmatrix} \mathbf{B}_t^\top\mathbf{B}_\perp^* & \mathbf{0} \\ \mathbf{0} & -\mathbf{B}_{t,\perp}^\top\mathbf{B}^* \end{bmatrix}.$$

It is easy to check that

$$\mathbf{B}_t\mathbf{B}_t^\top - \mathbf{B}^*\mathbf{B}^{*\top} = [\mathbf{B}_t \quad \mathbf{B}_{t,\perp}] \begin{bmatrix} \mathbf{B}_t^\top\mathbf{B}_\perp^* & \mathbf{0} \\ \mathbf{0} & -\mathbf{B}_{t,\perp}^\top\mathbf{B}^* \end{bmatrix} \begin{bmatrix} \mathbf{B}_\perp^{*\top} \\ \mathbf{B}^{*\top} \end{bmatrix}.$$

We can do singular value decomposition for the diagonal blocks; for ease of exposition, we drop the time index $t$ in the decomposition. That is

$$\mathbf{B}_t^\top\mathbf{B}_\perp^* = \mathbf{X_1}\mathbf{\Sigma_1}\mathbf{Y_1^\top}, \quad \text{and} \quad -\mathbf{B}_{\mathbf{t},\perp}^\top\mathbf{B}^* = \mathbf{X_2}\mathbf{\Sigma_2}\mathbf{Y_2^\top}.$$

Then, we have

$$\mathbf{B}_t\mathbf{B}_t^\top - \mathbf{B}^*\mathbf{B}^{*\top} = [\mathbf{B}_t \quad \mathbf{B}_{t,\perp}] \begin{bmatrix} \mathbf{X}_1 & \mathbf{0} \\ \mathbf{0} & \mathbf{X}_2 \end{bmatrix} \begin{bmatrix} \mathbf{\Sigma_1} & \mathbf{0} \\ \mathbf{0} & \mathbf{\Sigma_2} \end{bmatrix} \begin{bmatrix} \mathbf{Y}_1^\top & \mathbf{0} \\ \mathbf{0} & \mathbf{Y}_2^\top \end{bmatrix} \begin{bmatrix} \mathbf{B}_\perp^{*\top} \\ \mathbf{B}^{*\top} \end{bmatrix}$$

$$= [\mathbf{B}_t\mathbf{X}_1 \quad \mathbf{B}_{t,\perp}\mathbf{X}_2] \begin{bmatrix} \mathbf{\Sigma_1} & \mathbf{0} \\ \mathbf{0} & \mathbf{\Sigma_2} \end{bmatrix} \begin{bmatrix} (\mathbf{B}_\perp^*\mathbf{Y}_1)^\top \\ (\mathbf{B}^*\mathbf{Y}_2)^\top \end{bmatrix}.$$

So,

$$\|(\mathbf{P}_t - \mathbf{P}^*)z^{k,*}\| = \left\| [\mathbf{B}_t\mathbf{X}_1 \quad \mathbf{B}_{t,\perp}\mathbf{X}_2] \begin{bmatrix} \mathbf{\Sigma_1} & \mathbf{0} \\ \mathbf{0} & \mathbf{\Sigma_2} \end{bmatrix} \begin{bmatrix} (\mathbf{B}_\perp^*\mathbf{Y}_1)^\top \\ (\mathbf{B}^*\mathbf{Y}_2)^\top \end{bmatrix} z^{k,*} \right\|$$

$$= \left\| \begin{bmatrix} \mathbf{\Sigma_1} & \mathbf{0} \\ \mathbf{0} & \mathbf{\Sigma_2} \end{bmatrix} \begin{bmatrix} (\mathbf{B}_\perp^*\mathbf{Y}_1)^\top \\ (\mathbf{B}^*\mathbf{Y}_2)^\top \end{bmatrix} z^{k,*} \right\|$$

From the proof of (Chen et al., 2021, Lemma 2.4), we know that when $d \geq 2r$, $\mathbf{\Sigma_1}$ and $\mathbf{\Sigma_2}$ are identical up to permutation. Let $\sigma_1$ denote the largest singular value. From (Chen et al., 2021, Lemma 2.5), we know that

$$\sigma_1 = \|\mathbf{B}_t^\top\mathbf{B}_\perp^*\| = \|\mathbf{B}_{t,\perp}^\top\mathbf{B}^*\| = \|\mathbf{B}_t\mathbf{B}_t^\top - \mathbf{B}^*\mathbf{B}^{*\top}\| = \|m_t\|. \tag{41}$$

Let $\mathbf{v}_1$ and $\mathbf{v}_1^*$ denote the corresponding right singular vectors of $\mathbf{B}_t^\top\mathbf{B}_\perp^*$ and $\mathbf{B}_{t,\perp}^\top\mathbf{B}^*$. It is easy to see that $\mathbf{v}_1 \in \text{span}(\mathbf{B}^*)$ and $\mathbf{v}_1^* \in \text{span}(\mathbf{B}_\perp^*)$. Hence, we get

$$\|(\mathbf{P}_t - \mathbf{P}^*)z^{k,*}\| = \left\| \begin{bmatrix} \mathbf{\Sigma_1} & \mathbf{0} \\ \mathbf{0} & \mathbf{\Sigma_2} \end{bmatrix} \begin{bmatrix} (\mathbf{B}_\perp^*\mathbf{Y}_1)^\top \\ (\mathbf{B}^*\mathbf{Y}_2)^\top \end{bmatrix} z^{k,*} \right\|$$

$$\geq \sqrt{(\sigma_1\mathbf{v}_1^\top z^{k,*})^2 + (\sigma_1(\mathbf{v}_1^*)^\top z^{k,*})^2}$$

$$= \sigma_1|\mathbf{v}_1^\top z^{k,*}|,$$

where the last line follows from $z^{k,*} \in \text{span}(\mathbf{B}^*)$ and thus $(\mathbf{v}_1^*)^\top z^{k,*} = 0, \forall k$.

Averaging the reconstruction error across all agents, we get

$$\frac{1}{K}\sum_{k=1}^{K}\|x_t^k\|^2 \geq \frac{1}{K}\sum_{k=1}^{K}\sigma_1^2(\mathbf{v}_1^\top z^{k,*})^2 = \sigma_1^2\frac{1}{K}\sum_{k=1}^{K}(\mathbf{v}_1^\top z^{k,*})^2$$

$$=\sigma_1^2\frac{1}{K}\left\|\mathbf{v}_1^\top\mathbf{Z}^*\right\|^2$$

$$=\sigma_1^2\frac{1}{K}\mathbf{v}_1^\top\mathbf{Z}^*\mathbf{Z}^{*\top}\mathbf{v}_1.$$

Recall from Eq. (4) that $z^{k,*} = \mathbf{B}^*\omega^{k,*}$. Then,

$$\frac{\sigma_1^2}{K}\mathbf{v}_1^\top\mathbf{Z}^*\mathbf{Z}^{*\top}\mathbf{v}_1 \geq \frac{\sigma_1^2}{K}\lambda_{\min}^+(\mathbf{Z}^*\mathbf{Z}^{*\top}).$$

Therefore, we have

$$\frac{1}{K}\sum_{k=1}^{K}\|x_t^k\|^2 \geq \frac{\sigma_1^2}{K}\lambda_{\min}^+(\mathbf{Z}^*\mathbf{Z}^{*\top}) = \frac{\|m_t\|^2}{K}\lambda_{\min}^+(\mathbf{Z}^*\mathbf{Z}^{*\top}),$$

where the last equality follows from Eq. (41). Taking expectation, we have

$$\frac{1}{K}\sum_{k=1}^{K}\mathbb{E}\|x_t^k\|^2 \geq \frac{\mathbb{E}\|m_t\|^2}{K}\lambda_{\min}^+(\mathbf{Z}^*\mathbf{Z}^{*\top}) \geq \frac{\mathbb{E}\|m_t\|_F^2}{rK}\lambda_{\min}^+(\mathbf{Z}^*\mathbf{Z}^{*\top}),$$

completing the proof of the lemma.

## D    BOUNDS ON PRINCIPAL ANGLE DISTANCE

Recall from Algorithm 1 the update of $\bar{\mathbf{B}}$:

$$\bar{\mathbf{B}}_{t+1} = \mathbf{B}_t + \frac{\zeta_t}{K} \sum_{k=1}^{K} \delta_t^k \phi(s_t^k)(\omega_t^k)^\top$$

**Proof of Lemma 5.4**    Now we are ready to decompose the error.

$$\|\mathbf{B}_\perp^{*\top}\mathbf{B}_{t+1}\|_F^2 = \|\mathbf{B}_\perp^{*\top}\bar{\mathbf{B}}_{t+1}\mathbf{R}_{t+1}^{-1}\|_F^2 \leq \|\mathbf{B}_\perp^{*\top}\bar{\mathbf{B}}_{t+1}\|_F^2 \|\mathbf{R}_{t+1}^{-1}\|^2.$$

That is,

$$\|m_{t+1}\|_F^2 \leq \|\bar{m}_{t+1}\|_F^2 \|\mathbf{R}_{t+1}^{-1}\|^2.$$

For $\|\bar{m}_{t+1}\|_F^2$, we have

$$\|\bar{m}_{t+1}\|_F^2 = \|\mathbf{B}_\perp^{*\top}\bar{\mathbf{B}}_{t+1}\|_F^2 = \|\mathbf{B}_\perp^{*\top}\mathbf{B}_t + \frac{\zeta_t}{K} \sum_{k=1}^{K} \mathbf{B}_\perp^{*\top}\delta_t^k \phi(s_t^k)(\omega_t^k)^\top\|_F^2$$

$$= \|\mathbf{B}_\perp^{*\top}\mathbf{B}_t\|_F^2 + 2\langle \mathbf{B}_\perp^{*\top}\mathbf{B}_t, \frac{\zeta_t}{K} \sum_{k=1}^{K} \mathbf{B}_\perp^{*\top}\delta_t^k \phi(s_t^k)(\omega_t^k)^\top\rangle + \|\frac{\zeta_t}{K} \sum_{k=1}^{K} \mathbf{B}_\perp^{*\top}\delta_t^k \phi(s_t^k)(\omega_t^k)^\top\|_F^2$$

$$\overset{(a)}{=} \|\mathbf{B}_\perp^{*\top}\mathbf{B}_t\|_F^2 + \|\frac{\zeta_t}{K} \sum_{k=1}^{K} \mathbf{B}_\perp^{*\top}\delta_t^k \phi(s_t^k)(\omega_t^k)^\top\|_F^2 + 2\langle \mathbf{B}_\perp^{*\top}\mathbf{B}_t, \frac{\zeta_t}{K} \sum_{k=1}^{K} \mathbf{B}_\perp^{*\top}\xi_t^k(\omega_t^k)^\top\rangle$$

$$+ 2\langle \mathbf{B}_\perp^{*\top}\mathbf{B}_t, \frac{\zeta_t}{K} \sum_{k=1}^{K} \mathbf{B}_\perp^{*\top} y_t^k \phi(s_t^k)(\omega_t^k)^\top\rangle + 2\langle \mathbf{B}_\perp^{*\top}\mathbf{B}_t, \frac{\zeta_t}{K} \sum_{k=1}^{K} \mathbf{B}_\perp^{*\top} A^k x_t^k(\omega_t^k)^\top\rangle, \tag{42}$$

where equality (a) follows from Proposition 1. It turns out that the last inner product of Eq. (42) is related to the negative drift of the $m_t$. Specifically,

$$\langle \mathbf{B}_\perp^{*\top}\mathbf{B}_t, \frac{\zeta_t}{K} \sum_{k=1}^{K} \mathbf{B}_\perp^{*\top} A^k x_t^k(\omega_t^k)^\top\rangle$$

$$= \langle \mathbf{B}_\perp^{*\top}\mathbf{B}_t, \frac{\zeta_t}{K} \sum_{k=1}^{K} \mathbf{B}_\perp^{*\top} A^k(\mathbf{P}^* + \mathbf{P}_\perp^*) x_t^k(\omega_t^k)^\top\rangle$$

$$= \langle \mathbf{B}_\perp^{*\top}\mathbf{B}_t, \frac{\zeta_t}{K} \sum_{k=1}^{K} \mathbf{B}_\perp^{*\top} A^k \mathbf{B}^* \mathbf{B}^{*\top} x_t^k(\omega_t^k)^\top\rangle + \langle \mathbf{B}_\perp^{*\top}\mathbf{B}_t, \frac{\zeta_t}{K} \sum_{k=1}^{K} \mathbf{B}_\perp^{*\top} A^k \mathbf{B}_\perp^* \mathbf{B}_\perp^{*\top} x_t^k(\omega_t^k)^\top\rangle$$

$$\overset{(a)}{=} \langle \mathbf{B}_\perp^{*\top}\mathbf{B}_t, \frac{\zeta_t}{K} \sum_{k=1}^{K} \mathbf{B}_\perp^{*\top} A^k \mathbf{B}_\perp^* \mathbf{B}_\perp^{*\top}(\mathbf{B}_t\omega_t^k - \mathbf{B}^*\omega^{k,*})(\omega_t^k)^\top\rangle$$

$$= \frac{\zeta_t}{K} \sum_{k=1}^{K} (\mathbf{B}_\perp^{*\top}\mathbf{B}_t\omega_t^k)^\top \mathbf{B}_\perp^{*\top} A^k \mathbf{B}_\perp^* \mathbf{B}_\perp^{*\top}(\mathbf{B}_t\omega_t^k - \mathbf{B}^*\omega^{k,*})$$

$$= \frac{\zeta_t}{K} \sum_{k=1}^{K} (\mathbf{B}_\perp^{*\top}(\mathbf{B}_t\omega_t^k - \mathbf{B}^*\omega^{k,*}))^\top (\mathbf{B}_\perp^{*\top} A^k \mathbf{B}_\perp^*) \mathbf{B}_\perp^{*\top}(\mathbf{B}_t\omega_t^k - \mathbf{B}^*\omega^{k,*})$$

$$\leq -\lambda \frac{\zeta_t}{K} \sum_{k=1}^{K} (\mathbf{B}_\perp^{*\top}(\mathbf{B}_t\omega_t^k - \mathbf{B}^*\omega^{k,*}))^\top \mathbf{B}_\perp^{*\top}(\mathbf{B}_t\omega_t^k - \mathbf{B}^*\omega^{k,*})$$

(a) holds because we assume span($\mathbf{B}^*$) is $A^k$-invariant, i.e., $A^k v \in \text{span}(\mathbf{B}^*), \forall v \in \text{span}(\mathbf{B}^*)$. Consequently, $\mathbf{P}_\perp^* A^k \mathbf{P}^* = 0$. The first two inner products in Eq. (42) can be bounded similarly to

the derivation in the proof of Lemma 5.1. Thus, we have

$$\|\bar{m}_{t+1}\|_F^2 \leq \|m_t\|_F^2 + \frac{\zeta_t^2}{K} U_\delta^2 U_\omega^2 + 2\langle m_t, \frac{\zeta_t}{K} \sum_{k=1}^{K} \mathbf{B}_\perp^{*\top} \xi_t^k (\omega_t^k)^\top \rangle$$

$$+ 2\langle m_t, \frac{\zeta_t}{K} \sum_{k=1}^{K} \mathbf{B}_\perp^{*\top} y_t^k \phi(s_t^k)(\omega_t^k)^\top \rangle - \frac{2\lambda\zeta_t}{K} \sum_{k=1}^{K} \|x_t^k\|^2.$$

Applying Lemma 5.2 – the lower bound of $\frac{1}{K}\sum_{k=1}^{K}\|x_t^k\|^2$, we get

$$\|\bar{m}_{t+1}\|_F^2 \leq (1 - \frac{2\lambda\lambda_{\min}^+\zeta_t}{rK})\|m_t\|_F^2 + \frac{\zeta_t^2}{K} U_\delta^2 U_\omega^2$$

$$+ 2\langle m_t, \frac{\zeta_t}{K} \sum_{k=1}^{K} \mathbf{B}_\perp^{*\top} \xi_t^k (\omega_t^k)^\top \rangle + 2\langle m_t, \frac{\zeta_t}{K} \sum_{k=1}^{K} \mathbf{B}_\perp^{*\top} y_t^k \phi(s_t^k)(\omega_t^k)^\top \rangle.$$

By assumption 5, we further have

$$\|\bar{m}_{t+1}\|_F^2 \leq (1 - \frac{2\lambda\alpha\zeta_t}{r})\|m_t\|_F^2 + \frac{\zeta_t^2}{K} U_\delta^2 U_\omega^2$$

$$+ 2\langle m_t, \frac{\zeta_t}{K} \sum_{k=1}^{K} \mathbf{B}_\perp^{*\top} \xi_t^k (\omega_t^k)^\top \rangle + 2\langle m_t, \frac{\zeta_t}{K} \sum_{k=1}^{K} \mathbf{B}_\perp^{*\top} y_t^k \phi(s_t^k)(\omega_t^k)^\top \rangle.$$

Recall that $\|m_{t+1}\|_F^2 \leq \|\bar{m}_{t+1}\|_F^2 \|\mathbf{R}_{t+1}^{-1}\|^2$, we have

$$\|m_{t+1}\|_F^2 \leq (1 - \frac{2\lambda\alpha\zeta_t}{r})\|m_t\|_F^2 \|\mathbf{R}_{t+1}^{-1}\|^2 + \frac{\zeta_t^2}{K} U_\delta^2 U_\omega^2 \|\mathbf{R}_{t+1}^{-1}\|^2$$

$$+ 2\langle m_t, \frac{\zeta_t}{K} \sum_{k=1}^{K} \mathbf{B}_\perp^{*\top} \xi_t^k (\omega_t^k)^\top \rangle \|\mathbf{R}_{t+1}^{-1}\|^2 + 2\langle m_t, \frac{\zeta_t}{K} \sum_{k=1}^{K} \mathbf{B}_\perp^{*\top} y_t^k \phi(s_t^k)(\omega_t^k)^\top \rangle \|\mathbf{R}_{t+1}^{-1}\|^2.$$

For the first term, we use Eq. (17) that $\|\mathbf{R}_{t+1}^{-1}\| \leq (\frac{1}{1-4U_\delta U_\omega \zeta_t})^2$, and for the remaining terms, we use a crude bound that $\|\mathbf{R}_{t+1}^{-1}\| \leq (\frac{1}{1-4U_\delta U_\omega \zeta_t})^2 \leq 2$ when $4U_\delta U_\omega \zeta_t \leq \frac{1}{6}$, then,

$$\|m_{t+1}\|_F^2 \leq (1 - \frac{2\lambda\alpha\zeta_t}{r})(\frac{1}{1-4U_\delta U_\omega \zeta_t})^2 \|m_t\|_F^2 + 2\frac{\zeta_t^2}{K} U_\delta^2 U_\omega^2$$

$$+ 4\langle m_t, \frac{\zeta_t}{K} \sum_{k=1}^{K} \mathbf{B}_\perp^{*\top} \xi_t^k (\omega_t^k)^\top \rangle + 4\langle m_t, \frac{\zeta_t}{K} \sum_{k=1}^{K} \mathbf{B}_\perp^{*\top} y_t^k \phi(s_t^k)(\omega_t^k)^\top \rangle.$$

Moreover, since $4U_\delta U_\omega \zeta_t \leq \frac{1}{6}$, we have $(\frac{1}{1-4U_\delta U_\omega \zeta_t})^2 \leq 1 + 12U_\delta U_\omega \zeta_t$.

Suppose $\frac{\lambda\alpha}{r} \geq 12U_\delta U_\omega$,

$$(1 - \frac{2\lambda\alpha\zeta_t}{r})(\frac{1}{1-4U_\delta U_\omega \zeta_t})^2$$

$$\leq (1 - \frac{2\lambda\alpha\zeta_t}{r})(1 + 12U_\delta U_\omega \zeta_t) \leq (1 - \frac{\lambda\alpha\zeta_t}{r}).$$

Then,

$$\|m_{t+1}\|_F^2 \leq (1 - \frac{\lambda\alpha\zeta_t}{r})\|m_t\|_F^2 + 2\frac{\zeta_t^2}{K} U_\delta^2 U_\omega^2$$

$$+ 4\langle m_t, \frac{\zeta_t}{K} \sum_{k=1}^{K} \mathbf{B}_\perp^{*\top} \xi_t^k (\omega_t^k)^\top \rangle + 4\langle m_t, \frac{\zeta_t}{K} \sum_{k=1}^{K} \mathbf{B}_\perp^{*\top} y_t^k \phi(s_t^k)(\omega_t^k)^\top \rangle.$$

Taking expectation, we have

$$\mathbb{E}\|m_{t+1}\|_F^2 \leq (1 - \frac{\lambda\alpha\zeta_t}{r})\mathbb{E}\|m_t\|_F^2 + 2\frac{\zeta_t^2}{K}U_\delta^2 U_\omega^2$$

$$+ 4\mathbb{E}\langle m_t, \frac{\zeta_t}{K}\sum_{k=1}^K \mathbf{B}_\perp^{*\top}\xi_t^k(\omega_t^k)^\top\rangle + 4\frac{\zeta_t U_\omega}{K}\sum_{k=1}^K \sqrt{\mathbb{E}\|m_t\|^2}\sqrt{\mathbb{E}|y_t^k|^2}.$$

which implies

$$\frac{1}{T-\tau_T}\sum_{t=\tau_T}^{T-1}\mathbb{E}\|m_t\|_F^2 \leq \underbrace{\frac{1}{T-\tau_T}\frac{r}{\lambda\alpha}\sum_{t=\tau_T}^{T-1}\frac{1}{\zeta_t}(\mathbb{E}\|m_t\|_F^2 - \mathbb{E}\|m_{t+1}\|_F^2)}_{I_1}$$

$$+ \underbrace{\frac{1}{T-\tau_T}\sum_{t=\tau_T}^{T-1}\frac{2r\zeta_t}{\lambda\alpha K}U_\delta^2 U_\omega^2}_{I_2}$$

$$+ \underbrace{\frac{1}{T-\tau_T}\sum_{t=\tau_T}^{T-1}\sum_{k=1}^K\frac{4r}{\lambda\alpha K}\mathbb{E}\langle m_t, \mathbf{B}_\perp^{*\top}\xi_t^k(\omega_t^k)^\top\rangle}_{I_3}$$

$$+ \underbrace{\frac{1}{T-\tau_T}\sum_{t=\tau_T}^{T-1}\frac{4rU_\omega}{\lambda\alpha K}\sum_{k=1}^K\sqrt{\mathbb{E}\|m_t\|^2}\sqrt{\mathbb{E}|y_t^k|^2}}_{I_4}.$$

For $I_1$, we have

$$I_1 = \frac{\sqrt{T}}{T-\tau_T}\frac{r}{\lambda\alpha c_\zeta}(\mathbb{E}\|m_{\tau_T}\|_F^2 - \mathbb{E}\|m_T\|_F^2)$$

$$\leq \frac{\sqrt{T}}{T-\tau_T}\frac{r^2}{\lambda\alpha c_\zeta}$$

$$= \mathcal{O}(\frac{r^2}{\sqrt{T}}).$$

The last line holds because $T \geq 2\tau_T$.

For $I_2$, we have

$$\frac{1}{T-\tau_T}\sum_{t=\tau_T}^{T-1}\frac{2r\zeta_t}{\lambda K\alpha}U_\delta^2 U_\omega^2 \leq \frac{2rc_\zeta U_\delta^2 U_\omega^2}{\sqrt{T}\lambda K\alpha} = \mathcal{O}(\frac{r}{K\sqrt{T}}).$$

For $I_3$, we invoke Lemma G.3 and Lemma G.4,

$$\mathbb{E}\langle m_t, \mathbf{B}_\perp^{*\top}\xi_t^k(\omega_t^k)^\top\rangle$$

$$\leq \left(8U_\omega^2 + 4U_\omega U_r\right)\|\mathbf{B}_t - \mathbf{B}_{t-\tau}\| + (8U_\omega + 4U_r)\|\omega_t^k - \omega_{t-\tau}^k\| + (4U_r + 4U_\omega)U_\omega m\rho^{\tau-1}$$

$$\leq \left(8U_\omega^2 + 4U_\omega U_r\right)10\zeta_{t-\tau}\tau U_\delta U_\omega + (8U_\omega + 4U_r)\beta_{t-\tau}\tau U_\delta U_\mathbf{B} + (4U_r + 4U_\omega)U_\omega m\rho^{\tau-1}.$$

Then, choosing $\tau = \tau_T$, we get,

$$\frac{1}{T-\tau_T}\sum_{t=\tau_T}^{T-1}\sum_{k=1}^K\frac{4r}{\lambda K\alpha}\mathbb{E}\langle m_t, \mathbf{B}_\perp^{*\top}\xi_t^k(\omega_t^k)^\top\rangle$$

$$\leq \frac{4r}{\lambda\alpha\sqrt{T}}\left[\left(8U_\omega^2 + 4U_\omega U_r\right)10c_\zeta\tau_T U_\delta U_\omega + (8U_\omega + 4U_r)c_\beta\tau_T U_\delta U_\mathbf{B} + (4U_r + 4U_\omega)U_\omega\right]$$

$$= \mathcal{O}(\frac{r\log(T)}{\sqrt{T}}).$$

For $I_4$, by Cauchy-Schwartz, we have

$$\frac{1}{T-\tau_T} \sum_{t=\tau_T}^{T-1} \frac{4rU_\omega}{\lambda K\alpha} \sum_{k=1}^{K} \sqrt{\mathbb{E}\|m_t\|^2}\sqrt{\mathbb{E}|y_t^k|^2}$$

$$\leq \frac{4rU_\omega}{\lambda\alpha K} \sum_{k=1}^{K} \sqrt{\frac{1}{T-\tau_T} \sum_{t=\tau_T}^{T-1} \mathbb{E}\|m_t\|^2}\sqrt{\frac{1}{T-\tau_T} \sum_{t=\tau_T}^{T-1} \mathbb{E}|y_t^k|^2}.$$

Combining the above, we have,

$$M_T = \frac{1}{T-\tau_T} \sum_{t=\tau_T}^{T-1} \mathbb{E}\|m_t\|_F^2$$

$$\leq \frac{\sqrt{T}}{T-\tau_T} \frac{r^2}{\lambda\alpha c_\zeta} + \frac{2rc_\zeta U_\delta^2 U_\omega^2}{\sqrt{T}\lambda K\alpha} + \frac{4rC_1}{\lambda\alpha\sqrt{T}} + \frac{4rU_\omega}{\lambda K\alpha} \sum_{k=1}^{K} \sqrt{M_T Y_T^k},$$

where $C_1 = \left[\left(8U_\omega^2 + 4U_\omega U_r\right) 10c_\zeta \tau_T U_\delta U_\omega + (8U_\omega + 4U_r)c_\beta \tau_T U_\delta U_{\mathbf{B}} + (4U_r + 4U_\omega)U_\omega\right].$

## E  PROOF OF LEMMA 5.3

By Eq. (12) and the update of $\eta^k$ in Algorithm 1, we have

$$y_{t+1}^k = \eta_{t+1}^k - J^k = \eta_t^k + \gamma_t(r_t^k - \eta_t^k) - J^k = y_t^k + \gamma_t(r_t^k - \eta_t^k).$$

Thus,

$$
\begin{aligned}
(y_{t+1}^k)^2 &= (y_t^k)^2 + 2\gamma_t y_t^k(r_t^k - \eta_t^k) + (\gamma_t(r_t^k - \eta_t^k))^2 \\
&= (y_t^k)^2 + 2\gamma_t y_t^k(r_t^k - \eta_t^k + J^k - J^k) + (\gamma_t(r_t^k - \eta_t^k))^2 \\
&= (1 - 2\gamma_t)(y_t^k)^2 + 2\gamma_t y_t^k(r_t^k - J^k) + (\gamma_t(r_t^k - \eta_t^k))^2.
\end{aligned}
$$

Taking the expectation and rearranging the terms, we get

$$\mathbb{E}|y_t^k|^2 \le \frac{1}{2\gamma_t}(\mathbb{E}|y_t^k|^2 - \mathbb{E}|y_{t+1}^k|^2) + \mathbb{E}[y_t^k(r_t^k - J^k)] + \frac{\gamma_t}{2}\mathbb{E}[(r_t^k - \eta_t^k)^2].$$

Taking the time-average over $t$ from $\tau_T$ to $T-1$, we get,

$$\frac{1}{T - \tau_T}\sum_{t=\tau_T}^{T-1}\mathbb{E}|y_t^k|^2$$

$$\le \frac{1}{T - \tau_T}\sum_{t=\tau_T}^{T-1}\frac{1}{2\gamma_t}(\mathbb{E}|y_t^k|^2 - \mathbb{E}|y_{t+1}^k|^2) + \frac{1}{T - \tau_T}\sum_{t=\tau_T}^{T-1}\mathbb{E}y_t^k(r_t^k - J^k)$$

$$+ \frac{1}{T - \tau_T}\sum_{t=\tau_T}^{T-1}\frac{\gamma_t}{2}\mathbb{E}[(r_t^k - \eta_t^k)^2]$$

$$\overset{(a)}{\le} \frac{1}{T - \tau_T}\sum_{t=\tau_T}^{T-1}\frac{1}{2\gamma_t}(\mathbb{E}|y_t^k|^2 - \mathbb{E}|y_{t+1}^k|^2) + \frac{1}{T - \tau_T}\sum_{t=\tau_T}^{T-1}\mathbb{E}[y_t^k(r_t^k - J^k)] + 2\frac{1}{T - \tau_T}\sum_{t=\tau_T}^{T-1}\gamma_t U_r^2$$

$$\le \frac{\sqrt{T}}{T - \tau_T}\frac{2U_r^2}{c_\gamma} + \frac{1}{T - \tau_T}\sum_{t=\tau_T}^{T-1}\mathbb{E}[y_t^k(r_t^k - J^k)] + \frac{2c_\gamma U_r^2}{\sqrt{T}}, \tag{43}$$

where inequality (a) follows from the boundedness of $r_t^k$ and Lemma G.1, and the last inequality follows from the choice of stepsize $\gamma_t$.

It remains to bound $\frac{1}{T - \tau_T}\sum_{t=\tau_T}^{T-1}\mathbb{E}[y_t^k(r_t^k - J^k)]$. Here, the expectation is taken with respect to the Markovian sampling of the state-action trajectories. We will leverage the uniform ergodicity assumed in Assumption 2 to relate this quantity to the stationary distribution. In the remaining proof, we write out the underlying randomness in the expectation. Toward this, let $v_{0:t}^k$ denote the Markovian trajectory at agent $k$:

$$v_{0:t}^k = (s_0^k, a_0^k, s_1^k, a_1^k, ..., s_{t-\tau+1}^k, a_{t-\tau+1}^k, ..., s_t^k, a_t^k, s_{t+1}^k),$$

and $v_{0:t}$ denote the collection of the trajectories:

$$v_{0:t} = \{v_{0:t}^k\}_{k=1}^K.$$

Let $P_{t-\tau:t}^k(\cdot, \cdot)$ denote the probability distribution over state-action pairs $(s, a) \in \mathcal{S} \times \mathcal{A}$ at time $t$ given the past trajectory $v_{0:t-\tau}$.

$$
\begin{aligned}
\mathbb{E}_{v_{0:t}^k}[(\eta_t^k - J^k)(r_t^k - J^k)] &= \mathbb{E}_{v_{0:t}^k}[(\eta_t^k - \eta_{t-\tau}^k)(r_t^k - J^k)] + \mathbb{E}_{v_{0:t}^k}[(\eta_{t-\tau}^k - J^k)(r_t^k - J^k)] \\
&\le 2U_r\mathbb{E}_{v_{0:t}^k}[|\eta_t^k - \eta_{t-\tau}^k|] + \mathbb{E}_{v_{0:t-\tau}^k}[\mathbb{E}_{v_{t-\tau+1:t}^k}[(\eta_{t-\tau}^k - J^k)(r_t^k - J^k) \mid v_{0:t-\tau}^k]]
\end{aligned}
$$

By the update of $\eta^k$ in Algorithm 1 and Lemma G.1, we know that

$$2U_r\mathbb{E}_{v_{0:t}^k}[|\eta_t^k - \eta_{t-\tau}^k|] \le \frac{c_\gamma}{\sqrt{T}}\tau 4U_r^2.$$

In addition, we know under the steady-state distribution $\mu^k$,

$$\mathbb{E}_{\mu^k \otimes \pi}[(\eta_{t-\tau}^k - J^k)(r_t^k - J^k(\theta_{t-\tau}^k)) | v_{0:t-\tau}] = 0.$$

Thus

$$\mathbb{E}_{v_{0:t}^k}[(\eta_{t-\tau}^k - J^k)(r_t^k - J^k)]n - 0$$
$$=\mathbb{E}_{v_{0:t-\tau}^k}[\mathbb{E}_{v_{t-\tau+1:t}^k}[(\eta_{t-\tau}^k - J^k(\theta_{t-\tau}^k))(r_t^k - J^k(\theta_{t-\tau}^k))|v_{0:t-\tau}]$$
$$- \mathbb{E}_{\mu^k \otimes \pi}[(\eta_{t-\tau}^k - J^k(\theta_{t-\tau}^k))(r_t^k - J^k(\theta_{t-\tau}^k))|v_{0:t-\tau}]]$$
$$\leq 4U_r^2 d_{TV}(P_{t-\tau:t}^k(\cdot,\cdot), \mu^k \otimes \pi)$$
$$\leq 4U_r^2 m\rho^{\tau-1},$$

where the last inequality follows from Assumption 2. Thus,

$$\mathbb{E}_{v_{0:t}^k}\left[(\eta_t^k - J^k)(r_t^k - J^k)\right] \leq \frac{c_\gamma}{\sqrt{T}}\tau 4U_r^2 + 4U_r^2 m\rho^{\tau-1}. \tag{44}$$

Plugging the bound in Eq. (44) with $\tau = \tau_T$ back to Eq. (43), we get

$$\frac{1}{T-\tau_T}\sum_{t=\tau_T}^{T-1}\mathbb{E}|y_t^k|^2 \leq \frac{\sqrt{T}}{T-\tau_T}\frac{2U_r^2}{c_\gamma} + \frac{c_\gamma}{\sqrt{T}}\tau_T 4U_r^2 + 4U_r^2 m\rho^{\tau_T-1} + \frac{2c_\gamma U_r^2}{\sqrt{T}}$$

proving the lemma.

## F    PROOF OF THEOREM 1

From Lemma 5.3, Lemma 5.4, and Lemma 5.5, we have,

$$Y_T^k \leq \frac{\sqrt{T}}{T - \tau_T} \frac{2U_r^2}{c_\gamma} + \frac{c_\gamma \tau_T 4U_r^2 + 4U_r^2}{\sqrt{T}} + \frac{2c_\gamma U_r^2}{\sqrt{T}},$$

$$M_T \leq \frac{\sqrt{T}}{T - \tau_T} \frac{r^2}{\lambda \alpha c_\zeta} + \frac{2rc_\zeta U_\delta^2 U_\omega^2}{\sqrt{T}\lambda K \alpha} + \frac{4rC_1}{\lambda \alpha \sqrt{T}} + \frac{4rU_\omega}{\lambda K \alpha} \sum_{k=1}^{K} \sqrt{M_T Y_T^k},$$

$$X_T^k \leq \mathcal{O}(\frac{\log(T)}{\sqrt{T}}) + 16U_\omega \sqrt{X_T^k M_T} + \frac{1}{\lambda} \sqrt{X_T^k Y_T^k}$$

$$+ \frac{2c_\zeta U_\omega^2}{c_\beta \lambda} \frac{1}{K} \sum_{i=1}^{K} \sqrt{X_T^k X_T^i} + \frac{c_\zeta U_\omega^2}{c_\beta \lambda K} \sum_{i=1}^{K} \sqrt{X_T^k Y_T^i} + \mathcal{O}(1),$$

where $C_1 = \left[ \left(8U_\omega^2 + 4U_\omega U_r\right) 10 c_\zeta \tau_T U_\delta U_\omega + (8U_\omega + 4U_r) c_\beta \tau_T U_\delta U_{\mathbf{B}} + (4U_r + 4U_\omega) U_\omega \right]$.

By the definition of $\tau_T$, and $T \geq 2\tau_T$, we conclude

$$Y_T^k = \mathcal{O}(\frac{\log(T)}{\sqrt{T}}),$$

$$M_T \leq \mathcal{O}(\frac{r^2}{\sqrt{T}}) + \mathcal{O}(\frac{r}{K\sqrt{T}}) + \mathcal{O}(\frac{r\log(T)}{\sqrt{T}}) + \frac{4rU_\omega}{\lambda K \alpha} \sum_{k=1}^{K} \sqrt{M_T Y_T^k},$$

$$X_T^k \leq \mathcal{O}(\frac{\log(T)}{\sqrt{T}}) + 16U_\omega \sqrt{X_T^k M_T} + \frac{1}{\lambda} \sqrt{X_T^k Y_T^k}$$

$$+ \frac{2c_\zeta U_\omega^2}{c_\beta \lambda} \frac{1}{K} \sum_{i=1}^{K} \sqrt{X_T^k X_T^i} + \frac{c_\zeta U_\omega^2}{c_\beta \lambda K} \sum_{i=1}^{K} \sqrt{X_T^k Y_T^i} + \mathcal{O}(1).$$

Then, for $M_T$, we have

$$M_T \leq \mathcal{O}(\frac{r^2}{\sqrt{T}}) + \mathcal{O}(\frac{r\log(T)}{\sqrt{T}}) + \frac{1}{K} \sum_{k=1}^{K} \sqrt{M_T Y_T^k \left(\frac{4rU_\omega}{\lambda \alpha}\right)^2}$$

$$\leq \mathcal{O}(\frac{r^2}{\sqrt{T}}) + \mathcal{O}(\frac{r\log(T)}{\sqrt{T}}) + \frac{1}{K} \sum_{k=1}^{K} \frac{M_T}{2} + \frac{1}{K} \sum_{k=1}^{K} \frac{Y_T^k}{2} \left(\frac{4rU_\omega}{\lambda \alpha}\right)^2$$

$$= \mathcal{O}(\frac{r^2}{\sqrt{T}}) + \mathcal{O}(\frac{r\log(T)}{\sqrt{T}}) + \frac{M_T}{2} + \frac{1}{K} \sum_{k=1}^{K} \frac{Y_T^k}{2} \left(\frac{4rU_\omega}{\lambda \alpha}\right)^2.$$

We use Young's inequality for the second inequality to hold.

Then, by moving $\frac{M_T}{2}$ to the left and then multiply 2 for both sides, we have,

$$M_T \leq \mathcal{O}(\frac{r^2}{\sqrt{T}}) + \mathcal{O}(\frac{r\log(T)}{\sqrt{T}}) + \mathcal{O}(\frac{\log(T)}{\sqrt{T}})$$

$$= \mathcal{O}(\frac{r\log(T)}{\sqrt{T}}) \text{ when } r \leq \log(T).$$

Finally, for $\bar{X}_T$, we have

$$\bar{X}_T = \frac{1}{K} \sum_{k=1}^{K} X_T^k \leq 16U_\omega \frac{1}{K} \sum_{k=1}^{K} \sqrt{X_T^k M_T} + \frac{1}{\lambda} \frac{1}{K} \sum_{k=1}^{K} \sqrt{X_T^k Y_T^k}$$

$$+ \frac{2c_\zeta U_\omega^2}{c_\beta \lambda} \frac{1}{K} \sum_{k=1}^{K} \sqrt{X_T^k} \frac{1}{K} \sum_{i=1}^{K} \sqrt{X_T^i} + \frac{c_\zeta U_\omega^2}{c_\beta \lambda} \frac{1}{K} \sum_{k=1}^{K} \sqrt{X_T^k} \frac{1}{K} \sum_{i=1}^{K} \sqrt{Y_T^i} + \mathcal{O}(1). \quad (45)$$

By Cauchy Schwartz, we have

$$\frac{1}{K}\sum_{k=1}^{K}\sqrt{X_T^k} \leq \frac{1}{K}\sqrt{\sum_{k=1}^{K}X_T^k}\sqrt{K} = \sqrt{\frac{1}{K}\sum_{k=1}^{K}X_T^k} = \sqrt{\bar{X}_T}.$$

Then, equation 45 can be written as:

$$\bar{X}_T \leq \frac{1}{K}\sum_{k=1}^{K}\sqrt{X_T^k M_T(16U_\omega)^2} + \frac{1}{K}\sum_{k=1}^{K}\sqrt{X_T^k \frac{Y_T^k}{\lambda^2}}$$

$$+ \frac{2c_\zeta U_\omega^2}{c_\beta \lambda}\bar{X}_T + \frac{c_\zeta U_\omega^2}{c_\beta \lambda}\frac{1}{K}\sum_{i=1}^{K}\sqrt{\bar{X}_T Y_T^i} + \mathcal{O}(1)$$

$$\leq \frac{1}{K}\sum_{k=1}^{K}\frac{X_T^k}{2} + \frac{M_T(16U_\omega)^2}{2} + \frac{1}{K}\sum_{k=1}^{K}\frac{X_T^k}{4} + \frac{1}{K}\sum_{k=1}^{K}\frac{Y_T^k}{\lambda^2}$$

$$+ \frac{2c_\zeta U_\omega^2}{c_\beta \lambda}\bar{X}_T + \frac{c_\zeta U_\omega^2}{c_\beta \lambda}(\frac{\bar{X}_T}{2} + \frac{1}{K}\sum_{i=1}^{K}\frac{Y_T^i}{2}) + \mathcal{O}(1).$$

We can choose $\frac{c_\zeta}{c_\beta} \leq \frac{\lambda}{20U_\omega^2}$ such that $\frac{2c_\zeta U_\omega^2}{c_\beta \lambda} + \frac{c_\zeta U_\omega^2}{2c_\beta \lambda} \leq \frac{1}{8}$, then,

$$\bar{X}_T \leq \frac{7}{8}\bar{X}_T + \mathcal{O}(\frac{r\log(T)}{\sqrt{T}}) + \mathcal{O}(\frac{\log(T)}{\sqrt{T}}) + \mathcal{O}(1) \tag{46}$$

$$\Rightarrow \bar{X}_T \leq \mathcal{O}(1) \text{ by rearrangement.}$$

# G    SUPPORTING PROPOSITIONS AND LEMMAS

## G.1    BOUNDEDNESS

**Lemma G.1.** *For any $t \geq 0, k \in [K]$, it holds that $\eta_t^k \in [-U_r, U_r]$ – recalling that $U_r$ is the bound on the reward.*

*Proof.* We prove this by induction. The algorithm initializes $\eta_0^k = 0$;

Assume $\eta_i^k \in [-U_r, U_r]$ for $0 \leq i \leq t - 1$. Then,

$$\eta_t^k = \eta_{t-1}^k + \gamma_t(r_{t-1}^k - \eta_{t-1}^k)$$
$$= (1 - \gamma_t)\eta_{t-1}^k + \gamma_t r_{t-1}^k$$
$$\leq (1 - \gamma_t)U_r + \gamma_t U_r$$
$$= U_r.$$

Similarly, it can be shown that $\eta_t^k \geq -U_r$.

$\square$

**Lemma G.2.** *Suppose that Assumption 3 holds. For any $t \geq 0, k \in [K]$, it holds that $|\delta_t^k| \leq U_\delta$, where $U_\delta := 2U_r + 2U_\omega$.*

*Proof.*

$$|\delta_t^k| = |r_t^k - \eta_t^k + \omega_t^{k\top}\left(\mathbf{B}_t^k\right)^\top \phi(s_{t+1}^k) - \omega_t^{k\top}\left(\mathbf{B}_t^k\right)^\top \phi(s_t^k)|$$
$$\leq |r_t^k - \eta_t^k| + |\omega_t^{k\top}\left(\mathbf{B}_t^k\right)^\top \phi(s_{t+1}^k) - \omega_t^{k\top}\left(\mathbf{B}_t^k\right)^\top \phi(s_t^k)|$$
$$\leq 2U_r + 2U_\omega.$$

$\square$

**Lemma G.3.** *For any $j \geq t - \tau > 0, k \in [K]$, where $\beta_i$ and $\zeta_i$ are non-increasing for $i \geq 0$.*

$$\|\omega_j^k - \omega_{t-\tau}^k\| \leq \beta_{t-\tau}(j - t + \tau)U_\delta U_{\mathbf{B}},$$

$$\|\mathbf{B}_j^k - \mathbf{B}_{t-\tau}^k\| \leq 10\zeta_{t-\tau}(j - t + \tau)U_\delta U_\omega.$$

*Proof.*

$$\|\omega_j^k - \omega_{t-\tau}^k\| \leq \sum_{i=t-\tau}^{j-1} \|\omega_{i+1}^k - \omega_i^k\|$$
$$= \sum_{i=t-\tau}^{j-1} \|\Pi_{U_\omega}(\omega_i^k + \beta_i \delta_i^k(\mathbf{B}_i)^\top \phi(s_i^k)) - \Pi_{U_\omega}(\omega_i^k)\|$$
$$\leq \sum_{i=t-\tau}^{j-1} \|\beta_i \delta_i^k(\mathbf{B}_i)^\top \phi(s_i^k)\|$$
$$\leq \beta_{t-\tau}(j - t + \tau)U_\delta.$$

Moreover,

$$\|\mathbf{B}_j - \mathbf{B}_{t-\tau}\| \leq \sum_{i=t-\tau}^{j-1} \|\mathbf{B}_{i+1} - \mathbf{B}_i\|$$

$$= \sum_{i=t-\tau}^{j-1} \|(\mathbf{B}_i + \zeta_i \delta_i^k \phi(s_i^k)(\omega_i^k)^\top)\mathbf{R}_{i+1}^{-1} - \mathbf{B}_i \mathbf{R}_{i+1}^{-1} + \mathbf{B}_i \mathbf{R}_{i+1}^{-1} - \mathbf{B}_i\|$$

$$\leq \sum_{i=t-\tau}^{j-1} \|\zeta_i \delta_i^k \phi(s_i^k)(\omega_i^k)^\top \mathbf{R}_{i+1}^{-1}\| + \|\mathbf{B}_i \mathbf{R}_{i+1}^{-1} - \mathbf{B}_i\|$$

$$\leq \sum_{i=t-\tau}^{j-1} \|\zeta_i \delta_i^k \phi(s_i^k)(\omega_i^k)^\top\| \|\mathbf{R}_{i+1}^{-1}\| + \|\mathbf{R}_{i+1}^{-1} - \mathbf{I}\|$$

$$\leq 2\zeta_{t-\tau}(j - t + \tau)U_\delta U_\omega + (j - t + \tau)8U_\delta U_\omega \zeta_{t-\tau}$$

$$= 10\zeta_{t-\tau}(j - t + \tau)U_\delta U_\omega.$$

The last inequality follows from equation 18 and $\|\mathbf{R}_{i+1}^{-1}\| \leq 2$. $\qquad\square$

### G.2 MARKOVIAN NOISES

In this section, we are dealing with Markovian noises. We introduce some useful notations for the proofs.

First recall that

$$\xi_t^k = (r_t^k - J^k + (\phi(s_{t+1}^k) - \phi(s_t^k))^\top \mathbf{B}_t \omega_t^k)\phi(s_t^k)$$
$$- \mathbb{E}_{\mu^k, \pi, P^k}[(r_t^k - J^k + (\phi(s_{t+1}^k) - \phi(s_t^k))^\top \mathbf{B}_t \omega_t^k)\phi(s_t^k)].$$

We further introduce

$$g^k(O, \mathbf{B}, \omega) = (R(s,a) - J^k + (\phi(s') - \phi(s))^\top \mathbf{B}\omega)\phi(s)$$
$$\bar{g}^k(\mathbf{B}, \omega) = \mathbb{E}_{\mu^k, \pi, P^k}\left[(R(s,a) - J^k + (\phi(s') - \phi(s))^\top \mathbf{B}\omega)\phi(s)\right],$$

where $O$ is the sample we get from the Markov chain generated by the algorithm. For example, $O_t^k = (s_t^k, a_t^k, s_{t+1}^k)$. Then,

$$\xi_t^k = g^k(O_t^k, \mathbf{B}_t, \omega_t^k) - \bar{g}^k(\mathbf{B}_t, \omega_t^k).$$

Furthermore, we define

$$\Phi_1(O_t^k, \mathbf{B}_t, \omega_t^k) = \mathbb{E}\langle m_t, \mathbf{B}_\perp^{*\top} \xi_t^k(\omega_t^k)^\top \rangle,$$
$$\Phi_2(O_t^k, \mathbf{B}_t, \omega_t^k) = \mathbb{E}\langle x_t^k, \mathbf{P}_t \xi_t^k \rangle,$$
$$\Phi_3(O_t^k, \mathbf{B}_t, \omega_t^k, \omega_t^i) = \mathbb{E}\langle x_t^k, \xi_t^i(\omega_t^i)^\top \omega_t^k \rangle.$$

**Lemma G.4.** *For any* $t \geq \tau > 0, \forall k \in [K]$, *we have*

$$\mathbb{E}\langle m_t, \mathbf{B}_\perp^{*\top} \xi_t^k(\omega_t^k)^\top \rangle$$
$$\leq \left(8U_\omega^2 + 4U_\omega U_r\right)\|\mathbf{B}_t - \mathbf{B}_{t-\tau}\| + (8U_\omega + 4U_r)\|\omega_t^k - \omega_{t-\tau}^k\| + (4U_r + 4U_\omega)U_\omega m\rho^{\tau-1}.$$

**Proof of Lemma G.4**

*Proof.* We prove this Lemma in three steps, the first step is to show $\Phi_1(O, \mathbf{B}, \omega)$ is Lipschitz w.r.t. $\mathbf{B}$, the second step is to show the target term is Lipschitz w.r.t. $\omega$, and the third step is to use the uniform convergence of the Markov chain to show a decaying total variation distance.

Specifically, We decompose the target term into three subtraction terms:

$$\mathbb{E}\langle m_t, \mathbf{B}_\perp^{*\top} \xi_t^k (\omega_t^k)^\top \rangle$$
$$=\Phi_1(O_t^k, \mathbf{B}_t, \omega_t^k)$$
$$=\Phi_1(O_t^k, \mathbf{B}_t, \omega_t^k) - \Phi_1(O_t^k, \mathbf{B}_{t-\tau}, \omega_t^k)$$
$$+ \Phi_1(O_t^k, \mathbf{B}_{t-\tau}, \omega_t^k) - \Phi_1(O_t^k, \mathbf{B}_{t-\tau}, \omega_{t-\tau}^k)$$
$$+ \Phi_1(O_t^k, \mathbf{B}_{t-\tau}, \omega_{t-\tau}^k).$$

Step 1: for this step, we show that

$$\Phi_1(O_t^k, \mathbf{B}_t, \omega_t^k) - \Phi_1(O_t^k, \mathbf{B}_{t-\tau}, \omega_t^k) \leq \left(8U_\omega^2 + 4U_\omega U_r\right) \|\mathbf{B}_t - \mathbf{B}_{t-\tau}\|.$$

By the definition of $\Phi_1(O_t^k, \mathbf{B}_t, \omega_t^k)$, we have,

$$\Phi_1(O_t^k, \mathbf{B}_t, \omega_t^k) - \Phi_1(O_t^k, \mathbf{B}_{t-\tau}, \omega_t^k)$$
$$=\mathbb{E}\langle \mathbf{B}_\perp^{*\top}\mathbf{B}_t, \mathbf{B}_\perp^{*\top} \left( g^k(O_t^k, \mathbf{B}_t, \omega_t^k) - \bar{g}^k(\mathbf{B}_t, \omega_t^k)\right)(\omega_t^k)^\top \rangle$$
$$- \mathbb{E}\langle \mathbf{B}_\perp^{*\top}\mathbf{B}_{t-\tau}, \mathbf{B}_\perp^{*\top} \left( g^k(O_t^k, \mathbf{B}_{t-\tau}, \omega_t^k) - \bar{g}^k(\mathbf{B}_{t-\tau}, \omega_t^k)\right)(\omega_t^k)^\top \rangle$$
$$=\mathbb{E}\langle \mathbf{B}_\perp^{*\top}\mathbf{B}_t, \mathbf{B}_\perp^{*\top} \left( g^k(O_t^k, \mathbf{B}_t, \omega_t^k) - \bar{g}^k(\mathbf{B}_t, \omega_t^k)\right)(\omega_t^k)^\top \rangle$$
$$- \mathbb{E}\langle \mathbf{B}_\perp^{*\top}\mathbf{B}_{t-\tau}, \mathbf{B}_\perp^{*\top} \left( g^k(O_t^k, \mathbf{B}_t, \omega_t^k) - \bar{g}^k(\mathbf{B}_t, \omega_t^k)\right)(\omega_t^k)^\top \rangle$$
$$+ \mathbb{E}\langle \mathbf{B}_\perp^{*\top}\mathbf{B}_{t-\tau}, \mathbf{B}_\perp^{*\top} \left( g^k(O_t^k, \mathbf{B}_t, \omega_t^k) - \bar{g}^k(\mathbf{B}_t, \omega_t^k)\right)(\omega_t^k)^\top \rangle$$
$$- \mathbb{E}\langle \mathbf{B}_\perp^{*\top}\mathbf{B}_{t-\tau}, \mathbf{B}_\perp^{*\top} \left( g^k(O_t^k, \mathbf{B}_{t-\tau}, \omega_t^k) - \bar{g}^k(\mathbf{B}_{t-\tau}, \omega_t^k)\right)(\omega_t^k)^\top \rangle$$
$$\leq\mathbb{E}\|\mathbf{B}_\perp^{*\top}(\mathbf{B}_t - \mathbf{B}_{t-\tau})\|\|\mathbf{B}_\perp^{*\top} \left( g^k(O_t^k, \mathbf{B}_t, \omega_t^k) - \bar{g}^k(\mathbf{B}_t, \omega_t^k)\right)(\omega_t^k)^\top \|$$
$$+ \mathbb{E}\langle \mathbf{B}_\perp^{*\top}\mathbf{B}_{t-\tau}, \mathbf{B}_\perp^{*\top} \left( g^k(O_t^k, \mathbf{B}_t, \omega_t^k) - g^k(O_t^k, \mathbf{B}_{t-\tau}, \omega_t^k)\right)(\omega_t^k)^\top \rangle$$
$$- \mathbb{E}\langle \mathbf{B}_\perp^{*\top}\mathbf{B}_{t-\tau}, \mathbf{B}_\perp^{*\top} \left( \bar{g}^k(\mathbf{B}_t, \omega_t^k) - \bar{g}^k(\mathbf{B}_{t-\tau}, \omega_t^k)\right)(\omega_t^k)^\top \rangle$$
$$\leq\mathbb{E}\|\mathbf{B}_\perp^{*\top}(\mathbf{B}_t - \mathbf{B}_{t-\tau})\|\|\mathbf{B}_\perp^{*\top} \left( g^k(O_t^k, \mathbf{B}_t, \omega_t^k) - \bar{g}^k(\mathbf{B}_t, \omega_t^k)\right)(\omega_t^k)^\top \|$$
$$+ \mathbb{E}\|\mathbf{B}_\perp^{*\top}\mathbf{B}_{t-\tau}\|\|\mathbf{B}_\perp^{*\top}(\phi(s_{t+1}^k) - \phi(s_t^k))^\top (\mathbf{B}_t - \mathbf{B}_{t-\tau})\omega_t^k \phi(s_t^k)(\omega_t^k)^\top \|$$
$$+ \mathbb{E}\|\mathbf{B}_\perp^{*\top}\mathbf{B}_{t-\tau}\|\|\mathbf{B}_\perp^{*\top}\mathbb{E}_{\mu^k, \pi_\theta, P^k}\left[(\phi(s_{t+1}^k) - \phi(s_t^k))^\top(\mathbf{B}_t - \mathbf{B}_{t-\tau})\omega_t^k\right]\phi(s_t^k)(\omega_t^k)^\top \|$$
$$\leq \left(8U_\omega^2 + 4U_\omega U_r\right) \|\mathbf{B}_t - \mathbf{B}_{t-\tau}\|.$$

Step 2: We show that

$$\Phi_1(O_t^k, \mathbf{B}_{t-\tau}, \omega_t^k) - \Phi_1(O_t^k, \mathbf{B}_{t-\tau}, \omega_{t-\tau}^k) \leq (8U_\omega + 4U_r)\|\omega_t^k - \omega_{t-\tau}^k\|.$$

By definition, we have

$$\Phi_1(O_t^k, \mathbf{B}_{t-\tau}, \omega_t^k) - \Phi_1(O_t^k, \mathbf{B}_{t-\tau}, \omega_{t-\tau}^k)$$
$$=\mathbb{E}\langle \mathbf{B}_\perp^{*\top}\mathbf{B}_{t-\tau}, \mathbf{B}_\perp^{*\top} \left( g^k(O_t^k, \mathbf{B}_{t-\tau}, \omega_t^k) - \bar{g}^k(\mathbf{B}_{t-\tau}, \omega_t^k)\right)(\omega_t^k)^\top \rangle$$
$$- \mathbb{E}\langle \mathbf{B}_\perp^{*\top}\mathbf{B}_{t-\tau}, \mathbf{B}_\perp^{*\top} \left( g^k(O_t^k, \mathbf{B}_{t-\tau}, \omega_{t-\tau}^k) - \bar{g}^k(\mathbf{B}_{t-\tau}, \omega_{t-\tau}^k)\right)(\omega_{t-\tau}^k)^\top \rangle$$
$$=\mathbb{E}\langle \mathbf{B}_\perp^{*\top}\mathbf{B}_{t-\tau}, \mathbf{B}_\perp^{*\top} \left( g^k(O_t^k, \mathbf{B}_{t-\tau}, \omega_t^k) - \bar{g}^k(\mathbf{B}_{t-\tau}, \omega_t^k)\right)(\omega_t^k)^\top \rangle$$
$$- \mathbb{E}\langle \mathbf{B}_\perp^{*\top}\mathbf{B}_{t-\tau}, \mathbf{B}_\perp^{*\top} \left( g^k(O_t^k, \mathbf{B}_{t-\tau}, \omega_{t-\tau}^k) - \bar{g}^k(\mathbf{B}_{t-\tau}, \omega_{t-\tau}^k)\right)(\omega_t^k)^\top \rangle$$
$$+ \mathbb{E}\langle \mathbf{B}_\perp^{*\top}\mathbf{B}_{t-\tau}, \mathbf{B}_\perp^{*\top} \left( g^k(O_t^k, \mathbf{B}_{t-\tau}, \omega_{t-\tau}^k) - \bar{g}^k(\mathbf{B}_{t-\tau}, \omega_{t-\tau}^k)\right)(\omega_t^k)^\top \rangle$$
$$- \mathbb{E}\langle \mathbf{B}_\perp^{*\top}\mathbf{B}_{t-\tau}, \mathbf{B}_\perp^{*\top} \left( g^k(O_t^k, \mathbf{B}_{t-\tau}, \omega_{t-\tau}^k) - \bar{g}^k(\mathbf{B}_{t-\tau}, \omega_{t-\tau}^k)\right)(\omega_{t-\tau}^k)^\top \rangle$$
$$=\mathbb{E}\langle \mathbf{B}_\perp^{*\top}\mathbf{B}_{t-\tau}, \mathbf{B}_\perp^{*\top} \left( g^k(O_t^k, \mathbf{B}_{t-\tau}, \omega_t^k) - g^k(O_t^k, \mathbf{B}_{t-\tau}, \omega_{t-\tau}^k)\right)(\omega_t^k)^\top \rangle$$
$$- \mathbb{E}\langle \mathbf{B}_\perp^{*\top}\mathbf{B}_{t-\tau}, \mathbf{B}_\perp^{*\top} \left( \bar{g}^k(\mathbf{B}_{t-\tau}, \omega_t^k) - \bar{g}^k(\mathbf{B}_{t-\tau}, \omega_{t-\tau}^k)\right)(\omega_t^k)^\top \rangle$$
$$+ \mathbb{E}\langle \mathbf{B}_\perp^{*\top}\mathbf{B}_{t-\tau}, \mathbf{B}_\perp^{*\top} \left( g^k(O_t^k, \mathbf{B}_{t-\tau}, \omega_{t-\tau}^k) - \bar{g}^k(\mathbf{B}_{t-\tau}, \omega_{t-\tau}^k)\right)(\omega_t^k - \omega_{t-\tau}^k)^\top \rangle$$
$$\leq(8U_\omega + 4U_r)\|\omega_t^k - \omega_{t-\tau}^k\|.$$

Step 3:

We show that conditioning on past trajectory $v_{0:t-\tau}$, the only random part is the sample $O_t^k$ we draw from the Markov chain at current timestep, and the sample we draw from the steady-state distribution $\mu^k \otimes \pi \otimes P^k$. Consequently,

$$\Phi_1(O_t^k, \mathbf{B}_{t-\tau}, \omega_{t-\tau}^k) \leq (4U_r + 4U_\omega)U_\omega m\rho^{\tau-1}.$$

By definition, we have

$$
\begin{aligned}
&\Phi_1(O_t^k, \mathbf{B}_{t-\tau}, \omega_{t-\tau}^k)\\
&=\mathbb{E}\langle \mathbf{B}_\perp^{*\top}\mathbf{B}_{t-\tau}, \mathbf{B}_\perp^{*\top}\left(g^k(O_t^k, \mathbf{B}_{t-\tau}, \omega_{t-\tau}^k) - \bar{g}^k(\mathbf{B}_{t-\tau}, \omega_{t-\tau}^k)\right)(\omega_{t-\tau}^k)^\top \rangle\\
&=\mathbb{E}\left[\mathbb{E}\left[\langle \mathbf{B}_\perp^{*\top}\mathbf{B}_{t-\tau}, \mathbf{B}_\perp^{*\top}\left(g^k(O_t^k, \mathbf{B}_{t-\tau}, \omega_{t-\tau}^k) - \bar{g}^k(\mathbf{B}_{t-\tau}, \omega_{t-\tau}^k)\right)(\omega_{t-\tau}^k)^\top \rangle \mid v_{0:t-\tau}\right]\right]\\
&=\mathbb{E}\left[\langle \mathbf{B}_\perp^{*\top}\mathbf{B}_{t-\tau}, \mathbf{B}_\perp^{*\top}\mathbb{E}\left[\left(g^k(O_t^k, \mathbf{B}_{t-\tau}, \omega_{t-\tau}^k) - \bar{g}^k(\mathbf{B}_{t-\tau}, \omega_{t-\tau}^k)\right) \mid v_{0:t-\tau}\right](\omega_{t-\tau}^k)^\top \rangle\right]\\
&\leq(4U_r + 4U_\omega)U_\omega d_{TV}(P_{t-\tau:t}^k(\cdot, \cdot, \cdot), \mu^k \otimes \pi \otimes P^k)\\
&\leq(4U_r + 4U_\omega)U_\omega m\rho^{\tau-1}.
\end{aligned}
$$

Combining the above steps, we get,

$$
\begin{aligned}
&\mathbb{E}\langle m_t, \mathbf{B}_\perp^{*\top}\xi_t^k(\omega_t^k)^\top \rangle\\
&\leq \left(8U_\omega^2 + 4U_\omega U_r\right)\|\mathbf{B}_t - \mathbf{B}_{t-\tau}\| + (8U_\omega + 4U_r)\|\omega_t^k - \omega_{t-\tau}^k\| + (4U_r + 4U_\omega)U_\omega m\rho^{\tau-1}.
\end{aligned}
$$

$\square$

**Lemma G.5.** *For any $t \geq \tau > 0, \forall k \in [K]$, we have*

$$
\begin{aligned}
&\mathbb{E}\langle x_t^k, \mathbf{P}_t\xi_t^k \rangle\\
&\leq(20U_\omega U_r + 28U_\omega^2)\|\mathbf{B}_t - \mathbf{B}_{t-\tau}\| + (4U_r + 12U_\omega)\|\omega_t^k - \omega_{t-\tau}^k\| + 8U_\omega(U_r + U_\omega)m\rho^{\tau-1}.
\end{aligned}
$$

**Proof of Lemma G.5**

*Proof.* Similarly, we show $\Phi_1(O, \mathbf{B}, \omega)$ is Lipschitz w.r.t. $\mathbf{B}$ and $\omega$ respectively for the first two steps, and utilize uniform convergence property of the Markov chain for the last step.

We first decompose it into three steps:

$$
\begin{aligned}
&\mathbb{E}\langle x_t^k, \mathbf{P}_t\xi_t^k \rangle\\
&=\Phi_2(O_t^k, \mathbf{B}_t, \omega_t^k)\\
&=\Phi_2(O_t^k, \mathbf{B}_t, \omega_t^k) - \Phi_2(O_t^k, \mathbf{B}_{t-\tau}, \omega_t^k)\\
&\quad+ \Phi_2(O_t^k, \mathbf{B}_{t-\tau}, \omega_t^k) - \Phi_2(O_t^k, \mathbf{B}_{t-\tau}, \omega_{t-\tau}^k)\\
&\quad+ \Phi_2(O_t^k, \mathbf{B}_{t-\tau}, \omega_{t-\tau}^k).
\end{aligned}
$$

Step 1:

We show

$$\Phi_2(O_t^k, \mathbf{B}_t, \omega_t^k) - \Phi_2(O_t^k, \mathbf{B}_{t-\tau}, \omega_t^k) \leq (20U_\omega U_r + 28U_\omega^2)\|\mathbf{B}_t - \mathbf{B}_{t-\tau}\|.$$

By definition, we have

$$\Phi_2(O_t^k, \mathbf{B}_t, \omega_t^k) - \Phi_2(O_t^k, \mathbf{B}_{t-\tau}, \omega_t^k)$$

$$= \mathbb{E}\langle \mathbf{B}_t \omega_t^k - z^{k,*}, \mathbf{B}_t \mathbf{B}_t^\top \left( g^k(O_t^k, \mathbf{B}_t, \omega_t^k) - \bar{g}^k(\mathbf{B}_t, \omega_t^k) \right)\rangle$$
$$\quad - \mathbb{E}\langle \mathbf{B}_{t-\tau} \omega_t^k - z^{k,*}, \mathbf{B}_{t-\tau} \mathbf{B}_{t-\tau}^\top \left( g^k(O_t^k, \mathbf{B}_{t-\tau}, \omega_t^k) - \bar{g}^k(\mathbf{B}_{t-\tau}, \omega_t^k) \right)\rangle$$

$$\overset{(a)}{=} \mathbb{E}\langle \mathbf{B}_t \omega_t^k - z^{k,*}, \mathbf{B}_t \mathbf{B}_t^\top \left( g^k(O_t^k, \mathbf{B}_t, \omega_t^k) - \bar{g}^k(\mathbf{B}_t, \omega_t^k) \right)\rangle$$
$$\quad - \mathbb{E}\langle \mathbf{B}_t \omega_t^k - z^{k,*}, \mathbf{B}_t \mathbf{B}_t^\top \left( g^k(O_t^k, \mathbf{B}_{t-\tau}, \omega_t^k) - \bar{g}^k(\mathbf{B}_{t-\tau}, \omega_t^k) \right)\rangle$$
$$\quad + \mathbb{E}\langle \mathbf{B}_t \omega_t^k - z^{k,*}, \mathbf{B}_t \mathbf{B}_t^\top \left( g^k(O_t^k, \mathbf{B}_{t-\tau}, \omega_t^k) - \bar{g}^k(\mathbf{B}_{t-\tau}, \omega_t^k) \right)\rangle$$
$$\quad - \mathbb{E}\langle \mathbf{B}_t \omega_t^k - z^{k,*}, \mathbf{B}_t \mathbf{B}_{t-\tau}^\top \left( g^k(O_t^k, \mathbf{B}_{t-\tau}, \omega_t^k) - \bar{g}^k(\mathbf{B}_{t-\tau}, \omega_t^k) \right)\rangle$$
$$\quad + \mathbb{E}\langle \mathbf{B}_t \omega_t^k - z^{k,*}, \mathbf{B}_t \mathbf{B}_{t-\tau}^\top \left( g^k(O_t^k, \mathbf{B}_{t-\tau}, \omega_t^k) - \bar{g}^k(\mathbf{B}_{t-\tau}, \omega_t^k) \right)\rangle$$
$$\quad - \mathbb{E}\langle \mathbf{B}_t \omega_t^k - z^{k,*}, \mathbf{B}_{t-\tau} \mathbf{B}_{t-\tau}^\top \left( g^k(O_t^k, \mathbf{B}_{t-\tau}, \omega_t^k) - \bar{g}^k(\mathbf{B}_{t-\tau}, \omega_t^k) \right)\rangle$$
$$\quad + \mathbb{E}\langle \mathbf{B}_t \omega_t^k - z^{k,*}, \mathbf{B}_{t-\tau} \mathbf{B}_{t-\tau}^\top \left( g^k(O_t^k, \mathbf{B}_{t-\tau}, \omega_t^k) - \bar{g}^k(\mathbf{B}_{t-\tau}, \omega_t^k) \right)\rangle$$
$$\quad - \mathbb{E}\langle \mathbf{B}_{t-\tau} \omega_t^k - z^{k,*}, \mathbf{B}_{t-\tau} \mathbf{B}_{t-\tau}^\top \left( g^k(O_t^k, \mathbf{B}_{t-\tau}, \omega_t^k) - \bar{g}^k(\mathbf{B}_{t-\tau}, \omega_t^k) \right)\rangle$$

$$= \mathbb{E}\langle \mathbf{B}_t \omega_t^k - z^{k,*}, \mathbf{B}_t \mathbf{B}_t^\top \left( g^k(O_t^k, \mathbf{B}_t, \omega_t^k) - g^k(O_t^k, \mathbf{B}_{t-\tau}, \omega_t^k) \right)\rangle$$
$$\quad - \mathbb{E}\langle \mathbf{B}_t \omega_t^k - z^{k,*}, \mathbf{B}_t \mathbf{B}_t^\top \left( \bar{g}^k(\mathbf{B}_t, \omega_t^k) - \bar{g}^k(\mathbf{B}_{t-\tau}, \omega_t^k) \right)\rangle$$
$$\quad + \mathbb{E}\langle \mathbf{B}_t \omega_t^k - z^{k,*}, \mathbf{B}_t (\mathbf{B}_t - \mathbf{B}_{t-\tau})^\top \left( g^k(O_t^k, \mathbf{B}_{t-\tau}, \omega_t^k) - \bar{g}^k(\mathbf{B}_{t-\tau}, \omega_t^k) \right)\rangle$$
$$\quad + \mathbb{E}\langle \mathbf{B}_t \omega_t^k - z^{k,*}, (\mathbf{B}_t - \mathbf{B}_{t-\tau}) \mathbf{B}_{t-\tau}^\top \left( g^k(O_t^k, \mathbf{B}_{t-\tau}, \omega_t^k) - \bar{g}^k(\mathbf{B}_{t-\tau}, \omega_t^k) \right)\rangle$$
$$\quad + \mathbb{E}\langle (\mathbf{B}_t - \mathbf{B}_{t-\tau}) \omega_t^k, \mathbf{B}_{t-\tau} \mathbf{B}_{t-\tau}^\top \left( g^k(O_t^k, \mathbf{B}_{t-\tau}, \omega_t^k) - \bar{g}^k(\mathbf{B}_{t-\tau}, \omega_t^k) \right)\rangle$$

$$\overset{(b)}{\leq} 20 U_\omega (U_r + U_\omega) \|\mathbf{B}_t - \mathbf{B}_{t-\tau}\| + 8 U_\omega^2 \|\mathbf{B}_t - \mathbf{B}_{t-\tau}\|$$
$$= (20 U_\omega U_r + 28 U_\omega^2) \|\mathbf{B}_t - \mathbf{B}_{t-\tau}\|.$$

For (a), we add and subtract the term that replace each occurrence of $\mathbf{B}_t$ by $\mathbf{B}_{t-\tau}$ sequentially, and there are three such addition and subtractions in total since there are three ocurrence of $\mathbf{B}_t$ in $\Phi_2(O_t^k, \mathbf{B}_t, \omega_t^k)$. For (b), we use $\left( g^k(O_t^k, \mathbf{B}_t, \omega_t^k) - g^k(O_t^k, \mathbf{B}_{t-\tau}, \omega_t^k) \right) \leq 2 U_\omega \|\mathbf{B}_t - \mathbf{B}_{t-\tau}\|$, and $\left( \bar{g}^k(\mathbf{B}_t, \omega_t^k) - \bar{g}^k(\mathbf{B}_{t-\tau}, \omega_t^k) \right) \leq 2 U_\omega \|\mathbf{B}_t - \mathbf{B}_{t-\tau}\|$.

Step 2:

We show that

$$\Phi_2(O_t^k, \mathbf{B}_{t-\tau}, \omega_t^k) - \Phi_2(O_t^k, \mathbf{B}_{t-\tau}, \omega_{t-\tau}^k) \leq (4 U_r + 12 U_\omega) \|\omega_t^k - \omega_{t-\tau}^k\|.$$

By definition, we have

$$\Phi_2(O_t^k, \mathbf{B}_{t-\tau}, \omega_t^k) - \Phi_2(O_t^k, \mathbf{B}_{t-\tau}, \omega_{t-\tau}^k)$$

$$= \mathbb{E}\langle \mathbf{B}_{t-\tau} \omega_t^k - z^{k,*}, \mathbf{B}_{t-\tau} \mathbf{B}_{t-\tau}^\top \left( g^k(O_t^k, \mathbf{B}_{t-\tau}, \omega_t^k) - \bar{g}^k(\mathbf{B}_{t-\tau}, \omega_t^k) \right)\rangle$$
$$\quad - \mathbb{E}\langle \mathbf{B}_{t-\tau} \omega_{t-\tau}^k - z^{k,*}, \mathbf{B}_{t-\tau} \mathbf{B}_{t-\tau}^\top \left( g^k(O_t^k, \mathbf{B}_{t-\tau}, \omega_{t-\tau}^k) - \bar{g}^k(\mathbf{B}_{t-\tau}, \omega_{t-\tau}^k) \right)\rangle$$

$$= \mathbb{E}\langle \mathbf{B}_{t-\tau} \omega_t^k - z^{k,*}, \mathbf{B}_{t-\tau} \mathbf{B}_{t-\tau}^\top \left( g^k(O_t^k, \mathbf{B}_{t-\tau}, \omega_t^k) - \bar{g}^k(\mathbf{B}_{t-\tau}, \omega_t^k) \right)\rangle$$
$$\quad - \mathbb{E}\langle \mathbf{B}_{t-\tau} \omega_t^k - z^{k,*}, \mathbf{B}_{t-\tau} \mathbf{B}_{t-\tau}^\top \left( g^k(O_t^k, \mathbf{B}_{t-\tau}, \omega_{t-\tau}^k) - \bar{g}^k(\mathbf{B}_{t-\tau}, \omega_{t-\tau}^k) \right)\rangle$$
$$\quad + \mathbb{E}\langle \mathbf{B}_{t-\tau} \omega_t^k - z^{k,*}, \mathbf{B}_{t-\tau} \mathbf{B}_{t-\tau}^\top \left( g^k(O_t^k, \mathbf{B}_{t-\tau}, \omega_{t-\tau}^k) - \bar{g}^k(\mathbf{B}_{t-\tau}, \omega_{t-\tau}^k) \right)\rangle$$
$$\quad - \mathbb{E}\langle \mathbf{B}_{t-\tau} \omega_{t-\tau}^k - z^{k,*}, \mathbf{B}_{t-\tau} \mathbf{B}_{t-\tau}^\top \left( g^k(O_t^k, \mathbf{B}_{t-\tau}, \omega_{t-\tau}^k) - \bar{g}^k(\mathbf{B}_{t-\tau}, \omega_{t-\tau}^k) \right)\rangle$$

$$= \mathbb{E}\langle \mathbf{B}_{t-\tau} \omega_t^k - z^{k,*}, \mathbf{B}_{t-\tau} \mathbf{B}_{t-\tau}^\top \left( g^k(O_t^k, \mathbf{B}_{t-\tau}, \omega_t^k) - g^k(O_t^k, \mathbf{B}_{t-\tau}, \omega_{t-\tau}^k) \right)\rangle$$
$$\quad - \mathbb{E}\langle \mathbf{B}_{t-\tau} \omega_t^k - z^{k,*}, \mathbf{B}_{t-\tau} \mathbf{B}_{t-\tau}^\top \left( \bar{g}^k(\mathbf{B}_{t-\tau}, \omega_t^k) - \bar{g}^k(\mathbf{B}_{t-\tau}, \omega_{t-\tau}^k) \right)\rangle$$
$$\quad + \mathbb{E}\langle \mathbf{B}_{t-\tau} (\omega_t^k - \omega_{t-\tau}^k), \mathbf{B}_{t-\tau} \mathbf{B}_{t-\tau}^\top \left( g^k(O_t^k, \mathbf{B}_{t-\tau}, \omega_{t-\tau}^k) - \bar{g}^k(\mathbf{B}_{t-\tau}, \omega_{t-\tau}^k) \right)\rangle$$

$$\leq 8 U_\omega \|\omega_t^k - \omega_{t-\tau}^k\| + 4(U_r + U_\omega) \|\omega_t^k - \omega_{t-\tau}^k\|$$
$$= (4 U_r + 12 U_\omega) \|\omega_t^k - \omega_{t-\tau}^k\|.$$

**Step 3:**

Conditioning on the past trajectory $v_{0:t-\tau}$, the only random part is the sample $O_t^k$ we draw from the Markov chain at current timestep, and the sample we draw from the steady-state distribution $\mu^k \otimes \pi \otimes P^k$. By the uniform ergodicity assumption, we can bound the total variation distance:

$$
\mathbb{E}\langle \mathbf{B}_{t-\tau}\omega_{t-\tau}^k - z^{k,*}, \mathbf{B}_{t-\tau}\mathbf{B}_{t-\tau}^\top \left( g^k(O_t^k, \mathbf{B}_{t-\tau}, \omega_{t-\tau}^k) - \bar{g}^k(\mathbf{B}_{t-\tau}, \omega_{t-\tau}^k) \right) \rangle
$$
$$
= \mathbb{E}\left[ \mathbb{E}\left[ \langle \mathbf{B}_{t-\tau}\omega_{t-\tau}^k - z^{k,*}, \mathbf{B}_{t-\tau}\mathbf{B}_{t-\tau}^\top \left( g^k(O_t^k, \mathbf{B}_{t-\tau}, \omega_{t-\tau}^k) - \bar{g}^k(\mathbf{B}_{t-\tau}, \omega_{t-\tau}^k) \right) \rangle \mid v_{0:t} \right] \right]
$$
$$
= \mathbb{E}\left[ \langle \mathbf{B}_{t-\tau}\omega_{t-\tau}^k - z^{k,*}, \mathbf{B}_{t-\tau}\mathbf{B}_{t-\tau}^\top \mathbb{E}\left[ \left( g^k(O_t^k, \mathbf{B}_{t-\tau}, \omega_{t-\tau}^k) - \bar{g}^k(\mathbf{B}_{t-\tau}, \omega_{t-\tau}^k) \right) \rangle \mid v_{0:t} \right] \right]
$$
$$
\leq 8U_\omega(U_r + U_\omega) d_{TV}(P_{t-\tau:t}^k(\cdot, \cdot, \cdot), \mu^k \otimes \pi \otimes P^k)
$$
$$
\leq 8U_\omega(U_r + U_\omega) m\rho^{\tau-1}.
$$

Combining the above, we get,

$$
\mathbb{E}\langle x_t^k, \mathbf{P}_t \xi_t^k \rangle
$$
$$
\leq (20U_\omega U_r + 28U_\omega^2)\|\mathbf{B}_t - \mathbf{B}_{t-\tau}\| + (4U_r + 12U_\omega)\|\omega_t^k - \omega_{t-\tau}^k\| + 8U_\omega(U_r + U_\omega)m\rho^{\tau-1}.
$$

$\square$

**Lemma G.6.** *For any* $t \geq \tau > 0, \forall k \in [K]$, *we have*

$$
\frac{1}{K}\sum_{i=1}^K \mathbb{E}\langle x_t^k, \xi_t^i(\omega_t^i)^\top \omega_t^k \rangle \leq (8U_\omega^4 + 4(U_r + U_\omega)U_\omega^2)\|\mathbf{B}_t - \mathbf{B}_{t-\tau}\|
$$

$$
+ (12U_r U_\omega^2 + 12U_\omega^3)\|\omega_t^k - \omega_{t-\tau}^k\| + \frac{1}{K}\sum_{i=1}^K (16U_\omega^3 + 8U_r U_\omega^3)\|\omega_t^i - \omega_{t-\tau}^i\| + 4U_\omega^3 m\rho^{\tau-1}.
$$

**Proof of Lemma G.6**

*Proof.* We show for any $i, k$, $\Phi_3(O_t^k, \mathbf{B}_t, \omega_t^k, \omega_t^i)$ can be bounded using $\|\mathbf{B}_t - \mathbf{B}_{t-\tau}\|$, $\|\omega_t^k - \omega_{t-\tau}^k\|$, $\|\omega_t^i - \omega_{t-\tau}^i\|$ and the total variation distance between the steady state distribution and the Markov chain given past information.

We first decompose $\Phi_3(O_t^k, \mathbf{B}_t, \omega_t^k, \omega_t^i)$ into four parts:

$$
\Phi_3(O_t^k, \mathbf{B}_t, \omega_t^k, \omega_t^i)
$$
$$
= \Phi_3(O_t^k, \mathbf{B}_t, \omega_t^k, \omega_t^i) - \Phi_3(O_t^k, \mathbf{B}_{t-\tau}, \omega_t^k, \omega_t^i)
$$
$$
+ \Phi_3(O_t^k, \mathbf{B}_{t-\tau}, \omega_t^k, \omega_t^i) - \Phi_3(O_t^k, \mathbf{B}_{t-\tau}, \omega_{t-\tau}^k, \omega_t^i)
$$
$$
+ \Phi_3(O_t^k, \mathbf{B}_{t-\tau}, \omega_{t-\tau}^k, \omega_t^i) - \Phi_3(O_t^k, \mathbf{B}_{t-\tau}, \omega_{t-\tau}^k, \omega_{t-\tau}^i)
$$
$$
+ \Phi_3(O_t^k, \mathbf{B}_{t-\tau}, \omega_{t-\tau}^k, \omega_{t-\tau}^i).
$$

**Step 1:** We will show that

$$
\Phi_3(O_t^k, \mathbf{B}_t, \omega_t^k, \omega_t^i) - \Phi_3(O_t^k, \mathbf{B}_{t-\tau}, \omega_t^k, \omega_t^i) \leq (8U_\omega^4 + 4(U_r + U_\omega)U_\omega^2)\|\mathbf{B}_t - \mathbf{B}_{t-\tau}\|.
$$

By definition, we have

$$
\Phi_3(O_t^k, \mathbf{B}_t, \omega_t^k, \omega_t^i) - \Phi_3(O_t^k, \mathbf{B}_{t-\tau}, \omega_t^k, \omega_t^i)
$$
$$
= \mathbb{E}\langle \mathbf{B}_t\omega_t^k - z^{k,*}, \left( g^i(O_t^i, \mathbf{B}_t, \omega_t^i) - \bar{g}^i(\mathbf{B}_t, \omega_t^i) \right) (\omega_t^i)^\top \omega_t^k \rangle
$$
$$
- \mathbb{E}\langle \mathbf{B}_{t-\tau}\omega_t^k - z^{k,*}, \left( g^i(O_t^i, \mathbf{B}_{t-\tau}, \omega_t^i) - \bar{g}^i(\mathbf{B}_{t-\tau}, \omega_t^i) \right) (\omega_t^i)^\top \omega_t^k \rangle
$$
$$
= \mathbb{E}\langle \mathbf{B}_t\omega_t^k - z^{k,*}, \left( g^i(O_t^i, \mathbf{B}_t, \omega_t^i) - \bar{g}^i(\mathbf{B}_t, \omega_t^i) \right) (\omega_t^i)^\top \omega_t^k \rangle
$$
$$
- \mathbb{E}\langle \mathbf{B}_t\omega_t^k - z^{k,*}, \left( g^i(O_t^i, \mathbf{B}_{t-\tau}, \omega_t^i) - \bar{g}^i(\mathbf{B}_{t-\tau}, \omega_t^i) \right) (\omega_t^i)^\top \omega_t^k \rangle
$$
$$
+ \mathbb{E}\langle \mathbf{B}_t\omega_t^k - z^{k,*}, \left( g^i(O_t^i, \mathbf{B}_{t-\tau}, \omega_t^i) - \bar{g}^i(\mathbf{B}_{t-\tau}, \omega_t^i) \right) (\omega_t^i)^\top \omega_t^k \rangle
$$
$$
- \mathbb{E}\langle \mathbf{B}_{t-\tau}\omega_t^k - z^{k,*}, \left( g^i(O_t^i, \mathbf{B}_{t-\tau}, \omega_t^i) - \bar{g}^i(\mathbf{B}_{t-\tau}, \omega_t^i) \right) (\omega_t^i)^\top \omega_t^k \rangle
$$
$$
\leq (8U_\omega^4 + 4(U_r + U_\omega)U_\omega^2)\|\mathbf{B}_t - \mathbf{B}_{t-\tau}\|.
$$

Step 2:

We show
$$\Phi_3(O_t^k, \mathbf{B}_{t-\tau}, \omega_t^k, \omega_t^i) - \Phi_3(O_t^k, \mathbf{B}_{t-\tau}, \omega_{t-\tau}^k, \omega_t^i) \leq (12U_r U_\omega^2 + 12U_\omega^3)\|\omega_t^k - \omega_{t-\tau}^k\|.$$

By definition, we have,

$$\mathbb{E}\langle \mathbf{B}_{t-\tau}\omega_t^k - z^{k,*}, \left(g^i(O_t^i, \mathbf{B}_{t-\tau}, \omega_t^i) - \bar{g}^i(\mathbf{B}_{t-\tau}, \omega_t^i)\right)(\omega_t^i)^\top \omega_t^k\rangle$$
$$\quad - \mathbb{E}\langle \mathbf{B}_{t-\tau}\omega_{t-\tau}^k - z^{k,*}, \left(g^i(O_t^i, \mathbf{B}_{t-\tau}, \omega_t^i) - \bar{g}^i(\mathbf{B}_{t-\tau}, \omega_t^i)\right)(\omega_t^i)^\top \omega_{t-\tau}^k\rangle$$
$$=\mathbb{E}\langle \mathbf{B}_{t-\tau}\omega_t^k - z^{k,*}, \left(g^i(O_t^i, \mathbf{B}_{t-\tau}, \omega_t^i) - \bar{g}^i(\mathbf{B}_{t-\tau}, \omega_t^i)\right)(\omega_t^i)^\top \omega_t^k\rangle$$
$$\quad - \mathbb{E}\langle \mathbf{B}_{t-\tau}\omega_t^k - z^{k,*}, \left(g^i(O_t^i, \mathbf{B}_{t-\tau}, \omega_t^i) - \bar{g}^i(\mathbf{B}_{t-\tau}, \omega_t^i)\right)(\omega_t^i)^\top \omega_{t-\tau}^k\rangle$$
$$\quad + \mathbb{E}\langle \mathbf{B}_{t-\tau}\omega_t^k - z^{k,*}, \left(g^i(O_t^i, \mathbf{B}_{t-\tau}, \omega_t^i) - \bar{g}^i(\mathbf{B}_{t-\tau}, \omega_t^i)\right)(\omega_t^i)^\top \omega_{t-\tau}^k\rangle$$
$$\quad - \mathbb{E}\langle \mathbf{B}_{t-\tau}\omega_{t-\tau}^k - z^{k,*}, \left(g^i(O_t^i, \mathbf{B}_{t-\tau}, \omega_t^i) - \bar{g}^i(\mathbf{B}_{t-\tau}, \omega_t^i)\right)(\omega_t^i)^\top \omega_{t-\tau}^k\rangle$$
$$\leq (12U_r U_\omega^2 + 12U_\omega^3)\|\omega_t^k - \omega_{t-\tau}^k\|.$$

Step 3:

We show
$$\Phi_3(O_t^k, \mathbf{B}_{t-\tau}, \omega_{t-\tau}^k, \omega_t^i) - \Phi_3(O_t^k, \mathbf{B}_{t-\tau}, \omega_{t-\tau}^k, \omega_{t-\tau}^i) \leq (16U_\omega^3 + 8U_r U_\omega^3)\|\omega_t^i - \omega_{t-\tau}^i\|.$$

By definition, we have

$$\mathbb{E}\langle \mathbf{B}_{t-\tau}\omega_{t-\tau}^k - z^{k,*}, \left(g^i(O_t^i, \mathbf{B}_{t-\tau}, \omega_t^i) - \bar{g}^i(\mathbf{B}_{t-\tau}, \omega_t^i)\right)(\omega_t^i)^\top \omega_{t-\tau}^k\rangle$$
$$\quad - \mathbb{E}\langle \mathbf{B}_{t-\tau}\omega_{t-\tau}^k - z^{k,*}, \left(g^i(O_t^i, \mathbf{B}_{t-\tau}, \omega_{t-\tau}^i) - \bar{g}^i(\mathbf{B}_{t-\tau}, \omega_{t-\tau}^i)\right)(\omega_{t-\tau}^i)^\top \omega_{t-\tau}^k\rangle$$
$$=\mathbb{E}\langle \mathbf{B}_{t-\tau}\omega_{t-\tau}^k - z^{k,*}, \left(g^i(O_t^i, \mathbf{B}_{t-\tau}, \omega_t^i) - \bar{g}^i(\mathbf{B}_{t-\tau}, \omega_t^i)\right)(\omega_t^i)^\top \omega_{t-\tau}^k\rangle$$
$$\quad - \mathbb{E}\langle \mathbf{B}_{t-\tau}\omega_{t-\tau}^k - z^{k,*}, \left(g^i(O_t^i, \mathbf{B}_{t-\tau}, \omega_{t-\tau}^i) - \bar{g}^i(\mathbf{B}_{t-\tau}, \omega_{t-\tau}^i)\right)(\omega_t^i)^\top \omega_{t-\tau}^k\rangle$$
$$\quad + \mathbb{E}\langle \mathbf{B}_{t-\tau}\omega_{t-\tau}^k - z^{k,*}, \left(g^i(O_t^i, \mathbf{B}_{t-\tau}, \omega_{t-\tau}^i) - \bar{g}^i(\mathbf{B}_{t-\tau}, \omega_{t-\tau}^i)\right)(\omega_t^i)^\top \omega_{t-\tau}^k\rangle$$
$$\quad - \mathbb{E}\langle \mathbf{B}_{t-\tau}\omega_{t-\tau}^k - z^{k,*}, \left(g^i(O_t^i, \mathbf{B}_{t-\tau}, \omega_{t-\tau}^i) - \bar{g}^i(\mathbf{B}_{t-\tau}, \omega_{t-\tau}^i)\right)(\omega_{t-\tau}^i)^\top \omega_{t-\tau}^k\rangle$$
$$=\mathbb{E}\langle \mathbf{B}_{t-\tau}\omega_{t-\tau}^k - z^{k,*}, \left(g^i(O_t^i, \mathbf{B}_{t-\tau}, \omega_t^i) - g^i(O_t^i, \mathbf{B}_{t-\tau}, \omega_{t-\tau}^i)\right)(\omega_t^i)^\top \omega_{t-\tau}^k\rangle$$
$$\quad - \mathbb{E}\langle \mathbf{B}_{t-\tau}\omega_{t-\tau}^k - z^{k,*}, \left(\bar{g}^i(\mathbf{B}_{t-\tau}, \omega_t^i) - \bar{g}^i(\mathbf{B}_{t-\tau}, \omega_{t-\tau}^i)\right)(\omega_t^i)^\top \omega_{t-\tau}^k\rangle$$
$$\quad + \mathbb{E}\langle \mathbf{B}_{t-\tau}\omega_{t-\tau}^k - z^{k,*}, \left(g^i(O_t^i, \mathbf{B}_{t-\tau}, \omega_{t-\tau}^i) - \bar{g}^i(\mathbf{B}_{t-\tau}, \omega_{t-\tau}^i)\right)(\omega_t^i - \omega_{t-\tau}^i)^\top \omega_{t-\tau}^k\rangle$$
$$\leq (16U_\omega^3 + 8U_r U_\omega^3)\|\omega_t^i - \omega_{t-\tau}^i\|.$$

Step 4:

Conditioning on the past trajectory $v_{0:t-\tau}$, the only random part is the sample $O_t^k$ we draw from the Markov chain at the current timestep, and the sample we draw from the steady-state distribution $\mu^i \otimes \pi \otimes P^i$. By the uniform ergodicity assumption, we can bound the total variation distance:

$$\mathbb{E}\langle \mathbf{B}_{t-\tau}\omega_{t-\tau}^k - z^{k,*}, \left(g^i(O_t^i, \mathbf{B}_{t-\tau}, \omega_{t-\tau}^i) - \bar{g}^i(\mathbf{B}_{t-\tau}, \omega_{t-\tau}^i)\right)(\omega_{t-\tau}^i)^\top \omega_{t-\tau}^k\rangle$$
$$=\mathbb{E}\left[\mathbb{E}\left[\langle \mathbf{B}_{t-\tau}\omega_{t-\tau}^k - z^{k,*}, \left(g^i(O_t^i, \mathbf{B}_{t-\tau}, \omega_{t-\tau}^i) - \bar{g}^i(\mathbf{B}_{t-\tau}, \omega_{t-\tau}^i)\right)(\omega_{t-\tau}^i)^\top \omega_{t-\tau}^k\rangle \mid v_{0:t}\right]\right]$$
$$=\mathbb{E}\left[\langle \mathbf{B}_{t-\tau}\omega_{t-\tau}^k - z^{k,*}, \mathbb{E}\left[\left(g^i(O_t^i, \mathbf{B}_{t-\tau}, \omega_{t-\tau}^i) - \bar{g}^i(\mathbf{B}_{t-\tau}, \omega_{t-\tau}^i)\right) \mid v_{0:t}\right](\omega_{t-\tau}^i)^\top \omega_{t-\tau}^k\rangle\right]$$
$$\leq 4U_\omega^3 d_{TV}(P_{t-\tau:t}^i(\cdot, \cdot, \cdot), \mu^i \otimes \pi \otimes P^i)$$
$$\leq 4U_\omega^3 m\rho^{\tau-1}.$$

Combining the above, we get

$$\frac{1}{K}\sum_{i=1}^K \mathbb{E}\langle x_t^k, \xi_t^i(\omega_t^i)^\top \omega_t^k\rangle \leq (8U_\omega^4 + 4(U_r + U_\omega)U_\omega^2)\|\mathbf{B}_t - \mathbf{B}_{t-\tau}\|$$

$$+ (12U_r U_\omega^2 + 12U_\omega^3)\|\omega_t^k - \omega_{t-\tau}^k\| + \frac{1}{K}\sum_{i=1}^K (16U_\omega^3 + 8U_r U_\omega^3)\|\omega_t^i - \omega_{t-\tau}^i\| + 4U_\omega^3 m\rho^{\tau-1}.$$

$$\square$$

