# OpenReview forum: "Cooperative TD Learning in Heterogeneous Environments via Joint Linear Approximation"
_ICLR.cc/2026/Conference — ICLR 2026 Conference Withdrawn Submission_

### Official Review · Reviewer_4TJf · 2025-10-28

**Soundness:** 2
**Presentation:** 1
**Contribution:** 2
**Rating:** 4
**Confidence:** 4

**Summary:**

This paper studies the problem of cooperative TD learning in heterogeneous environments, where the value functions of $K$ agents share an unknown $r$-dimensional linear subspace. The paper proposes a Federated Single-Timescale TD algorithm (Algorithm 1), where agents iteratively estimate this common subspace and their respective local "heads".

The main contribution is a finite-time convergence analysis under Markovian sampling, showing that reward estimation and subspace estimation errors decay at rates of $\mathcal{O}(\log T / \sqrt{T})$ and $\mathcal{O}(r \log T / \sqrt{T})$, respectively. The analysis suggests the method is resilient to heterogeneity but does not show a clear acceleration from cooperation.

**Strengths:**

1. The paper delivers a finite-time convergence analysis for the proposed FSTTD algorithm (Algorithm 1) under Markovian sampling, which is a high standard in RL theory.

2. The paper identifies a core technical difficulty: the lack of a direct contraction for the evolution of the principal angle distance ($m_t$) between the estimated and optimal subspaces.

**Weaknesses:**

1. The overall presentation of the manuscript falls significantly below the expected standard. The reviewer encountered substantial difficulty in reading and understanding the paper due to numerous undefined notations and typographical errors. A major issue lies in the fact that the paper investigates the infinite-horizon average-reward setting rather than the more common discounted-reward formulation. However, this critical distinction is never explicitly clarified in the text. As a result, Equations (2)–(3) are particularly hard to interpret for readers without prior knowledge of average-reward reinforcement learning, since many of the symbols and assumptions appear to be taken for granted.

2. The novelty and overall contribution of this work appear limited. As discussed in the related work section, both MARL and personalized federated RL (PFRL) have already been extensively explored. Although most existing studies focus on the discounted-reward setting, extending these frameworks to the average-reward formulation does not, by itself, constitute a sufficiently novel contribution. The primary methodological idea, introducing a shared low-dimensional “subspace” structure, is directly inspired by the PFLg approach of Collins et al. (2021). The technical development mainly involves combining this subspace-based representation with standard analytical tools for handling Markovian sampling in conventional MARL literature.

3. The main contribution is a finite-time convergence analysis under Markovian sampling, showing that reward estimation and subspace estimation errors decay at rates of $\mathcal{O}(\log T / \sqrt{T})$ and $\mathcal{O}(r \log T / \sqrt{T})$, respectively. However, the dependece with the number of agents K and heterogeneity level is unknown.  Moreover, the analysis suggests the method is resilient to heterogeneity but does not show a clear acceleration from cooperation.

4. Assumption 4 (that $span(B^\*)$ is $A^k$, invariant for all $k$) is a very strong assumption. It requires that the dynamics of all agents (encoded in $A^k$, which depends on $P^k$ and $\phi(s)$) must "preserve" the same unknown shared subspace $B^\*$. This assumption is used critically in the proof  to make a difficult cross-term disappear ($P_{\perp}^{*}A^{k}P^{\*}=0$), which is essential for deriving the contraction in the subspace error.  The paper provides no discussion of when Assumption 4 might hold in practice, even in simple synthetic MDPs, or how much it limits the scope of problems being solved. A similar assumption may hold in PFL like Collins et al. (2021) in a federated supervised learning setting, but it is not true for RL setting.

5. An $\mathcal{O}(1)$ error term means the algorithm, in theory, cannot get arbitrarily close to the optimal value function. The authors attribute this to the "gap" from QR decomposition and projection.

6. A major downside of this paper is the lack of empirical validation. Can the authors show the advantage of the proposed algorithm over existing benchmarks?

7. Many issues in mathematical notations and formulations. Just name a few in below:

what is $\eta$ in Eq. (3)?
The TD(0) update for $z_{t+1}^k$ in Eq. (3) appears to be missing the feature vector $\phi(s_t^k)$.
What does Eq. (10) mean?

**Questions:**

Please see the Weakness above.

---

### Official Review · Reviewer_vPG2 · 2025-10-30

**Soundness:** 3
**Presentation:** 4
**Contribution:** 4
**Rating:** 6
**Confidence:** 3

**Summary:**

This paper studies cooperative temporal-difference (TD) learning with heterogeneous agents. the existence of a common subspace may accelerate the learning of individual agents, yet heterogeneity in state transition kernels can lead to misaligned learning signals across agents . The authors try to answer the question: do the benefits of collaboration outweigh the drawbacks . An algorithm is devised in which agents iteratively estimate the common subspace and local heads, and the convergence of the algorithm is analyzed. Theoretical results show that the proposed algorithm can filter out conflicting signals and mitigate the negative impacts of misaligned signals with certain assumptions.

**Strengths:**

1. The core question this paper tries to answer is interesting and meaningful. To investigate the beneficial or harmful effects of multiple agents cooperating is essential for practical applications.

2. The idea of personalized federated learning is introduced to devise the algorithm, which is straightforward and novel.

3. The theoretical analysis is solid, and provide insights to understand the interactions of multiple agents when they cooperate in heterogeneous environments.

4. The theoretical results show that the convergence rate does not decrease in the number of agents, which suggests that the algorithm does not suffer from the heterogeneity.

5. The proof sketch provides a good intuition to understand how the authors deal with the coupled error terms in the convergence analysis.

**Weaknesses:**

1. The theoretical results show that the cooperation of multiple agents does not speedup the convergence rate of TD algorithm. The proposed algorithm does not exploit the benefits of low dimensional subspace ($r<<d$).

2. No empirical results are provided. Experimental analysis using whether synthetic or real-world data will make the work more solid and complete.

3. Not enough explanation for some assumptions. The convergence might not be guaranteed (since $\bar{X}_T = O(1)$) if the gap between between the raw updates and the updates after QR decomposition and projection. The gap requires more dedicated treatment.

Typo: line 86/87, “under the heterogeneity” should be removed.

**Questions:**

1. How the heterogeneity of environments is characterized? It seems that the paper only considers multiple agents with different transition matrix $P^k$. But how different $P^k$'s are from each other? Maybe for different agent, $P^k$ is associated with different $\rho$ in assumption 1. An extreme example: there are an agent with $\rho=0.9$, and all others with $\rho=0.1$. The agent with $\rho=0.9$ might decrease the convergence rate compared with the case where all agents are with $\rho=0.1$.

2. What does assumption 4 mean? Apart from mathematical descriptions, can you provide more explanation on what does it mean in the multiple agent learning system. Does it hold in practice or is it just for the simplification of theoretical analysis.

3. In Theorem 1, there is an unavoidable O(1) error for the weight. What is the explicit form of this error term? If it is too large, the weight might not converge at all, which might destroy the convergence of the whole system (i.e., the convergence analysis might be vacuous). Can you eliminate the step for QR decomposition of $B$ to get rid of this error term. Or maybe more advanced techniques are required to deal with the error term from the QR decomposition.

4. Is it possible to leverage the dimension information $r$ (assume that we know it a prior, or we enforce a $r$-dimension subspace to approximate the optimal subspace) in the algorithm do achieve speedup.

---

### Official Review · Reviewer_zMJJ · 2025-11-01

**Soundness:** 3
**Presentation:** 3
**Contribution:** 2
**Rating:** 2
**Confidence:** 4

**Summary:**

This paper studies federated temporal-difference (TD) learning among heterogeneous agents, each interacting with distinct environments, and learning jointly their value functions. Assuming that agents’ optimal parameters lie in a shared low-dimensional subspace, the authors propose a single-timescale federated TD algorithm that jointly estimates a shared subspace and agent-specific heads. They provide a finite-time convergence analysis under Markovian sampling, showing an $O(\log T / \sqrt{T})$ rate for the reward estimation error.

**Strengths:**

- The paper addresses a timely and relevant problem: personalised federated TD learning under heterogeneous environments. The work provides non-asymptotic convergence guarantees for cooperative TD under Markovian noise.

- The theoretical analysis is careful, mathematically sound, and the assumptions are clearly stated.

**Weaknesses:**

-  $\textbf{Main concern:}$ The proposed method closely mirrors the  PFEDTD-REP algorithm introduced in [1]. That earlier work already analyzed a two-timescale federated TD algorithm with shared representations and proved linear convergence speedup with respect to the number of agents under Markovian noise.
    In contrast, the current paper adopts a single-timescale variant and explicitly states that no acceleration or speedup is achieved (see Lines 91–100 of the submission). Thus, the contribution is primarily incremental: removing the two-timescale update but offering weaker theoretical performance.

- The proposed method appears to address a $\textit{distributed}$ setting rather than a truly $\textit{federated}$ one. In particular, there are no mechanisms such as local updates, compression, or partial participation that are typically used in federated learning to reduce communication complexity. Adding mechanisms to reduce the communication complexity would strengthen the paper.

 - The paper provides no experimental validation or comparison against prior works such as PFEDTD-REP[1], making it difficult to evaluate the practical benefits of the proposed single-timescale approach.

- The following two important works on heterogeneous federated (personalized) TD learning are missing in the related works section:
    - Guojun Xiong et al, On the linear speedup of personalised Federated reinforcement learning with shared representations, ICLR 2025
    - Paul Mangold et al, SCAFFLSA: Taming Heterogeneity in Federated Linear Stochastic Approximation and TD Learning, NeurIPS 2024.


    [1] Guojun Xiong et al, On the linear speedup of personalised Federated reinforcement learning with shared representations, ICLR 2025

**Questions:**

- How does the proposed cooperative single-timescale TD method differ technically from the PFEDTD-REP algorithm in [1]?
Specifically, what new theoretical or algorithmic insights does this paper provide beyond removing the two-timescale design?


- Since the paper explicitly mentions that no acceleration or speedup is achieved, could the authors clarify whether the absence of linear speedup is fundamental to the single-timescale formulation, or a limitation of the current analysis?
Would modifying the assumptions (e.g., relaxing bounded features) allow for a speedup result comparable to [1]?


- The proposed algorithm resembles a distributed averaging scheme rather than a federated learning setup.
Could the authors clarify whether their framework supports typical federated mechanisms such as partial participation, local updates, or communication compression? How would such extensions affect the convergence guarantees?

- Could the authors include empirical comparisons with PFEDTD-REP or other federated TD baselines to illustrate the practical implications of the single-timescale design?
Are there any qualitative benefits that compensate for the lack of speedup?

---

### Official Review · Reviewer_pMr4 · 2025-11-05

**Soundness:** 2
**Presentation:** 4
**Contribution:** 1
**Rating:** 2
**Confidence:** 4

**Summary:**

This paper studies the problem of cooperative TD learning, where multiple agents aim to collaboratevily estimate the value function of a shared policy in the average-reward setting.
The proposed method simultaneously estimates (i) the projection from the full-dimensional space to a reduced-dimension space, (ii) the weights in the underlying projection space.
This allows flexibility between the global and the personalized settings, depending on the size of the projection subspace.
A theoretical convergence analysis is performed, showing convergence under the assumption that a common structure exist, showing that the method is able to exploit this structure without require knowledge of it.

**Strengths:**

1. The proposed method is general enough to encompass both global federated TD learning and personalized TD learning.
2. In the case where a low-dimensional structure exist, the method is able to retrieve it without requiring prior knowledge of it.
3. Convergence rates are given in a general case where noise is Markovian.

**Weaknesses:**

1. The proposed approach does not lead to linear speed-up: the author claim that this may be due to the assumption that features maps are uniformly bounded by $1$, which may obfuscate the dependency on the dimension. This is surprising especially since similar work manage to obtain linear speed-up:
   - in [1], a shared feature map is directly learned with linear speed-up; while the approaches are different, it seems that learning the feature map should be a more difficult problem;
   - in federated TD, [2] obtain linear speed-up with the exact same assumption on the feature maps (but without personalization);
   - in federated learning (not RL), a very similar algorithm was studied by [3], with linear speed-up.
Although all these approaches indeed differ from the setting considered here, it suggests that the lack of linear speed-up may be an artefact of the analysis. It seems that these references are also missing from the discussion of related work.
1. The assumption that there exists a low-dimensional subspace containing all solutions seem to be a bit artificial.
2. Some of the theoretical results are difficult to interepret, providing a convergence rate of order $O(1/\sqrt{T})$ up to a $O(1)$ error, which dominates all other terms. In its current form, this may invalidate many of the theoretical results presented in the paper.

[1] Personalized Federated Learning Ali Dadras, Sebastian U. Stich, Alp Yurtsever, TMLR 2025.

[2] SCAFFLSA: Taming Heterogeneity in Federated Linear Stochastic Approximation and TD Learning, Paul Mangold, Sergey Samsonov, Safwan Labbi, Ilya Levin, Reda Alami, Alexey Naumov, Eric Moulines, NeurIPS 2024.


[3] On the Linear Speedup of Personalized Federated Reinforcement Learning with Shared Representations, Guojun Xiong, Shufan Wang, Daniel Jiang, Jian Li, ICLR 2025.

**Questions:**

1. How could assumption 3 be adapted to uncover the linear speed-up? What could be other reasons for the lack of speed-up?
2. Are there practical problems where the personalized solutions lie within a subspace of low dimension?
3. Is it possible to give an explicit, non-asymptotic expression for the additive error terms $O(1)$ that appear in multiple of the derived upper bounds?

There is a typo in Assumption 1 ("expLoration")

---

### Note · Authors · 2025-11-12

I have read and agree with the venue's withdrawal policy on behalf of myself and my co-authors.